# UNDERSTANDING THE PERFORMANCE GAP IN PREFERENCE LEARNING: A DICHOTOMY OF RLHF AND DPO

## ABSTRACT

We present a fine-grained theoretical analysis of the performance gap between reinforcement learning from human feedback (RLHF) and direct preference optimization (DPO) under a representation gap. Our study decomposes this gap into two sources: an explicit representation gap under exact optimization and an implicit representation gap under finite samples. In the exact optimization setting, we characterize how the relative capacities of the reward and policy model classes influence the final policy qualities. We show that RLHF, DPO, or online DPO can outperform one another depending on type of model mis-specifications. Notably, online DPO can outperform both RLHF and standard DPO when the reward and policy model classes are isomorphic and both mis-specified. In the approximate optimization setting, we provide a concrete construction where the ground-truth reward is implicitly sparse and show that RLHF requires significantly fewer samples than DPO to recover an effective reward model—highlighting a statistical advantage of two-stage learning. Together, these results provide a comprehensive understanding of the performance gap between RLHF and DPO under various settings, and offer practical insights into when each method is preferred.

## 1 INTRODUCTION

Reinforcement learning from human feedback (RLHF, Christiano et al. (2017); Ziegler et al. (2019)) is an important paradigm improving the natural language understanding and generation capabilities of large language models (LLMs). The core idea of RLHF is to utilize pair-wise comparison between responses from human annotators, as directly collecting absolute reward signals is hard. There are two stages in RLHF: the reward modeling stage and the policy optimization stage. The reward modeling stage assumes human preferences follow the Bradley-Terry (BT) model (Bradley and Terry, 1952), allowing a prompt-response pair to be assigned a scalar reward. Thus, a reward model $r_\phi$ could be trained using negative log-likelihood loss function from human preferences. In the policy optimization stage, the base LM is "online" fine-tuned with RL algorithms such as proximal policy optimization (PPO, Schulman et al. (2017)), based on $r_\phi$ under a Kullback-Leibler (KL) divergence-regularized bandit setting. And the key assumption behind this two-stage pipeline is the *realizability* of the ground-truth reward.

The above RLHF paradigm falls inside a broader problem, preference-based policy learning (Wirth et al., 2017). Another popular algorithm in this area is direct preference optimization (DPO, Rafailov et al. (2023)), which utilizes the closed-form solution (assuming *realizability* as well) for the policy optimization stage to bypass the reward modeling stage and directly fine-tune the base LM as a policy model $\pi_\theta$ using the preference dataset. Due to its inherent supervised learning (offline and RL-free) nature, DPO training is more stable than RLHF. And its iterative online version (Guo et al., 2024; Dong et al., 2024) has been shown to have better convergence rates (Shi et al., 2025), and milder coverage conditions (Song et al., 2024; Xiong et al., 2024), than vanilla DPO. The key assumption behind DPO's design is the *realizability* of the closed-form solution of the optimal policy.

Notably, in the foundational work of preference learning (Zhu et al., 2023), the ground-truth reward is assumed to lie in a linear model class; and in Rafailov et al. (2023), both the reward class and policy class are *tabular parameterized*, making their optimal solutions realizable. The *realizability*

condition is commonly assumed in theoretical studies of preference learning (Xiong et al., 2024; Shi et al., 2025; Feng et al., 2025; Yao et al., 2025; Swamy et al., 2025), or DPO-style algorithm designs to derive the loss functions for neural policy classes (Azar et al., 2023; Zhou et al., 2024; Liu et al., 2024b; Xu et al., 2024a). Importantly, under the *realizability* assumption, it is straightforward to derive the equivalence between the ideal performances of RLHF and DPO (Swamy et al., 2025).

However, the assumptions of *tabular parameterization* and *realizability* often do not hold in practice, particularly when the reward model is significantly smaller than the policy model (e.g., 6B vs. 175B in Ouyang et al. (2022), indicating a clear disparity in representational capacity), when the policy model class is heavily restricted due to limited computational resources, or when the reward model is sub-optimal owing to limited preference data. These situations are examples of *model mis-specification*, a common issue in practice due to limitations in model capacity or data. Consequently, one should not expect DPO to perform identically to RLHF under model mis-specifications. This motivates the central question of our investigation:

*Under what conditions is DPO equivalent, superior, or inferior to RLHF in performance?*

To quantify the problem, we choose the performance metric as the expected value of the original regularized bandit problem using the ground-truth reward $r^\star$ ($x$ is a prompt, and $y$ is a response): $V_{r^\star}^\pi := \mathbb{E}_{x\sim\rho}\left[\mathbb{E}_{y\sim\pi(\cdot|x)}[r^\star(x,y)] - \beta\mathsf{KL}\left(\pi(\cdot|x)\|\pi_{\mathsf{ref}}(\cdot|x)\right)\right]$, where $\rho$ is a pre-fixed distribution over prompts, $\pi$ is a distribution over responses given prompts, and $\pi_{\mathsf{ref}}$ is a fixed reference policy. Let $\pi^\star := \mathrm{argmax}_\pi V_{r^\star}^\pi$ be the ideal optimal policy.

**Our contributions.** We study the performance differences between two-stage RLHF and DPO under a representation gap, from an optimization perspective. Our contributions are listed as follows:

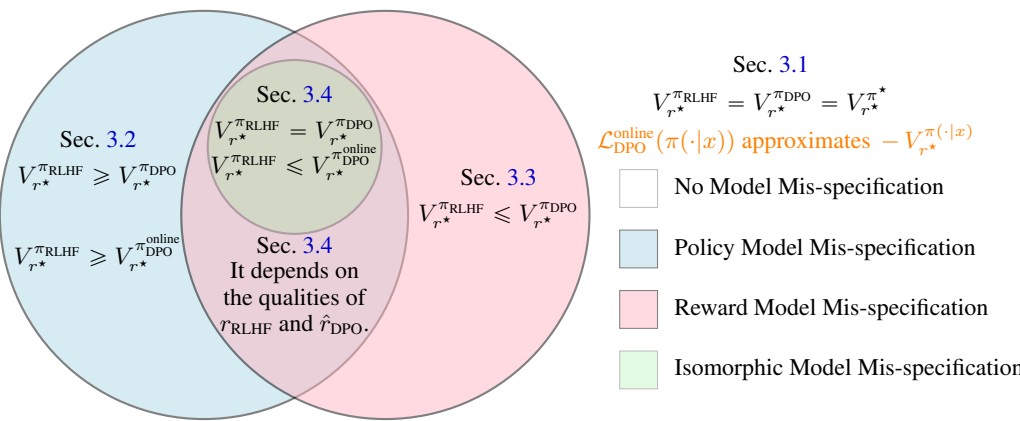

Figure 1: Main results on performance gap induced by model mis-specification scenarios.

• When assuming ***exact optimization***, *i.e.*, optimization with infinite data, we study the ***fine-grained representation gap*** under different settings of ***reward and policy class mis-specifications*** in Section 3. Main results are visualized in Figure 1.

① *No model mis-specification:* We show that the RLHF and DPO policies both achieve the performance of $\pi^\star$, and online DPO can further close the gap between optimization paths.

② *Policy model mis-specification:* We show that the RLHF policy is still optimal under the model class, while the DPO policy can be sub-optimal, and online DPO cannot bridge the gap.

③ *Reward model mis-specification:* We show that the DPO policy is still optimal, while the RLHF policy can be sub-optimal due to learning based on a sub-optimal reward model.

④ *Double model mis-specification:* When policy and reward model classes are isomorphic, then they should have identical performance, while online DPO can outperform both of them. Otherwise, there is no consistent performance gap, and the comparison result depends on the qualities of (surrogate) reward models. We also give a preliminary guide for reward learning under mis-specifications.

• For *approximate optimization*, *i.e.*, the finite-sample regime, we study the ***implicit representation gap*** incurred by ***statistical efficiencies*** in Section 4. We construct a simple task where the ground-truth reward to is a dual-token linear function with feature dimension $d$ and implicit sparsity $k$, and the total number of samples is $n$. Even without mis-specifications, we can reveal a separation between RLHF and DPO under this setting: the estimation error of DPO is $\Omega(d/n)$, while reward learning in RLHF can effectively leverage sparsity, decreasing the error to $\tilde{\mathcal{O}}(\sqrt{k \log d/n})$. This result indicates that DPO is less data-efficient than RLHF, leading to inferior performance.

Finally, we conduct numerical experiments to corroborate these theoretical findings in Section 5.

## 2 PRELIMINARIES

**Notation.** Let $\sigma : \mathbb{R} \to \mathbb{R}$ be the sigmoid function, where $\sigma(x) = 1/(1 + \exp(-x))$. For any set $\mathcal{X}$, $\Delta(\mathcal{X})$ represents the set of probability distributions over $\mathcal{X}$. sg $()$ is the stopping-gradient operator, where $\nabla_\theta[\text{sg}(f(\theta))] = \mathbf{0}$. Let $e_k$ be a one-hot vector with 1 on its $k^{\text{th}}$ entry and 0 on other entries. For any vector $x$, let $x_k$ be its $k^{\text{th}}$ entry. We use $f(\theta) \overset{\nabla}{=} g(\theta)$ to indicate $\nabla_\theta f(\theta) = \nabla_\theta g(\theta)$.

**Bandits and Policies.** A bandit is defined by a state space $\mathcal{X}$, an action space $\mathcal{Y}$, and a reward function $r : \mathcal{X} \times \mathcal{Y} \to \mathbb{R}$. A policy $\pi : \mathcal{X} \to \Delta(\mathcal{Y})$ represents a probability distribution over actions given a state. Note that, we sometimes omit the prompt $x$ for simplicity, so that $\pi \in \Delta(\mathcal{Y})$.

**Model class and value function.** Let $\mathcal{F} = \{r_\phi : \phi \in \mathbb{R}^{d_R}\}$ denote the reward model class, and $\Pi = \{\pi_\theta : \theta \in \mathbb{R}^{d_P}\}$ denote the policy model class, where $d_R, d_P \in \mathbb{N}$. For a reward function $r$ and policy $\pi$, we define the regularized value function as:

$$V_r^{\pi(\cdot|x)} := \left[ \mathop{\mathbb{E}}_{y \sim \pi(\cdot|x)}[r(x,y)] - \beta \, \mathsf{KL}\left(\pi(\cdot|x) \| \pi_{\text{ref}}(\cdot|x)\right) \right], \quad V_r^\pi := \mathop{\mathbb{E}}_{x \sim \rho} V_r^{\pi(\cdot|x)},$$

where $\beta > 0$ is the regularization coefficient, $\rho \in \Delta(\mathcal{X})$ is a pre-fixed distribution over prompts, and $\pi_{\text{ref}}$ is a fixed reference policy. Let $r^\star$ denote the ground-truth reward function, and $\pi^\star$ denote the optimal policy for $V_{r^\star}^\pi$. A well-known fact (Rafailov et al., 2023) is that $\pi^\star(y|x) = \pi_{\text{ref}}(y|x) \exp(r^\star(x,y)/\beta)/Z(x)$, where $Z(x) := \sum_{y \in \mathcal{Y}} \pi_{\text{ref}}(y|x) \exp(r^\star(x,y)/\beta)$ is the partition function. The goal of preference-based policy learning is to find a policy $\pi_\theta \in \Pi$ that maximizes $V_{r^\star}^{\pi_\theta}$. We define the oracle value as $V_{r^\star}^\Pi := \max_{\pi \in \Pi} V_{r^\star}^\pi$.

**Bradley-Terry (BT) model.** Given an implicit reward oracle $r : \mathcal{X} \times \mathcal{Y} \to \mathbb{R}$, Bradley and Terry (1952) assume that human preference distribution $p^\star : \mathcal{X} \times \mathcal{Y} \times \mathcal{Y} \to \Delta(\{0,1\})$ satisfies:

$$p^\star(y_1 > y_2|x) = \sigma\left(r^\star(x, y_1) - r^\star(x, y_2)\right).$$

This means response $y_1$ is favored over $y_2$ with probability $p^\star(y_1 > y_2|x)$ by human annotators.

**Human preference dataset.** In practice, people first collect a pair dataset $\mathcal{D}^\dagger = \{x^{(i)}, y_1^{(i)}, y_2^{(i)}\}_{i=1}^n$, and then ask human annotators to label these pairs to get a human preference dataset $\mathcal{D} = \{x^{(i)}, y_w^{(i)}, y_l^{(i)}\}_{i=1}^n$. Following BT model, $y_1^{(i)}$ is preferred over $y_2^{(i)}$ given prompt $x^{(i)}$, (*i.e.* $y_w = y_1$ and $y_l = y_2$), w.p. $p^\star(y_1^{(i)} > y_2^{(i)}|x^{(i)})$.

**Two-stage approach of RLHF.** RLHF proceeds in two stages. First, the reward learning stage finds a reward model $r_{\text{RLHF}} \in \mathcal{F}$ by maximizing the population MLE objective:

$$r_{\text{RLHF}} = \underset{r_\phi \in \mathcal{F}}{\text{argmax}} \mathop{\mathbb{E}}_{x \sim \rho; y,y' \sim \pi_{\text{ref}}(\cdot|x)} \sum_{\{y_1,y_2\}=\{y,y'\}} p^\star(y_1 > y_2|x) \log \sigma(r_\phi(x, y_1) - r_\phi(x, y_2)).$$

And for approximate optimization, $r_{\text{RLHF}}$ is estimated from a finite human preference dataset. Then using the reward model $r_{\text{RLHF}}$, the policy learning stage returns $\pi_{\text{RLHF}} = \text{argmax}_{\pi \in \Pi} V_{r_{\text{RLHF}}}^\pi$.

**Direct approach of DPO.** By leveraging the surrogate reward $\hat{r}_\theta(x, y) := \beta \log \frac{\pi_\theta(x,y)}{\pi_{\text{ref}}(x,y)}$, DPO bypasses reward learning and directly learns the policy from preference data:

$$\pi_{\text{DPO}} = \underset{\pi_\theta \in \Pi}{\text{argmax}} \mathop{\mathbb{E}}_{x \sim \rho; y,y' \sim \pi_{\text{ref}}(\cdot|x)} \sum_{\{y_1,y_2\}=\{y,y'\}} p^\star(y_1 > y_2|x) \log \sigma\left(\hat{r}_\theta(x, y_1) - \hat{r}_\theta(x, y_2)\right).$$

For approximate optimization, $\pi_{\text{DPO}}$ is estimated from a finite human preference dataset. We also consider an online variant of DPO (Xiong et al., 2024), where the pairwise data are sampled from a distribution $\pi^{\text{s}}$ which could depend on the current policy. It then minimizes the modified loss:

$$\mathcal{L}_{\text{DPO}}^{\text{online}}(\pi_\theta(\cdot|x)) = - \mathop{\mathbb{E}}_{y,y' \sim \text{sg}(\pi^{\text{s}}(\cdot|x))} \sum_{\{y_1,y_2\}=\{y,y'\}} p^\star(y_1 > y_2|x) \log \sigma\left(\hat{r}_\theta(x,y_1) - \hat{r}_\theta(x,y_2)\right) .$$

## 3 Exact Optimization: Fine-grained Performance Gap Induced by Model Mis-specification

We analyze the behavior of RLHF and DPO in the idealized setting of exact optimization, where both methods have access to infinite preference data and can optimize their respective objectives without statistical or computational error. Recall that $r_{\text{RLHF}} \in \mathcal{F}$ is the solution computed by exact optimization of reward learning, $\pi_{\text{RLHF}} \in \Pi$ is the solution computed by exact optimization of policy learning given $r_{\text{RLHF}}$, and $\pi_{\text{DPO}} \in \Pi$ is the solution computed by exact optimization of DPO. We can bound the sub-optimality of each algorithm using the mis-specification error (see calculations in Appendix C.11), but in this section our focus is on the performance gap induced by model mis-specification, that is, the difference between the best policy each method can produce, as determined by the expressiveness of the reward and policy model classes.

### 3.1 No Model Mis-specification

We begin with the fully realizable setting, where both the ground-truth reward function and the optimal policy lie within their respective model classes. While this assumption is often unrealistic in practice, it serves as a clean baseline and has been the main focus of most prior theoretical analyses (Xiong et al., 2024; Shi et al., 2025; Feng et al., 2025; Swamy et al., 2025).

**Condition 1** (Strong Reward Model, Strong Policy Model). $r^\star \in \mathcal{F}$, $\pi^\star \in \Pi$.

Both RLHF and DPO are capable of recovering the true optimal policy under ideal conditions. In this regime, RLHF directly optimizes $V_{r^\star}^{\pi_\theta}$ in the policy learning stage. Proof deferred to Appendix C.1.

**Proposition 1.** *Under Condition 1, $V_{r^\star}^{\pi_{\text{RLHF}}} = V_{r^\star}^{\pi_{\text{DPO}}} = V_{r^\star}^{\Pi}$.*

Although RLHF and DPO share a same solution, they differ in optimization trajectories and convergence rates. Shi et al. (2025) propose a sampling strategy to accelerate convergence in online DPO, and Feng et al. (2025) further refine this approach, showing its connection to the RLHF objective from a gradient-based perspective. Below, we show a result which is analogous to Theorem 4.1 in (Feng et al., 2025), but from the objective perspective rather than the gradient perspective.

**Definition 1** (PILAF Sampler (Shi et al., 2025; Feng et al., 2025)). *PILAF Sampler is a probabilistic mixture of two sampler pairs:*

$$① \begin{cases} \pi^{\text{s1}}(y|x) = \pi_\theta(y|x) , \\ \pi^{\text{s2}}(y|x) = \pi_\theta(y|x) , \end{cases} ② \begin{cases} \pi^{\text{s1}}(y|x) \propto \pi_\theta^{1+\beta}(y|x)\pi_{\text{ref}}^{-\beta}(y|x) , \\ \pi^{\text{s2}}(y|x) \propto \pi_\theta^{1-\beta}(y|x)\pi_{\text{ref}}^{\beta}(y|x) , \end{cases}$$

*with a ratio $\alpha_1 = 1$ and $\alpha_2 = \mathop{\mathbb{E}}_{y,y' \sim \pi_\theta} \exp(\hat{r}_\theta(x,y) - \hat{r}_\theta(x,y')) .$*

**Remark 1.** Given a prompt $x$, we first randomly choose a sampler pair: select sampler ① w.p. $\alpha_1/(\alpha_1 + \alpha_2)$ and sampler ② otherwise. Then sample $y_1 \sim \pi^{\text{s1}}(\cdot \mid x)$ and $y_2 \sim \pi^{\text{s2}}(\cdot \mid x)$.

**Theorem 2.** *Given $R_{\max}, \delta \in \mathbb{R}_+, x \in \mathcal{X}$, s.t. $0 \leqslant r^\star(x,y) \leqslant R_{\max}$, $\forall y \in \mathcal{Y}$, and $|(r^\star(x,y) - r^\star(x,y')) - (\hat{r}_\theta(x,y) - \hat{r}_\theta(x,y'))| \leqslant \delta$, $y,y' \in \mathcal{Y}$, then with $\pi^{\text{s}}$ defined in Definition 1, we have:*

$$\mathcal{L}_{\text{DPO}}^{\text{online}}(\pi_\theta(\cdot|x)) \stackrel{\nabla}{=} \frac{2\beta}{\text{sg}(Z_\theta(x))} \left\{ -\left[ \mathop{\mathbb{E}}_{y \sim \pi_\theta(\cdot|x)}[r(x,y)] - \beta\text{KL}(\pi_\theta(\cdot|x) \| \pi_{\text{ref}}(\cdot|x)) \right] \right.$$

$$\left. + \frac{1}{4\beta} \mathop{\mathbb{E}}_{y,y' \sim \text{sg}(\pi_\theta(\cdot|x))} \left[ \epsilon_{y,y'} \cdot \left[ (r^\star(x,y) - r^\star(x,y')) - (\hat{r}_\theta(x,y) - \hat{r}_\theta(x,y')) \right]^2 \right] \right\} ,$$

*where $\epsilon_{y,y'} \in \mathbb{R}$ are noises s.t. $|\epsilon_{y,y'}| \leqslant \frac{\delta}{6\sqrt{3}\sigma'(R_{\max}+\delta)}$ and $Z_\theta(x) := \mathop{\mathbb{E}}_{y,y' \sim \pi_\theta(\cdot|x)} 1/\sigma'(\hat{r}_\theta(x,y) - \hat{r}_\theta(x,y'))$ can be viewed as adaptive step sizes for different prompts.*

**Remark 2.** This result indicates that, with an appropriate sampler, the objective of online DPO can approximate the true value function in prompt level. However, the second-order deviation can become substantial when $R_{\max}$ is large, or the ground-truth reward is poorly fitted. In such scenarios, the objective of online DPO may significantly deviate from the value function, leading to degraded convergence or even divergence. Proof deferred to Appendix C.2.

## 3.2 POLICY MODEL MIS-SPECIFICATION

We now examine the setting where the ground-truth reward function is realizable ($r^\star \in \mathcal{F}$), but the optimal policy is non-realizable by the policy class ($\pi^\star \notin \Pi$). This case can be referred to Nika et al. (2024), who point out that the optimal policy could be more complicated than the optimal reward, and Swamy et al. (2025), who attribute this scenario to generation-verification gaps in fine-tuning.

**Condition 2** (Strong Reward Model, Weak Policy Model). $r^\star \in \mathcal{F}$, $\pi^\star \notin \Pi$.

In this case, RLHF has a structural advantage: it can recover the exact reward and then compute the best possible policy within $\Pi$. In contrast, DPO bypasses reward modeling and directly learns a policy from preferences, which may lead to sub-optimal behavior due to mismatches between preference-based objectives and reward-based value functions. The following proposition provides a concrete example where DPO fails to recover the best achievable policy, even under exact optimization. Proof deferred to Appendix C.3.

**Proposition 3.** *Under Condition 2, $V_{r^\star}^{\Pi} = V_{r^\star}^{\pi_{\text{RLHF}}} \geqslant V_{r^\star}^{\pi_{\text{DPO}}}$, and there exists an environment s.t. $V_{r^\star}^{\pi_{\text{RLHF}}} > V_{r^\star}^{\pi_{\text{DPO}}}$.*

Furthermore, we show that online DPO cannot close this gap, even when equipped with PILAF sampler. A numerical proof is deferred to Appendix C.8.

**Proposition 4.** *Under Condition 2, $V_{r^\star}^{\pi_{\text{RLHF}}} \geqslant V_{r^\star}^{\pi_{\text{DPO}}^{\text{online}}}$, and there exists an environment s.t. $V_{r^\star}^{\pi_{\text{RLHF}}} > V_{r^\star}^{\pi_{\text{DPO}}^{\text{online}}} = V_{r^\star}^{\pi_{\text{DPO}}}$ where the online sampler is PILAF sampler (Definition 1).*

**Remark 3.** Our key insight is that a strict performance gap between RLHF and DPO can exist under policy model mis-specification, and importantly, even sophisticated samplers like PILAF may fail to close the gap, an important nuance that, to our knowledge, has been overlooked in prior studies.

## 3.3 REWARD MODEL MIS-SPECIFICATION

We now consider the setting where the ground-truth reward function $r^\star$ is not realizable by the reward model class $\mathcal{F}$, while the optimal policy $\pi^\star$ lies within the policy class $\Pi$. As discussed in Swamy et al. (2024), two-stage RLHF can only lose information during reward learning, which will be highlighted under reward model mis-specification.

**Condition 3** (Weak Reward Model, Strong Policy Model). $r^\star \notin \mathcal{F}$, $\pi^\star \in \Pi$.

In this setting, RLHF is vulnerable to reward mis-specification: the learned mis-specified reward model $r_{\text{RLHF}}$ could significantly deviate from the ground-truth reward $r^\star$, causing the subsequent policy optimization to yield a sub-optimal solution even though $\pi^\star \in \Pi$. Conversely, DPO has a clear advantage: it can directly fit a policy to the observed preference data and thus recover $\pi^\star$ without incurring reward modeling error. Proof deferred to Appendix C.4.

**Proposition 5.** *Under Condition 3, $V_{r^\star}^{\pi_{\text{RLHF}}} \leqslant V_{r^\star}^{\pi_{\text{DPO}}} = V_{r^\star}^{\Pi}$, and there exists an environment s.t. $V_{r^\star}^{\pi_{\text{RLHF}}} < V_{r^\star}^{\pi_{\text{DPO}}}$.*

**Observation under token-level parameterization.** To assess the practicality of Condition 3 for auto-regressive language models, we specialize our general bandit model to the token-level parameterization. In this setting, the optimal policy admits the closed-form characterization of Rafailov et al. (2024), which we restate with an explicit separation between $\pi_{\text{ref}}$ and the $q^\star$ function (see Appendix C.11 for details):

$$\pi^\star(y_t|x, y_{0...t-1}) \propto \pi_{\text{ref}}(y_t|x, y_{0...t-1}) \exp\left( \frac{q^\star(y_t|x, y_{0...t-1})}{\beta} \right) , \tag{1}$$

where the $q^\star$ function is determined in a recursive way:

$$q^\star(y_t|x, y_{0\ldots t-1}) = \begin{cases} \beta \log \sum_{s \in \mathcal{V}} \pi_{\mathsf{ref}}(s|x, y_{0\ldots t}) \exp(q^\star(s|x, y_{0\ldots t})/\beta) & y_t \text{ is not the terminal token;} \\ r^\star(x, y_{0\ldots t}) & y_t \text{ is the terminal token,} \end{cases}$$

$\mathcal{V}$ is the vocabulary, and $s \in \mathcal{V}$ is the token. This observation shows that while the reward model in RLHF only needs to approximate $r^\star$, the policy model in DPO must capture the token-level $q^\star$ function, which recursively entangles the reward signal with the base model $\pi_{\mathsf{ref}}$. As a result, the policy model faces a substantially more demanding learning objective, making it more prone to mis-specification than the reward model of the same scale. and suggesting that the "weak reward, strong policy model" regime may be less common in practice.

## 3.4 DOUBLE MODEL MIS-SPECIFICATION

We now consider the most challenging setting, where neither the ground-truth reward function nor the optimal policy is realizable by their respective model classes.

**Condition 4** (Weak Reward Model, Weak Policy Model). $r^\star \notin \mathcal{F}$, $\pi^\star \notin \Pi$.

To enable a fine-grained comparison between RLHF and DPO under this double mis-specified regime, we introduce the surrogate reward model class induced by the policy class as $\mathcal{F}_\Pi = \{\hat{r}_\theta : \theta \in \mathbb{R}^{d_P}, \hat{r}_\theta(x, y) = \beta \log \frac{\pi_\theta(y|x)}{\pi_{\mathsf{ref}}(y|x)}, \forall x \in \mathcal{X}, y \in \mathcal{Y}\}$. Pairwise preferences depend only on reward differences, so reward functions are equivalent if they differ by a constant. We compare the expressiveness of the original reward model class $\mathcal{F}$ and the surrogate class $\mathcal{F}_\Pi$, modulo constant shifts, and analyze three representative regimes characterizing their relative capacities:

**Condition 5** (Isomorphism). $r^\star \notin \mathcal{F}$, $\pi^\star \notin \Pi$. $\mathcal{F} = \mathcal{F}_\Pi$.

**Condition 6** (Policy Model Class Is Relatively Stronger). $r^\star \notin \mathcal{F}$, $\pi^\star \notin \Pi$. $\mathcal{F} \subset \mathcal{F}_\Pi$.

**Condition 7** (Reward Model Class Is Relatively Stronger). $r^\star \notin \mathcal{F}$, $\pi^\star \notin \Pi$. $\mathcal{F} \supset \mathcal{F}_\Pi$.

**Remark 4.** Note that certain cases involve partially overlapping model classes. However, we do not consider these intermediate regimes for the sake of a principled analysis.

**Analysis of the isomorphic case.** Condition 5 indicates the scenario when the reward model class and policy model class are *isomorphic*—meaning there exists a shared parameterization or a deterministic mapping between rewards and policies. This structure allows us to directly compare RLHF and DPO when both operate under the same representational constraints, and to investigate whether bypassing reward modeling, as in DPO, provides any advantage. In RLHF, reward learning is decoupled from the current policy, and thus lacks access to its distributional information; while DPO can mitigate this limitation through online sampling. Therefore, RLHF under Condition 5 is comparable to offline DPO, but could underperform online DPO. Proofs deferred to Appendices C.5 and C.9.

**Proposition 6.** *Under Condition 5, $V_{r^\star}^{\pi_{\mathrm{RLHF}}} = V_{r^\star}^{\pi_{\mathrm{DPO}}}$.*

**Proposition 7.** *Under Condition 5, there exists an environment where online DPO can produce a solution $\pi_{\mathrm{DPO}}^{\mathrm{online}}$, s.t. $V_{r^\star}^{\pi_{\mathrm{RLHF}}} < V_{r^\star}^{\pi_{\mathrm{DPO}}^{\mathrm{online}}}$.*

On the other hand, under Conditions 6 and 7, either method may outperform the other depending on the environment. Proofs deferred to Appendices C.6 and C.7.

**Proposition 8.** *Under Condition 6, there exists an environment s.t. $V_{r^\star}^{\pi_{\mathrm{RLHF}}} < V_{r^\star}^{\pi_{\mathrm{DPO}}}$, and another environment s.t. $V_{r^\star}^{\pi_{\mathrm{RLHF}}} > V_{r^\star}^{\pi_{\mathrm{DPO}}}$.*

**Proposition 9.** *Under Condition 7, there exists an environment s.t. $V_{r^\star}^{\pi_{\mathrm{RLHF}}} > V_{r^\star}^{\pi_{\mathrm{DPO}}}$, and another environment s.t. $V_{r^\star}^{\pi_{\mathrm{RLHF}}} < V_{r^\star}^{\pi_{\mathrm{DPO}}}$.*

Though there is no consistent performance gap between RLHF and DPO in certain settings, revisiting the framework can reveal a structural parallel: RLHF can yield the best policy given the learned reward model $r_{\mathrm{RLHF}}$, and the DPO policy is directly the optimal one given the surrogate reward model $\hat{r}_{\mathrm{DPO}}$. And online DPO serves to enhance the quality of $\hat{r}_{\mathrm{DPO}}$ (Xiong et al., 2024). Formally,

$$\pi_{\mathrm{RLHF}} = \operatorname*{argmax}_{\pi \in \Pi} V_{r_{\mathrm{RLHF}}}^\pi \ , \ \pi_{\mathrm{DPO}} = \operatorname*{argmax}_{\pi \in \Pi} V_{\hat{r}_{\mathrm{DPO}}}^\pi \ . \tag{2}$$

This result implies a general principle: the performance gap is reflected in the quality gap between the (surrogate) reward models: $r_{\text{RLHF}}$ and $\hat{r}_{\text{DPO}}$. Better reward learning yields higher expected value.

As revealed in Appendix C of Ouyang et al. (2022) and Section 3.3 of Swamy et al. (2025), it is uncommon to deploy a reward model with a larger scale than the policy model. And thus to ensure practical relevance, we focus on the regime $\mathcal{F} \subseteq \mathcal{F}_\Pi$, and pose the following relevant question:

*What key property enables a (surrogate) reward model to subsequently help learn good policies?*

As an answer to this question, we note that in the context of preference learning, the reward model quality can be measured using an $\ell_2$ distance of pairwise difference, derived by simple calculations:

$$V_{r_\phi}^{\pi_{\theta^\star(r_\phi)}} \text{ can be measured by } -\mathbb{E}_{y,y'\sim\mathsf{sg}\left(\pi_{\theta^\star(r_\phi)}\right)}\left[(r^\star(y)-r^\star(y'))-(r_\phi(y)-r_\phi(y'))\right]^2 \ , \quad (3)$$

where $\pi_{\theta^\star(r_\phi)} := \operatorname{argmax}_{\pi\in\Pi} \ V_{r_\phi}^\pi$ and we omit prompts for simplicity. Detailed calculations and further discussions deferred to Appendix B. Using this metric, we can further establish a separation.

**Concluding remarks.** Although we adopt relatively simple techniques, these results can provide valuable insights for the fundamental differences between RLHF and DPO. In the next section, we demonstrate that these insights extend naturally to more practical and realistic scenarios.

## 4 APPROXIMATE OPTIMIZATION: PERFORMANCE GAP INDUCED BY STATISTICAL EFFICIENCY DIFFERENCES IN REWARD LEARNING

With limited preference data, we are not able to directly compute exact solutions, and thus obtain weaker reward models and policy models due to estimation error. This scenario can be viewed as inducing an implicit model mis-specification, whose effects have been widely discussed in Section 3.4. And since we can only lose information in reward learning (Swamy et al., 2025), Equation (2) still holds asymptotically with on-policy sampling. Thus by assuming $F \subseteq F_\Pi$, we only need to compare the reward model quality measure shown in Equation (3). We adopt an empirical proxy for this notion, data-induced semi-norm (details in Definition 2 in Appendix C.10, see also Zhu et al. (2023)): $\frac{1}{n}\sum_{i=1}^n \left[(r^\star(y_w^{(i)})-r^\star(y_l^{(i)}))-(r_\phi(y_w^{(i)})-r_\phi(y_l^{(i)}))\right]^2$, where $\mathcal{D} = \{(y_w^{(i)}, y_l^{(i)})\}_{i=1}^n$ is an empirical preference dataset and we omit prompts from now on.

**Difference in token-level linear parameterization.** In this section, to rigorously establish a separation, we focus on a specific token-level linear parameterization, which is a special case of the general bandit model; therefore, previous results continue to hold. The common reward model shares the same architecture with LM but replaces the last layer with a linear head, *i.e.*, it takes the whole prompt-response pair as the input and predicts one value. Therefore, if we view the last-layer hidden state as the feature vector, it is natural to assume the reward model to be parameterized as a linear MDP model[1]: $r_{\theta_r}(y) = \beta\sum_{t=0}^{|y|-1}\theta_{r,t}^\top\psi(y_{0\dots t})$, where $\theta_{r,t}, \psi(y_{0\dots t}) \in \mathbb{R}^d$. While for the policy model, one needs to go through the softmax results of all tokens and multiply them[2]:

$$\pi_{\theta_p}(y) = \prod_{t=0}^{|y|-1}\pi_{\theta_{p,t}}(y_t|y_{0\dots t-1}) = \prod_{t=0}^{|y|-1}\frac{\pi_{\text{ref}}(y_t|y_{0\dots t-1})\exp(\theta_{p,t}^\top\psi(y_{0\dots t}))}{\sum_{s\in\mathcal{V}}\pi_{\text{ref}}(s|y_{0\dots t-1})\exp(\theta_{p,t}^\top\psi(y_{0\dots t-1},s))} \ ,$$

where $\theta_{p,t} \in \mathbb{R}^d$, and the surrogate reward model is $\hat{r}_{\theta_p}(y) = \beta\sum_{t=0}^{|y|-1}\log\pi_{\theta_{p,t}}(y_t|y_{0\dots t-1})$. Let the ground truth reward be $r^\star(y) = \beta\sum_{t=0}^{|y|-1}(\theta_t^\star)^\top\psi(y_{0\dots t})$, then the optimal solution for the reward model is $\theta_{r,t}^\star = \theta_t^\star$. And recall Equation (1), the optimal solution for the policy model is:

$$\pi_{\theta_{p,t}^\star}(y_t|y_{0\dots t-1})\propto\pi_{\text{ref}}(y_t|y_{0\dots t-1})\exp\left(\frac{q^\star(y_t|y_{0\dots t-1})}{\beta}\right) \ .$$

---

[1]It is also common to assume the reward model to be a linear bandit model (Zhu et al., 2023), while the stronger linear MDP model assumption here is for fair comparison with the following policy model.

[2]Our parameterization assumption on the token-level policy model is different from Razin et al. (2025a), which utilizes a form of token matrix, since we intend to ensure that $d_P = d_R$.

Benefiting from the token-level $q^\star$ function, models trained in this way can simulate a process reward model to provide fine-grained information (Yuan et al., 2024; Cui et al., 2025; Shi et al., 2024; Xu et al., 2025). However, simultaneously, learning the $q^\star$ function sacrifices statistical efficiency due to the need to model the complicated structure. Next, we will present a concrete example to illustrate the statistical gap between pure reward learning and surrogate reward learning.

> **Dual-token sparse prediction (DTSP) task.** Let $\mathcal{V}$ be the vocabulary, and $\mathcal{Y} = \mathcal{V}^2$. The policy model is required to sequentially output two tokens $a, b$, and the ground-truth reward is:
>
> $$r^\star(a, b) = \beta \mathbf{r}_{\text{sparse}}^\top \psi(a) + \beta e_1^\top \psi(a, b) ,$$
>
> where $a, b \in \mathcal{V}$, $\psi(a), \psi(a, b) \in \mathbb{R}^d$, $\mathbf{r}_{\text{sparse}} \in \mathbb{R}^d$, $\|\mathbf{r}_{\text{sparse}}\|_0 = k$, and $k \ll d$.

We let $\theta_r^\star$ denote the optimal solution for pure reward learning, and $\theta_p^\star$ the optimal solution for surrogate reward learning. Note that for the second token, $\theta_r^\star$ and $\theta_p^\star$ share the same optimal solution:

$$(\theta_{r,1}^\star)^\top \psi(a, b) = e_1^\top \psi(a, b) + C_1 , \quad (\theta_{p,1}^\star)^\top \psi(a, b) = e_1^\top \psi(a, b) + C_2 ,$$

where $C_1, C_2 \in \mathbb{R}$ are offsets. And for the first token $a$, there is a distinction:

$$(\theta_{r,0}^\star)^\top \psi(a) = \mathbf{r}_{\text{sparse}}^\top \psi(a) + C_3 ,$$

$$(\theta_{p,0}^\star)^\top \psi(a) = \log \mathbb{E}_{w \sim \pi_{\text{ref}}(\cdot|a)} \exp(r^\star(a, b)/\beta) + C_4$$

$$= \mathbf{r}_{\text{sparse}}^\top \psi(a) + \log \mathbb{E}_{w \sim \pi_{\text{ref}}(\cdot|a)} \exp(\psi(a, b)_1) + C_4 ,$$

where $\mathbf{r}_{\text{sparse}}$ gets entangled with $\pi_{\text{ref}}$ in $\theta_{p,0}^\star$. Note that if $\log \mathbb{E}_{w \sim \pi_{\text{ref}}(\cdot|a)} \exp(\psi(a, b)_1)$ can be mapped to certain non-linear function of $\psi(a)$, then the policy model is mis-specified while the reward model is not, as in Condition 2. And even without explicit model mis-specification, we can establish a separation in (surrogate) reward model qualities due to statistical efficiency differences.

**Theorem 10** (Informal). *Under token-level linear parameterization and mild assumptions, there exists an environment for DTSP task, s.t. by estimating from a preference dataset $\mathcal{D}$ with $n$ samples under $\theta_1 = e_1$ constraint, the estimation error of the reward model $\hat{\theta}_r$ can be reduced to $\tilde{\mathcal{O}}(\sqrt{k \log d/n})$ using a (computationally efficient) $\ell_1$-regularized estimator, i.e., w.p. $1 - \delta$,*

$$\frac{1}{n} \sum_{i=1}^n \left[ (r^\star(y_w^{(i)}) - r^\star(y_l^{(i)})) - (r_{\hat{\theta}_r}(y_w^{(i)}) - r_{\hat{\theta}_r}(y_l^{(i)})) \right]^2 = \mathcal{O}\left( \sqrt{\frac{k \log(d) + k \log(1/\delta)}{n}} \right) ,$$

*while the estimation error of the DPO model $\hat{\theta}_p$ is lower bounded by $\Omega(d/n)$:*

$$\frac{1}{n} \sum_{i=1}^n \left[ (r^\star(y_w^{(i)}) - r^\star(y_l^{(i)})) - (\hat{r}_{\hat{\theta}_p}(y_w^{(i)}) - \hat{r}_{\hat{\theta}_p}(y_l^{(i)})) \right]^2 = \Omega\left( \frac{d}{n} \right) .$$

**Remark 5.** By fixing the optimal $\theta_1$, which is relatively easier to estimate, we can reduce the dual-token prediction problem to a single-token prediction problem, where $\theta_{r,0}^\star$ is sparse while $\theta_{p,0}^\star$ is dense. Leveraging the results of Yao et al. (2025) then yields the separation. Formal statement and detailed proof deferred to Appendix C.10.

**Theorem 11** (informal). *Based on Theorem 10, there exists an environment for DTSP task, s.t. we have a separation on the sub-optimality of RLHF and DPO:*

$$V_{r^\star}^{\pi^\star} - V_{r^\star}^{\pi_{\text{RLHF}}} = \tilde{\mathcal{O}}\left( \sqrt[4]{\frac{k \log d}{n}} \cdot \sqrt{\Lambda_1} \right) ,$$

$$V_{r^\star}^{\pi^\star} - V_{r^\star}^{\pi_{\text{DPO}}} = \Omega\left( \frac{d}{n} \cdot \Lambda_2 \right) ,$$

*where $\pi_{\text{RLHF}} = \arg\max_{\pi \in \Pi} V_{r_{\hat{\theta}_r}}^\pi$, $\pi_{\text{DPO}} = \pi_{\hat{\theta}_p}$, and $\Lambda_1, \Lambda_2$ are geometric quantities of data. Formal statement and detailed proof deferred to Appendix C.10.5.*

**Concluding remarks.** This section shows that the estimation error can also induce an implicit model mis-specification. From the perspective of sparse recovery, we can see that the DPO could suffer from severe statistical inefficiency compared with pure reward learning, even with the same model scale. Although our task construction is specific, it reveals a general phenomenon: DPO can distort the intrinsic structure of the true reward function. For general policy model class beyond log-linear model class, Equation (1) still holds. This observation shows that the policy model must learn the $q^\star$ function, while the reward model only needs to learn the reward. Because $q^\star$ mixes both $r^\star$ and $\pi_{\text{ref}}$, the policy model faces a more complex target, making it more vulnerable to model mis-specification and sample inefficiency. And to prevent policy model mis-specification, $d_P$ is often required to be larger than $d_R$, which further leads to increased sample complexity. Given the insight that real-world rewards are often sparse and simple (Yao et al., 2025), we can infer that the reward model's quality typically surpasses that of the surrogate reward model. This further explains why two-stage RLHF is empirically observed to outperform DPO (Ivison et al., 2024; Xu et al., 2024b).

## 5 EXPERIMENTAL VERIFICATIONS

**Experiment setup.** We now verify our analysis in practical settings. We consider one common dataset, `PKU-SafeRLHF` (Ji et al., 2023). We first fine-tune a **GPT-2-LARGE-774M** model (Radford et al., 2019) on 5k samples of `PKU-SafeRLHF-QA`, and obtain the **SFT** model. We adopt the **GPT2-LARGE-HARMLESS** model (Yang et al., 2024) as the ground-truth reward oracle. All experiments are repeated for 3 seeds. Please see Appendix D for more details.

**Implementation details.** For exact optimization, we compute the exact BT loss using the ground-truth reward oracle for each pair in the DPO training dataset. For approximate optimization, we instead compute the empirical BT loss. We adopt a pairwise regression surrogate instead of PPO to improve training stability: $\mathcal{L}_{\text{RL}}(\theta) = \mathbb{E}_{y_1, y_2 \sim \text{sg}(\pi_\theta)} \left[ (r(y_1) - r(y_2)) - (\hat{r}_\theta(y_1) - \hat{r}_\theta(y_2)) \right]^2$. During deployment, the reward score will be scaled by a coefficient $r_{\text{margin}}$. Besides, since PILAF sampler (see Definition 1) is very close to purely online sampler when $\beta = 0.1$, we directly sample $y_1, y_2 \sim \pi_\theta$ in the implementation of online DPO.

**Verifications of Section 3.** We train online DPO and RLHF on `PKU-SafeRLHF-Prompt`, following the practice of Dong et al. (2024); Shi et al. (2025). For the strong reward condition, we directly adopt the **GPT2-LARGE-HARMLESS** model as a perfectly-learned reward model. For the weak reward condition, we train the **SFT** model on `PKU-SafeRLHF-safer` by replacing the projection matrix with a linear head, freezing all layers except the linear head and the last block. For the strong policy condition, we fully train the **SFT** model, while for the weak policy condition, we freeze the first half of the blocks of the **SFT** model. Results are shown in Figures 2 and 3. The empirical findings align closely with our theoretical predictions:

- Figure 2 (Condition 1) aligns with Proposition 1 and theorem 2: increasing the reward scale amplifies the second-order deviation in online DPO's objective, causing larger deviation from the RLHF optimum, as our theory predicts.

- Figure 3 (left, Condition 2) confirms Propositions 3 and 4: with a realizable reward model but restricted policy class, RLHF outperforms DPO.

- Figure 3 (middle, Condition 3) confirms Proposition 5: with a mis-specified reward model but realizable policy class, DPO outperforms RLHF.

- Figure 3 (right, Condition 4) exhibits behavior consistent with our double-mis-specification analysis: relative performance can depend on the comparative expressive power of $\mathcal{F}$ versus $\mathcal{F}_\Pi$. In our setup, the reward model is less expressive, leading RLHF to underperform.

**Verifications of Section 4.** We train DPO and reward learning on `PKU-SafeRLHF`, following the practice of Zhou et al. (2024). We train on two types of preference: "better" and "safer", and down-sample the corresponding training datasets to 1k-9k samples. For DPO training, we directly train the **SFT** model using DPO; while for pure reward learning, we replace the projection matrix of the **SFT** model with a linear head. The models are trained under the same setting, and all achieve at least 85% training accuracy. Results are shown in Figure 4, demonstrating that as the number of samples decreases, reward learning outperforms surrogate reward learning across two tasks. This corroborates our theoretical separation result in Theorem 10: pure reward learning is statistically more sample-efficient than the surrogate reward learning performed by DPO.

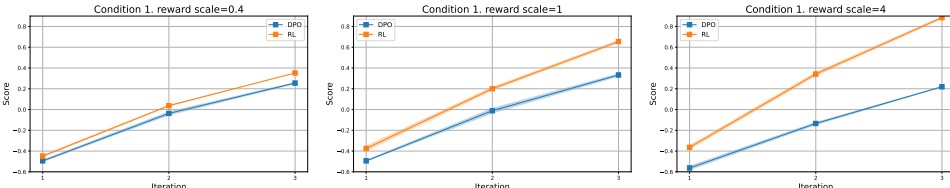

Figure 2: **Experimental Results for Condition 1.** Experiments with different reward scales $\{0.4, 1, 4\}$ align with Theorem 2: as the reward scale increases, the second-order deviation in the online DPO objective grows, giving RLHF a clear advantage.

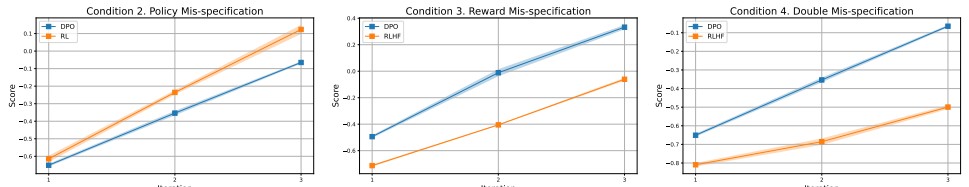

Figure 3: **Experimental Results for Conditions 2 to 4.** The first two plots (Conditions 2 and 3) are consistent with Propositions 3 and 5. The gap in the last plot can be attributed to the mis-specified reward model being too weak.

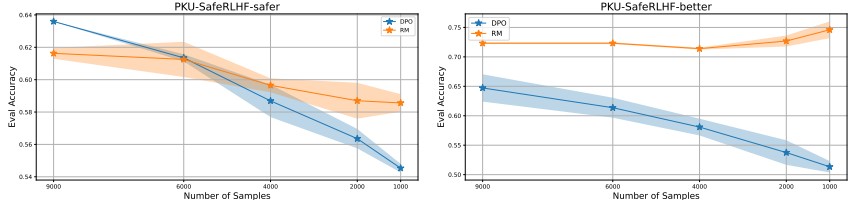

Figure 4: **Experimental Results on Statistical Efficiency.** We experiment on two preference types. Pure reward learning is shown to be more data-efficient than surrogate reward learning.

## 6 RELATED WORK

Due to page limit, a comprehensive review of related work is deferred to Appendix A. Here, we focus on comparing with the most relevant prior study, Nika et al. (2024). First, unlike Nika et al. (2024) which chooses the un-regularized value function as the performance metric, we adopt the regularized version for two reasons: 1) it is the shared original optimization goal of RLHF and DPO, so our choice is to ensure fairness; 2) it can help circumvent the unavoidable upper bound of policy bias in the unregularized version. Second, we provide a fine-grained analysis of different model mis-specifications under exact optimization, *i.e.*, more detailed comparative analysis on reward approximation error and $\mathcal{O}(\mathsf{KL}\left(\pi_{\theta_{\mathrm{DPO}}} \| \pi^\star\right))$ when $n \rightarrow +\infty$, and our results are not limited to linear reward and log-linear policy model classes. Third, we improve the statistical analysis of Nika et al. (2024) on DPO ($\Theta(d_P/n)$) and RLHF ($\Theta(\sqrt{d_R/n})$), and show that even when $d_P = d_R = d$ and under realizability assumption, there can still be a large gap between DPO ($\Omega(d/n)$) and RLHF ($\tilde{\mathcal{O}}(\sqrt{k \log d/n})$) where $k \ll d$ is the parameter sparsity.

## 7 CONCLUSION

This paper provides a fine-grained analysis of the performance gap between two-stage and direct approaches to preference-based policy learning. We theoretically demonstrate a dichotomy of RLHF and DPO under different mis-specification scenarios, and further reveal an implicit representation gap induced by statistical efficiency. Our claims are supported by empirical experiments on LMs.

It is also important to acknowledge our limitations. 1) While we identify a limitation of training reward models based on BT model, we do not provide a theoretically grounded and practically effective alternative. 2) Due to computational constraints, our experiments are limited to small-scale models. We hope our insights can motivate the community to further investigate these directions.

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

# Appendix

## Table of Contents

# A    SUPPLEMENTARY RELATED WORKS

**Reinforcement learning from human feedback (RLHF).** Seminal contributions that showcased RLHF's applicability to LLMs include foundational work by Christiano et al. (2017), and subsequent research focusing on tasks such as summarization (Stiennon et al., 2020), instruction following (Ouyang et al., 2022), question answering using web-retrieved information (Nakano et al., 2021), and broader AI alignment objectives (Bai et al., 2022). Theoretical studies of RLHF include pessimism in policy learning (Zhu et al., 2023), overoptimization (Zhu et al., 2024; Liu et al., 2024c), online RLHF (Xiong et al., 2024; Song et al., 2024), robustness (Mandal et al., 2025), and reward models (Wang et al., 2024; Razin et al., 2025b; Huang et al., 2025; Yao et al., 2025).

**Direct preference optimization (DPO).** There is a rich literature studying offline (Rafailov et al., 2024; Feng et al., 2024), iterative (Dong et al., 2024; Liu et al., 2024a), and online (Guo et al., 2024; Tajwar et al., 2024; Ding et al., 2024; Shi et al., 2025; Chen et al., 2025; Feng et al., 2025) DPO. There are other DPO-style algorithms to directly optimize the policy model from preference signals, such as $\Psi$-PO (Azar et al., 2023), RSO (Liu et al., 2024b), RS-DPO (Khaki et al., 2024), CPO (Xu et al., 2024a), SimPO (Meng et al., 2024), XPO (Xie et al., 2024), VPO (Cen et al., 2024), and OAIF (Guo et al., 2024).

**Performance gap between RLHF and DPO.** Recently, there have been works investigating the performance gap between RLHF and DPO policies. Xu et al. (2024b) found that DPO might find biased solutions that exploit out-of-distribution responses, and iterative DPO might be a better approach; meanwhile, PPO with advantage normalization, large batch-size, and exponential moving update of the reference model can consistently outperform DPO on benchmarks.

Swamy et al. (2025) first showed that when the reward class and policy class are isomorphic, RLHF and DPO output policies with equal performances. Then, they proposed a hypothesis that when the ground-truth reward is simpler than the soft optimal policies, and the reward class reduces the sample complexity to learn such a reward, then reward modeling essentially reduces the policy search space. This hypothesis is supported by their experiments. In our work, we comprehensively *extend* upon their first class isomorphic result by studying model mis-specification (Section 3), and we construct concrete examples to further *support* the existence of the "simpler ground-truth reward" and "reduced sample complexity" (Section 4).

Nika et al. (2024) provided sub-optimality upper bounds for RLHF and DPO when assuming linear reward class and log-linear policy class, with the *un-regularized* value as performance metric. Three cases were studied: 1) realizable ground-truth reward and exact optimization, 2) realizable ground-truth reward but approximate optimization, as well as 3) non-realizable reward and exact optimization. Let $n$ be the size of the fixed dataset and $d$ be the feature dimension. For case 1, both algorithms have a policy bias due to the un-regularized metric, while RLHF has an additional $\Theta(\sqrt{d/n})$ statistical error and that for DPO is $\Theta(d/(\beta n))$. For case 2, RLHF and DPO both obey a linear convergence to statistical errors and policy biases when using projected gradient descent. For case 3, aside from statistical errors and policy biases, RLHF has an extra approximation error between the ground-truth reward and best achievable reward, while DPO has an extra bias between the optimal regularized policy and the ideal optimal regularized policy.

## B  BONUS: HOW CAN WE BETTER MODEL REWARD FROM PREFERENCE SIGNALS?

As motivated by Equation (2), a reward model $r_\phi$ can be mapped to a policy via:

$$\pi_{\theta^\star(r_\phi)} := \underset{\pi \in \Pi}{\arg\max}\, V_{r_\phi}^\pi = \underset{\pi \in \Pi}{\arg\max}\, \underset{y \sim \pi}{\mathbb{E}} \left[ r_\phi(y) \right] - \beta\, \mathsf{KL}\left( \pi \| \pi_{\mathsf{ref}} \right) \ .$$

If $\mathcal{F} \subseteq \mathcal{F}_\Pi$, this solution further admits the closed form $\pi_{\theta^\star(r_\phi)}(y) = \pi_{\mathsf{ref}}(y) \exp\left( r_\phi(y)/\beta \right) / Z(\phi)$, where $Z(\phi) := \sum_{y \in \mathcal{Y}} \pi_{\mathsf{ref}}(y) \exp\left( r_\phi(y)/\beta \right)$ is the partition function. If the goal is to output a policy that performs well under the ground-truth reward $r^\star$, then reward learning should aim to find a model $r_\phi$ such that the resulting policy $\pi_{\theta^\star(r_\phi)}$ maximizes the underlying "real" objective:

$$r_{\phi^\star} = \underset{r_\phi \in \mathcal{F}}{\arg\max}\, V_{r^\star}^{\pi_{\theta^\star(r_\phi)}} = \underset{r_\phi \in \mathcal{F}}{\arg\min}\, \underbrace{-\beta \log Z(\phi) - \underset{y \sim \pi_{\theta^\star(r_\phi)}}{\mathbb{E}} \left[ r^\star(y) - r_\phi(y) \right]}_{=:\mathcal{L}_{\mathrm{new}}(\phi)} \ .$$

Following the policy gradient theorem (Sutton et al., 1999), the gradient of this new objective is (see detailed calculations in Appendix C.11):

$$\nabla_\phi \mathcal{L}_{\mathrm{new}}(\phi) = -\frac{1}{2} \underset{y,y' \sim \pi_{\theta^\star(r_\phi)}}{\mathbb{E}} \left[ \nabla_\phi r_\phi(y) - \nabla_\phi r_\phi(y') \right] \left[ (r^\star(y) - r^\star(y')) - (r_\phi(y) - r_\phi(y')) \right] \ , \quad (4)$$

which corresponds to the gradient of an $\ell_2$ distance of pairwise difference:

$$\mathcal{L}_{\mathrm{new}}(\phi) \overset{\nabla}{=} \frac{1}{4} \underset{y,y' \sim \mathsf{sg}\left(\pi_{\theta^\star(r_\phi)}\right)}{\mathbb{E}} \left[ (r^\star(y) - r^\star(y')) - (r_\phi(y) - r_\phi(y')) \right]^2 \ . \quad (5)$$

**Comparison with MLE.** The reward model $r_\phi$ are typically learned via MLE from preference data, which does not consider the fact that the learned reward will ultimately be used to induce a policy. Let the distribution of the preference data be $\mu$ (by default $\mu$ is $\pi_{\mathsf{ref}}$, but can be any distribution here). Now revisit the MLE objective:

$$\mathcal{L}_{\mathrm{MLE}}(\phi) = - \underset{y,y' \sim \mu}{\mathbb{E}} \left[ \sigma(r^\star(y) - r^\star(y')) \log \sigma(r_\phi(y) - r_\phi(y')) + \sigma(r^\star(y') - r^\star(y)) \log \sigma(r_\phi(y') - r_\phi(y)) \right] \ ,$$

whose gradient is (see detailed calculations in Appendix C.11):

$$\nabla_\phi \mathcal{L}_{\mathrm{MLE}}(\phi) = - \underset{y,y' \sim \mu}{\mathbb{E}} \left[ \nabla_\phi r_\phi(y) - \nabla_\phi r_\phi(y') \right] \left[ \sigma(r^\star(y) - r^\star(y')) - \sigma(r_\phi(y) - r_\phi(y')) \right] \ . \quad (6)$$

Following Equation (6), we can see that the gradient of DPO is

$$\nabla_\theta \mathcal{L}_{\mathrm{DPO}}(\theta) \propto - \mathbb{E}_{y,y' \sim \mu} [\sigma(r^\star(y) - r^\star(y')) - \sigma(\hat{r}_\theta(y) - \hat{r}_\theta(y'))][\nabla(\hat{r}_\theta(y) - \hat{r}_\theta(y'))] \ ,$$

and the gradient of reward modeling is

$$\nabla_\phi \mathcal{L}_{\mathrm{RM}}(\phi) \propto - \mathbb{E}_{y,y' \sim \mu} [\sigma(r^\star(y) - r^\star(y')) - \sigma(r_\phi(y) - r_\phi(y'))][\nabla(r_\phi(y) - r_\phi(y'))] \ .$$

Comparing Equation (4) with Equation (6), a natural idea is to apply Taylor's expansion to extract the $\sigma(\cdot)$ in Equation (6) to further align it with Equation (4). And this will induce an additional coefficient $\sigma'(r_\phi(y) - r_\phi(y'))$ on the data distribution $\mu(y, y')$. And this by-product explains why is PILAF sampler (a variant of online sampler, see Definition 1) introduced to align the distorted distribution $\tilde{\mu}(y, y') \propto \mu(y, y') \cdot \sigma'(r_\phi(y) - r_\phi(y'))$ with $\pi_{\theta^\star(r_\phi)}$. If the reward model is a surrogate reward model, then we can directly deploy PILAF sampler or online sampler; while if it is a pure reward model, then we can implement PILAF sampler or online sampler through logit mixing (Shi et al., 2024; Xu et al., 2025) only when it can provide token-level reward information. However, it is worth noting that model mis-specification can lead the second-order Taylor remainder to be extremely large, as shown in Theorem 2. Therefore, when faced with a representation gap, it could be beneficial to train the (surrogate) reward model on a distribution close to PILAF sampler but is still limited.

To alleviate this issue, we could learn the preference with alternative modeling approaches to circumvent the BT model setting, which has already shown success in Sun et al. (2025); Calandriello et al. (2024). For example, we can look into the training objective of online IPO (Calandriello et al., 2024; Zhou et al., 2025) (see detailed calculations in Appendix C.11):

$$\mathcal{L}_{\mathrm{IPO}}^{\mathrm{online}}(\theta) \overset{\nabla}{=} \underset{(y_1,y_2) \sim \mathsf{sg}(\rho_\theta)}{\mathbb{E}} \left[ (\hat{r}_\theta(y_1) - \hat{r}_\theta(y_2)) - \frac{p^\star(y_1 > y_2) - p^\star(y_2 > y_1)}{2} \right]^2 \ ,$$

where $\rho(\theta)$ is an online sampling distribution, and it thus can optimize an $\ell_2$ distance in an online way. The classification model deployed in Sun et al. (2025) is also promising. We leave this interesting direction for future exploration.

## C  OMITTED PROOFS

Note that in this section, we omit all prompts without loss of generality. For the constructive proof, we can set the number of states to 1; for the other proofs, we can simply sum over different prompts to extend them to the general case.

### C.1  PROOF OF PROPOSITION 1

Since $r^\star \in \mathcal{F}$, RLHF exactly recovers $r^\star$ during reward learning. The policy optimization stage then solves $\pi_{\text{RLHF}} = \underset{\pi \in \Pi}{\arg\max}\, V_{r^\star}^\pi$, so by definition, $V_{r^\star}^{\pi_{\text{RLHF}}} = V_{r^\star}^\Pi$.

On the other hand, DPO is trained on preferences induced by $r^\star$. When $\pi^\star \in \Pi$, the preference structure is realizable, and the DPO loss is minimized by $\pi^\star$. Hence, $\pi_{\text{DPO}} = \pi^\star$, which achieves the maximum of $V_{r^\star}^\pi$ over $\Pi$.

### C.2  PROOF OF THEOREM 2

By Taylor's expansion, we have that:

$$\nabla_\theta \mathcal{L}_{\text{DPO}}^{\text{online}}(\pi_\theta)$$

$$= -\beta \underset{y,y' \sim \pi^s}{\mathbb{E}} \left[ \nabla_\theta \log \pi_\theta(y) - \nabla_\theta \log \pi_\theta(y') \right] \cdot \sigma'(\hat{r}_\theta(y) - \hat{r}_\theta(y')) \cdot \left[ (r^\star(y) - r^\star(y')) - (\hat{r}_\theta(y) - \hat{r}_\theta(y')) \right]$$

$$- \beta \underset{y,y' \sim \pi^s}{\mathbb{E}} \left[ \nabla_\theta \log \pi_\theta(y) - \nabla_\theta \log \pi_\theta(y') \right] \cdot \sigma''(\xi_{y,y'}) \cdot \left[ (r^\star(y) - r^\star(y')) - (\hat{r}_\theta(y) - \hat{r}_\theta(y')) \right]^2 ,$$

where $\xi_{y,y'}$ is an intermediate value between $r^\star(y) - r^\star(y')$ and $\hat{r}_\theta(y) - \hat{r}_\theta(y')$.

Therefore, if we have:

- $0 \leqslant r(y) \leqslant R_{\max}, \forall y \in \mathcal{Y}$;
- $|(r^\star(y) - r^\star(y')) - (\hat{r}_\theta(y) - \hat{r}_\theta(y'))| \leqslant \delta, \forall y, y' \in \mathcal{Y}$;
- $\pi^s(y, y') \propto \pi_\theta(y)\pi_\theta(y')/\sigma'(\hat{r}_\theta(y) - \hat{r}_\theta(y'))$, *i.e.*, $\pi^s$ is PILAF sampler,

then the formula can be rewritten as:

$$\mathcal{L}_{\text{DPO}}^{\text{online}}(\pi_\theta) \overset{\nabla}{=} \frac{1}{2\mathsf{sg}\,(Z_\theta)} \underset{y,y' \sim \pi_\theta}{\mathbb{E}} (1 + \epsilon_{y,y'}) \cdot \left[ (r^\star(y) - r^\star(y')) - (\hat{r}_\theta(y) - \hat{r}_\theta(y')) \right]^2 ,$$

where

$$|\epsilon_{y,y'}| = \left| \frac{\sigma''(\xi_{y,y'})}{\sigma'(\hat{r}_\theta(y) - \hat{r}_\theta(y'))} \right| \cdot |(r^\star(y) - r^\star(y')) - (\hat{r}_\theta(y) - \hat{r}_\theta(y'))| \leqslant \frac{\delta}{6\sqrt{3}\sigma'(R_{\max} + \delta)} ,$$

and

$$Z_\theta := \sum_{y,y' \in \mathcal{Y}} \pi_\theta(y)\pi_\theta(y')/\sigma'(\hat{r}_\theta(y) - \hat{r}_\theta(y')) .$$

Note that:

$$\nabla_\theta \left[ \underset{y \sim \pi_\theta}{\mathbb{E}} [r^\star(x, y)] - \beta \mathsf{KL}\,(\pi_\theta \| \pi_{\text{ref}}) \right] \tag{7}$$

$$= \nabla_\theta \underset{y \sim \pi_\theta}{\mathbb{E}} [r^\star(y) - \hat{r}_\theta(y)]$$

$$= \underset{y \sim \pi_\theta}{\mathbb{E}} \nabla_\theta \log \pi_\theta(y)[r^\star(y) - \hat{r}_\theta(y)] \qquad \text{(policy gradient theorem)}$$

$$= \underset{y,y' \sim \pi_\theta}{\mathbb{E}} \nabla_\theta \log \pi_\theta(y)[(r^\star(y) - r^\star(y')) - (\hat{r}_\theta(y) - \hat{r}_\theta(y'))] \qquad \text{(policy gradient theorem)}$$

$$= \frac{1}{2} \underset{y,y' \sim \pi_\theta}{\mathbb{E}} \left[ \nabla_\theta \log \pi_\theta(y) - \nabla_\theta \log \pi_\theta(y') \right] [(r^\star(y) - r^\star(y')) - (\hat{r}_\theta(y) - \hat{r}_\theta(y'))] , \qquad \text{(symmetry)}$$

thus

$$\underset{y \sim \pi_\theta}{\mathbb{E}} [r^\star(x, y)] - \beta \mathsf{KL}\,(\pi_\theta \| \pi_{\text{ref}}) \overset{\nabla}{=} -\frac{1}{4\beta} \underset{y,y' \sim \pi_\theta}{\mathbb{E}} \left[ (r^\star(y) - r^\star(y')) - (\hat{r}_\theta(y) - \hat{r}_\theta(y')) \right]^2 .$$

Therefore we have:

$$\mathcal{L}_{\text{DPO}}^{\text{online}}(\pi_\theta) \overset{\nabla}{=} \frac{2\beta}{\text{sg}(Z_\theta)} \left\{ - \left[ \mathop{\mathbb{E}}_{y \sim \pi_\theta} [r^\star(x, y)] - \beta \text{KL}(\pi_\theta \| \pi_{\text{ref}}) \right] \right.$$

$$\left. + \frac{1}{4\beta} \mathop{\mathbb{E}}_{y, y' \sim \text{sg}(\pi_\theta)} \left[ \epsilon_{y,y'} \cdot \left[ \left( r^\star(y) - r^\star(y') \right) - \left( \hat{r}_\theta(y) - \hat{r}_\theta(y') \right) \right]^2 \right] \right\} \, .$$

## C.3 PROOF OF PROPOSITION 3

Since $r^\star \in \mathcal{F}$, RLHF recovers $r^\star$ exactly and then solves $\pi_{\text{RLHF}} = \text{argmax}_{\pi \in \Pi} V_{r^\star}^\pi$, by definition achieving $V_{r^\star}^{\pi_{\text{RLHF}}} = V_{r^\star}^\Pi$. DPO, instead, minimizes a proxy loss defined over pairwise preferences. Since $\pi_{\text{DPO}} \in \Pi$, we have $V_{r^\star}^{\pi_{\text{DPO}}} \leqslant \max_{\pi \in \Pi} V_{r^\star}^\pi = V_{r^\star}^\Pi = V_{r^\star}^{\pi_{\text{RLHF}}}$, which proves the first claim.

For the strict gap, we consider a multi-armed bandit setting with the action space $\mathcal{Y} = \{a_1, a_2, a_3\}$. Let the ground-truth reward function satisfy:

$$r = r^\star(a_1) = r^\star(a_2) \geqslant r^\star(a_3) = 0 \, .$$

Assume the linear feature mapping $\psi : \mathcal{Y} \to \mathbb{R}^d$ satisfies:

$$\psi(a_1) \neq \psi(a_2) \, , \ \psi(a_3) = \tfrac{1}{2}\psi(a_1) + \tfrac{1}{2}\psi(a_2) \, .$$

Define the log-linear policy class $\Pi = \{\pi_\theta : \theta \in \mathbb{R}^d\}$ by $\pi_\theta(a) \propto \pi_{\text{ref}}(a) \exp(\theta^\top \psi(a))$, where $\pi_{\text{ref}} = \text{Unif}(\mathcal{Y})$. Since $r^\star$ is realizable, RLHF exactly recovers it and solves:

$$\pi_{\text{RLHF}} = \text{argmax}_{\pi_\theta \in \Pi} V_{r^\star}^{\pi_\theta} = \text{argmax}_{\pi_\theta \in \Pi} \sum_{a \in \mathcal{Y}} \pi_\theta(a) r^\star(a) - \beta \, \text{KL}(\pi_\theta \| \pi_{\text{ref}}) \, .$$

For a fixed $r > 0$, as the regularization parameter $\beta \to 0$, the optimal policy under RLHF places vanishing probability on $a_3$: $\pi_{\text{RLHF}}(a_3) \to 0$. In contrast, as $\beta \to \infty$, the regularization dominates and the optimal policy converges to the uniform reference policy: $\pi_{\text{RLHF}} \to \pi_{\text{ref}}$.

Now consider the DPO objective, which relies on pairwise preference probabilities and directly optimizes over the policy class:

$$\mathcal{L}_{\text{DPO}}(\pi_\theta) = - \sum_{a \neq a'} \left[ \sigma(r^\star(a) - r^\star(a')) \log \sigma\left( \beta \, \theta^\top (\psi(a) - \psi(a')) \right) \right]$$

$$= -\tfrac{1}{2} \log \sigma(\beta \Delta^\top \theta) - \tfrac{1}{2} \log \sigma(-\beta \Delta^\top \theta) - \log \sigma(\tfrac{1}{2}\beta \Delta^\top \theta) - \log \sigma(-\tfrac{1}{2}\beta \Delta^\top \theta) \, ,$$

where $\Delta := \psi(a_1) - \psi(a_2)$. This expression is always minimized when $\Delta^\top \theta = 0$, which corresponds to a uniform distribution.

Thus, unlike RLHF, the DPO solution remains fixed at uniform distribution, independent of the reward magnitude $r$ and the regularization parameter $\beta$, and fails to suppress the sub-optimal action $a_3$ even when $\beta$ is sufficiently small.

## C.4 PROOF OF PROPOSITION 5

Since $r^\star \notin \mathcal{F}$, RLHF recovers an approximation $\hat{r} \in \mathcal{F}$ via reward learning. It then computes a policy $\pi_{\text{RLHF}}$ that maximizes $V_{\hat{r}}^\pi$ over $\Pi$. In general, this policy is sub-optimal under $r^\star$ (see Proposition 3), and thus $V_{r^\star}^{\pi_{\text{RLHF}}} \leqslant \max_{\pi \in \Pi} V_{r^\star}^\pi = V_{r^\star}^\Pi$.

DPO directly optimizes a preference-based loss over $\Pi$. Since $\pi^\star \in \Pi$ and DPO is given access to exact preference data consistent with $r^\star$, it can recover $\pi^\star$, and hence $V_{r^\star}^{\pi_{\text{DPO}}} = V_{r^\star}^{\pi^\star} = V_{r^\star}^\Pi$.

For the strict gap, consider a multi-armed bandit setting analogous to Appendix C.3: first, define the action space $\mathcal{Y} = \{a_1, a_2, a_3\}$. Let the ground-truth reward function satisfy:

$$r = r^\star(a_1) = r^\star(a_2) \geqslant r^\star(a_3) = 0 \, .$$

Assume the linear feature mapping $\psi : \mathcal{Y} \to \mathbb{R}^d$ satisfies:

$$\psi(a_1) \neq \psi(a_2) \, , \ \psi(a_3) = \tfrac{1}{2}\psi(a_1) + \tfrac{1}{2}\psi(a_2) \, .$$

The key difference from the earlier construction lies in the choice of model classes. We define: the linear reward class $\mathcal{F} = \{r_\phi : \phi \in \mathbb{R}^d\}$ by $r_\phi(a) := \phi^\top \psi(a)$, and the policy class $\Pi = \Delta(\mathcal{Y})$ with reference policy $\pi_{\mathsf{ref}} = \mathrm{Unif}(\mathcal{Y})$. This setup satisfies Condition 3 because $r^\star \notin \mathcal{F}$: for any $\phi$, the constraint on $\psi$ implies $r_\phi(a_3) = \frac{1}{2}(r_\phi(a_1) + r_\phi(a_2))$ so $r_\phi(a_3) = r$ whenever $r_\phi(a_1) = r_\phi(a_2) = r$, contradicting the ground-truth reward $r^\star(a_3) = 0$.

In RLHF, the reward model is learned by solving the population MLE objective:

$$r_{\mathrm{RLHF}} = \operatorname*{argmax}_{r_\phi \in \mathcal{F}} \sum_{a \neq a'} \left[ \sigma(r^\star(a) - r^\star(a')) \log \sigma(\beta \phi^\top (\psi(a) - \psi(a'))) \right]$$

$$= \operatorname*{argmax}_{r_\phi \in \mathcal{F}} -\tfrac{1}{2} \log \sigma(\beta \Delta^\top \phi) - \tfrac{1}{2} \log \sigma(-\beta \Delta^\top \phi) - \log \sigma(\tfrac{1}{2}\beta \Delta^\top \phi) - \log \sigma(-\tfrac{1}{2}\beta \Delta^\top \phi) \ ,$$

where $\Delta := \psi(a_1) - \psi(a_2)$. This expression is minimized maximized at $\Delta^\top \phi = 0$, which implies $r_\phi(a_1) = r_\phi(a_2)$ and $r_\phi(a_3) = r_\phi(a_1)$, $i.e.$, the learned reward is constant: $r_{\mathrm{RLHF}}(a) = C$ for all $a \in \mathcal{Y}$.

The policy learning stage then solves:

$$\pi_{\mathrm{RLHF}} = \operatorname*{argmax}_{\pi \in \Delta(\mathcal{Y})} \mathop{\mathbb{E}}_{a \sim \pi} [C] - \beta \, \mathsf{KL}\left(\pi \| \pi_{\mathsf{ref}}\right) \ ,$$

whose solution is simply $\pi_{\mathrm{RLHF}} = \pi_{\mathsf{ref}}$, independent of $r$ and $\beta$.

In contrast, DPO directly optimizes the policy using preference comparisons. Since $\Pi = \Delta(\mathcal{Y})$ and the preferences are consistent with the ground-truth reward $r^\star$, DPO can recover the optimal policy $\pi^\star \propto \exp(r^\star/\beta)$, which is not uniform. Therefore, DPO achieves the optimal regularized value $V_\Pi^\star = V_{r^\star}^{\pi^\star}$, while RLHF only returns the uniform policy. This yields a strict gap:

$$V_{r^\star}^{\pi_{\mathrm{RLHF}}} < V_{r^\star}^{\pi_{\mathrm{DPO}}} = V_{r^\star}^\Pi \ .$$

## C.5 Proof of Proposition 6

By definition, the reward learned by RLHF and the surrogate reward learned by DPO are obtained by solving the following population objectives:

$$r_{\mathrm{RLHF}} = \operatorname*{argmax}_{r_\phi \in \mathcal{F}} \mathop{\mathbb{E}}_{y,y' \sim \pi_{\mathsf{ref}}} \left[ p^\star(y > y') \log \sigma(r_\phi(y) - r_\phi(y')) + p^\star(y' > y) \log \sigma(r_\phi(y') - r_\phi(y)) \right] \ ,$$

$$\hat{r}_{\mathrm{DPO}} = \operatorname*{argmax}_{\hat{r}_\theta \in \mathcal{F}_\Pi} \mathop{\mathbb{E}}_{y,y' \sim \pi_{\mathsf{ref}}} \left[ p^\star(y > y') \log \sigma(\hat{r}_\theta(y) - \hat{r}_\theta(y')) + p^\star(y' > y) \log \sigma(\hat{r}_\theta(y') - \hat{r}_\theta(y)) \right] \ ,$$

Under Condition 5, we have $\mathcal{F} = \mathcal{F}_\Pi$, so both objectives are optimized over the same function class. Hence, it follows that: $r_{\mathrm{RLHF}} = \hat{r}_{\mathrm{DPO}}$.

Recalling from Equation (2):

$$\pi_{\mathrm{RLHF}} = \operatorname*{argmax}_{\pi \in \Pi} V_{r_{\mathrm{RLHF}}}^\pi \ , \ \ \pi_{\mathrm{DPO}} = \operatorname*{argmax}_{\pi \in \Pi} V_{\hat{r}_{\mathrm{DPO}}}^\pi \ .$$

and substituting $r_{\mathrm{RLHF}} = \hat{r}_{\mathrm{DPO}}$, we can conclude that

$$\pi_{\mathrm{RLHF}} = \pi_{\mathrm{DPO}} \ \ \text{and hence} \ \ V_{r^\star}^{\pi_{\mathrm{RLHF}}} = V_{r^\star}^{\pi_{\mathrm{DPO}}} \ .$$

## C.6 Proof of Proposition 8

**Construction 1:** $V_{r^\star}^{\pi_{\mathrm{RLHF}}} < V_{r^\star}^{\pi_{\mathrm{DPO}}}$. We first construct an environment satisfying Condition 6 such that $V_{r^\star}^{\pi_{\mathrm{RLHF}}} < V_{r^\star}^{\pi_{\mathrm{DPO}}}$. Consider the same setup as in Appendix C.4, but define the policy class as $\Pi = \Delta(\mathcal{Y}) \backslash \{\pi^\star\}$, where $\pi^\star$ is the optimal policy under $r^\star$. This ensures that $\pi^\star \notin \Pi$, while $\mathcal{F} \subset \mathcal{F}_\Pi$, satisfying Condition 6.

As shown in Appendix C.4, RLHF learns a constant reward model and returns the uniform policy $\pi_{\mathrm{RLHF}} = \pi_{\mathsf{ref}}$, independent of $r$ and $\beta$. In contrast, DPO directly optimizes policy parameters from preference data and can converge to a policy arbitrarily close to $\pi^\star$, which lies on the boundary of $\Pi$. This yields a strict sub-optimality gap:

$$V_{r^\star}^{\pi_{\mathrm{RLHF}}} < V_{r^\star}^{\pi_{\mathrm{DPO}}} \ .$$

**Construction 2:** $V_{r^\star}^{\pi_{\mathrm{RLHF}}} > V_{r^\star}^{\pi_{\mathrm{DPO}}}$**.** Next, we construct an environment satisfying Condition 6 such that $V_{r^\star}^{\pi_{\mathrm{RLHF}}} > V_{r^\star}^{\pi_{\mathrm{DPO}}}$. Consider a multi-armed bandit with action space $\mathcal{Y} = \{a_1, a_2, a_3\}$ and ground-truth reward:

$$r^\star(a_1) = r^\star(a_2) = 1 \ , \ r^\star(a_3) = 0 \ .$$

Let the linear feature mapping $\psi : \mathcal{Y} \to \mathbb{R}^2$ be:

$$\psi(a_1) = \begin{bmatrix} 1 \\ 0 \end{bmatrix}, \ \psi(a_2) = \begin{bmatrix} 0 \\ 1 \end{bmatrix}, \ \psi(a_3) = \begin{bmatrix} 1/2 \\ 1/2 \end{bmatrix} .$$

Define the log-linear policy class $\Pi = \{\pi_\theta : \theta \in \mathbb{R}^2\}$ with

$$\pi_\theta(a) \propto \pi_{\mathsf{ref}}(a) \exp(\theta^\top \psi(a)) \ , \ \pi_{\mathsf{ref}} = \mathrm{Unif}(\mathcal{Y}).$$

The corresponding surrogate reward class is $\mathcal{F}_\Pi = \{\hat{r}_\theta : \hat{r}_\theta(a) = \beta \theta^\top \psi(a), \ \theta \in \mathbb{R}^2\}$. We now define a strictly smaller reward model class $\mathcal{F} = \{\hat{r}_{\theta_R}\}$ where

$$\theta_R = \begin{bmatrix} 1 \\ -1 \end{bmatrix} .$$

We set the regularization parameter to $\beta = 0.1$. Then, $\mathcal{F} \subset \mathcal{F}_\Pi$ and Condition 6 holds.

Under this setup, RLHF learns the fixed reward $\hat{r}_{\theta_R}$ and optimizes:

$$\pi_{\mathrm{RLHF}} = \pi_{\theta_R} \ , \ \text{where} \ \ \pi_{\theta_R}(a) \propto \exp(\theta_R^\top \psi(a)) \ .$$

Concretely:

$$\pi_{\theta_R}(a_1) = \frac{\exp(1)}{Z} \ , \ \pi_{\theta_R}(a_2) = \frac{\exp(-1)}{Z} \ , \ \pi_{\theta_R}(a_3) = \frac{1}{Z} \ , \ Z = \exp(1) + \exp(-1) + 1 \ .$$

The value of this policy under $r^\star$ is:

$$V_{r^\star}^{\pi_{\mathrm{RLHF}}} = \pi_{\theta_R}(a_1) + \pi_{\theta_R}(a_2) - \beta \, \mathrm{KL}(\pi_{\theta_R} \| \pi_{\mathsf{ref}}) \approx 0.729 \ .$$

In contrast, DPO learns the uniform policy $\pi_{\mathrm{DPO}} = \pi_{\mathsf{ref}}$, as shown in Appendix C.3. Its value is:

$$V_{r^\star}^{\pi_{\mathrm{DPO}}} = \frac{2}{3} \ .$$

This results in a strict sub-optimality gap in the opposite direction:

$$V_{r^\star}^{\pi_{\mathrm{RLHF}}} > V_{r^\star}^{\pi_{\mathrm{DPO}}} \ .$$

### C.7 PROOF OF PROPOSITION 9

**Construction 1:** $V_{r^\star}^{\pi_{\mathrm{RLHF}}} > V_{r^\star}^{\pi_{\mathrm{DPO}}}$**.** We construct an environment satisfying Condition 7 such that $V_{r^\star}^{\pi_{\mathrm{RLHF}}} > V_{r^\star}^{\pi_{\mathrm{DPO}}}$. Consider a multi-armed bandit with action space $\mathcal{Y} = \{a_1, a_2, a_3\}$ and ground-truth reward function:

$$r^\star(a_1) = r^\star(a_2) = 1 \ , \ r^\star(a_3) = 0 \ .$$

Let the linear feature mapping $\psi : \mathcal{Y} \to \mathbb{R}^2$ be:

$$\psi(a_1) = \begin{bmatrix} 1 \\ 0 \end{bmatrix}, \ \psi(a_2) = \begin{bmatrix} 0 \\ 1 \end{bmatrix}, \ \psi(a_3) = \begin{bmatrix} 1/2 \\ 1/2 \end{bmatrix} .$$

Define the log-linear policy class $\Pi = \{\pi_\theta : \theta \in \mathbb{R}^2\}$ with:

$$\pi_\theta(a) \propto \pi_{\mathsf{ref}}(a) \exp(\theta^\top \psi(a)) \ , \ \pi_{\mathsf{ref}} = \mathrm{Unif}(\mathcal{Y}) \ .$$

The corresponding surrogate reward class is $\mathcal{F}_\Pi = \{\hat{r}_\theta : \hat{r}_\theta(a) = \beta \theta^\top \psi(a), \ \theta \in \mathbb{R}^2\}$. Now define a strictly larger reward model class:

$$\mathcal{F} = \mathcal{F}_\Pi \cup \{\bar{r}\} \ , \ \text{where} \ \bar{r}(a_1) = \bar{r}(a_2) = 2, \ \bar{r}(a_3) = 0 \ .$$

Then $\mathcal{F}_\Pi \subset \mathcal{F}$, and thus Condition 7 holds.

From Appendix C.3, we know that DPO learns a constant reward model under this feature structure and returns the uniform policy $\pi_{\text{DPO}} = \pi_{\text{ref}}$, independent of $r$ and $\beta$.

RLHF, on the other hand, optimizes the MLE objective over the larger class $\mathcal{F}$ and selects $\bar{r}$, which achieves a higher likelihood than any function in $\mathcal{F}_\Pi$. Then, the learned policy is:

$$\pi_{\text{RLHF}} = \underset{\pi_\theta \in \Pi}{\arg\max}\ V_{\bar{r}}^{\pi_\theta}\ .$$

As $\beta \to 0$, the optimal policy $\pi_{\text{RLHF}}$ places vanishing mass on $a_3$, since $\bar{r}(a_3) = 0$ while $\bar{r}(a_1) = \bar{r}(a_2) = 2$. Hence, $\pi_{\text{RLHF}}(a_3) \to 0$.

This leads to a strictly better policy under $r^\star$ than the uniform policy returned by DPO. Thus:

$$V_{r^\star}^{\pi_{\text{RLHF}}} > V_{r^\star}^{\pi_{\text{DPO}}}\ .$$

**Construction 2: $V_{r^\star}^{\pi_{\text{RLHF}}} < V_{r^\star}^{\pi_{\text{DPO}}}$.** We now construct an environment satisfying Condition 7 such that $V_{r^\star}^{\pi_{\text{RLHF}}} < V_{r^\star}^{\pi_{\text{DPO}}}$. Consider a multi-armed bandit with action space $\mathcal{Y} = \{a_1, a_2, a_3\}$ and ground-truth reward function:

$$r^\star(a_1) = r^\star(a_2) = 1\ ,\ r^\star(a_3) = 0\ .$$

Let the linear feature mapping $\psi : \mathcal{Y} \to \mathbb{R}^2$ be:

$$\psi(a_1) = \begin{bmatrix} 1 \\ 0 \end{bmatrix},\ \psi(a_2) = \begin{bmatrix} 0 \\ 1 \end{bmatrix},\ \psi(a_3) = \begin{bmatrix} 1/2 \\ 1/2 \end{bmatrix}.$$

We define a constrained log-linear policy class:

$$\Pi = \left\{ \pi_\theta : \theta \in \mathbb{R}^2,\ \theta^\top \begin{bmatrix} 1 \\ -1 \end{bmatrix} \geqslant 20 \right\}\ ,\ \pi_\theta(a) \propto \pi_{\text{ref}}(a) \exp(\theta^\top \psi(a))\ ,$$

where $\pi_{\text{ref}} = \text{Unif}(\mathcal{Y})$. The corresponding surrogate reward class is:

$$\mathcal{F}_\Pi = \left\{ \hat{r}_\theta : \hat{r}_\theta(a) = \beta \theta^\top \psi(a)\ ,\ \theta^\top \begin{bmatrix} 1 \\ -1 \end{bmatrix} \geqslant 20 \right\}\ .$$

Now define a strictly larger reward model class:

$$\mathcal{F} = \mathcal{F}_\Pi \cup \{\bar{r}\}\ ,\ \text{where } \bar{r}(a_1) = \bar{r}(a_2) = 2\ ,\ \bar{r}(a_3) = 0\ .$$

We set the regularization parameter to $\beta = 0.1$. Since $\bar{r} \notin \mathcal{F}_\Pi$, we have $\mathcal{F}_\Pi \subset \mathcal{F}$, and thus Condition 7 holds.

Under this setup, RLHF first learns the reward model by optimizing the MLE objective over the larger class $\mathcal{F}$ and selects $\bar{r}$, which achieves strictly higher likelihood than any element in $\mathcal{F}_\Pi$. In the policy learning stage, RLHF computes the policy $\pi_{\text{RLHF}} = \pi_{\theta_{\text{RLHF}}}$ by solving:

$$\pi_{\theta_{\text{RLHF}}} = \underset{\pi_\theta \in \Pi}{\arg\max}\ V_{\bar{r}}^{\pi_\theta} = \underset{\pi_\theta \in \Pi}{\arg\max}\ \left\{ 2\big(\pi_\theta(a_1) + \pi_\theta(a_2)\big) - \beta\, \text{KL}\left(\pi_\theta \| \pi_{\text{ref}}\right) \right\}\ .$$

In contrast, DPO directly optimizes the reward via MLE:

$$\hat{r}_{\text{DPO}} = \underset{\hat{r}_\theta \in \mathcal{F}_\Pi}{\arg\max}\ \underset{y,y' \sim \pi_{\text{ref}}}{\mathbb{E}} \left[ p^\star(y > y') \log \sigma(\hat{r}_\theta(y) - \hat{r}_\theta(y')) + p^\star(y' > y) \log \sigma(\hat{r}_\theta(y') - \hat{r}_\theta(y)) \right]\ ,$$

whose optimal solution corresponds to $\theta$ satisfying $\theta^\top \begin{bmatrix} 1 \\ -1 \end{bmatrix} = 20$. Therefore, the learned policy is

$\pi_{\text{DPO}} = \pi_{\theta_{\text{DPO}}}$ with $\theta_{\text{DPO}}^\top \begin{bmatrix} 1 \\ -1 \end{bmatrix} = 20$.

To compare the values $V_{r^\star}^{\pi_{\text{RLHF}}}$ and $V_{r^\star}^{\pi_{\text{DPO}}}$, we rewrite the value function for any $\pi_\theta$ as:

$$V_{r^\star}^{\pi_\theta} = \pi_\theta(a_1) + \pi_\theta(a_2) - \beta\, \text{KL}\left(\pi_\theta \| \pi_{\text{ref}}\right)$$

$$= \frac{e^{x/2} + e^{-x/2}}{Z(x)} - \beta \left[ \frac{e^{x/2}}{Z(x)} \log\left(\frac{e^{x/2}}{Z(x)}\right) + \frac{e^{-x/2}}{Z(x)} \log\left(\frac{e^{-x/2}}{Z(x)}\right) + \frac{1}{Z(x)} \log\left(\frac{1}{Z(x)}\right) \right] + \text{(constant)}\ ,$$

where $x := \theta^\top \begin{bmatrix} 1 \\ -1 \end{bmatrix}$ and $Z(x) := e^{x/2} + e^{-x/2} + 1$.

It can be verified that $V_{r^\star}^{\pi_\theta}$ is strictly decreasing in $x$ for $x \geqslant 20$. Since RLHF learns $x_{\text{RLHF}} \approx 40$ and DPO learns $x_{\text{DPO}} = 20$, we conclude that

$$V_{r^\star}^{\pi_{\text{RLHF}}} < V_{r^\star}^{\pi_{\text{DPO}}}\ ,$$

demonstrating that a more expressive reward model class may lead RLHF to overfitting in the presence of a constrained policy class, resulting in inferior performance compared to DPO.

## C.8   NUMERICAL PROOF OF PROPOSITION 4

Since the exact solution for online DPO is hard to compute, we didn't find elegant proofs for these two propositions. They are examined correct numerically.

By Proposition 3, we have $V_{r^\star}^{\pi_{\mathrm{RLHF}}} = V_{r^\star}^{\Pi} = \max_{\pi \in \Pi} V_{r^\star}^{\pi} \geqslant V_{r^\star}^{\pi_{\mathrm{DPO}}^{\mathrm{online}}}$. Now we construct an environment under Condition 2, such that online DPO cannot outperform DPO, even with PILAF sampler. Consider a multi-armed bandit with action space $\mathcal{Y} = \{a_1, a_2, a_3\}$ and ground-truth reward:

$$r^\star(a_1) = 12 \ , \ r^\star(a_2) = 12 \ , \ r^\star(a_3) = 0 \ .$$

Let the linear feature mapping $\psi : \mathcal{Y} \to \mathbb{R}^d$ satisfies:

$$\psi(a_1) \neq \psi(a_2) \ , \ \psi(a_3) = \frac{1}{2}\psi(a_1) + \frac{1}{2}\phi(a_2) \ .$$

Taking $\beta = 1$, let $x(\theta)$ denote $\log \frac{\pi_\theta(a_1)}{\pi_{\mathrm{ref}}(a_1)} - \log \frac{\pi_\theta(a_2)}{\pi_{\mathrm{ref}}(a_2)}$. Define the bounded log-linear policy class $\Pi = \{\pi_\theta : \theta \in \mathbb{R}^d \ , \ |x(\theta)| \leqslant 4\}$ with

$$\pi_\theta(a) \propto \pi_{\mathrm{ref}}(a) \exp(\theta^\top \psi(a)) \ , \ \ \pi_{\mathrm{ref}} = \mathrm{Unif}(\mathcal{Y}) \ .$$

Note that we can use $x(\theta)$ to represent the whole distribution thanks to the feature mapping. Now we numerically compute the gradients of the loss functions of RL, DPO, and online DPO with PILAF sampler, in the interval $x(\theta) \in [-4, 4]$. And the curves along with respective solutions are shown in the left panel of Figure 5, where the gradient values are rescaled for clarity of presentation. We find that both DPO and online DPO will converge to the same sub-optimal solution, while RL can obtain an optimal solution.

## C.9   NUMERICAL PROOF OF PROPOSITION 7

By Proposition 6, we have $\pi_{\mathrm{RLHF}} = \pi_{\mathrm{DPO}}$. Now we only need to construct an environment under Condition 5, such that online DPO can outperform offline DPO. We can borrow the whole setting in Appendix C.8, while resetting the ground-truth reward as:

$$r^\star(a_1) = 24 \ , \ r^\star(a_2) = 12 \ , \ r^\star(a_3) = 0 \ .$$

Now we numerically compute the gradients of the loss functions of DPO and online DPO with a pure online sampler, in the interval $x(\theta) \in [-4, 4]$. And the curves along with respective solutions are shown in the right panel of Figure 5, where the gradient values are rescaled for clarity of presentation. We find that online DPO can help obtain better solution than DPO, which indicates that under Condition 5, online DPO can produce a solution $\pi_{\mathrm{DPO}}^{\mathrm{online}}$, such that $V_{r^\star}^{\pi_{\mathrm{RLHF}}} < V_{r^\star}^{\pi_{\mathrm{DPO}}^{\mathrm{online}}}$.

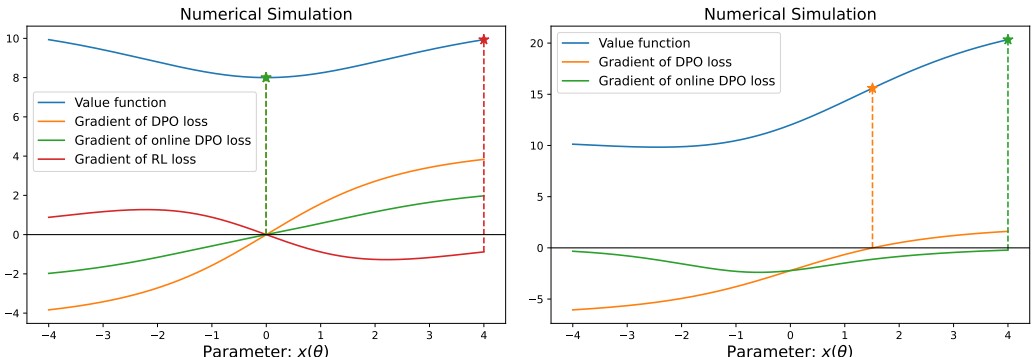

Figure 5: Numerically Computed Curves of Gradient Functions and Value Functions.

## C.10   FORMAL STATEMENT OF THEOREMS 10 AND 11 AND PROOFS

### C.10.1   PRELIMINARIES OF SINGLE-TOKEN PREDICTION

Before proceeding, we first prepare some ingredients for the single-token prediction task.

**Basic setting.** Recall that to train a (surrogate) reward model, people first collect a dataset $\mathcal{D}^\dagger = \{y_1^{(i)}, y_2^{(i)}\}_{i=1}^n$, and then ask human annotators to label these pairs to get a human preference dataset $\mathcal{D} = \{y_w^{(i)}, y_l^{(i)}\}_{i=1}^n$. Following BT model, $y_1$ is preferred over $y_2$, (*i.e.* $y_w = y_1$ and $y_l = y_2$), w.p. $\sigma(r^\star(y_1) - r^\star(y_2))$, where $r^\star(y) = (\theta^\star)^\top \psi(y)$, $\theta^\star \in \mathbb{R}_+$ is the ground-truth reward vector, $\psi(y)$ is the feature vector satisfying $\|\psi(y)\|_2 \leqslant L$, and $L \in \mathbb{R}_+$. The MLE estimator is defined as:

$$\hat{\theta}_{\mathrm{MLE}} \in \underset{\theta \in \Theta_B}{\arg\min} -\frac{1}{n} \sum_{i=1}^n \log \sigma(\theta^\top(\psi(y_w^{(i)}) - \psi(y_l^{(i)}))) , \tag{8}$$

where $\Theta_B = \{\theta \in \mathbb{R}_d : \|\theta\|_2 \leqslant B\}$, $B \in \mathbb{R}_+$. And we assume $\theta^\star \in \Theta_B$. The empirical performance measure is the data-induced semi-norm (see, e.g., (Zhu et al., 2023)), defined as:

**Definition 2** (Data-induced semi-norm). *The empirical error of an estimate $\hat{\theta}$ is defined as:*

$$\|\hat{\theta} - \theta^\star\|_{\Sigma_{\mathcal{D}}}^2 := \frac{1}{n} \sum_{i=1}^n \left[ (r_{\hat{\theta}}(y_w^{(i)}) - r_{\hat{\theta}}(y_l^{(i)})) - (r^\star(y_w^{(i)}) - r^\star(y_l^{(i)})) \right]^2 ,$$

*where $\Sigma_{\mathcal{D}}$ is the Gram matrix:*

$$\Sigma_{\mathcal{D}} := \frac{1}{n} \sum_{i=1}^n (\psi(y_w^{(i)}) - \psi(y_i^{(i)}))(\psi(y_w^{(i)}) - \psi(y_i^{(i)}))^\top .$$

And we assume $\Sigma_{\mathcal{D}}$ to be non-singular.

Note that the lemmas below only work for the single-token scenario, and we will adopt them in the dual-token prediction task later. The results quoted below from (Yao et al., 2025) follow directly from a long line of work on compressed sensing and sparse recovery based on restricted isometry (or restricted eigenvalue) properties (Candes et al., 2006), recast for the preference learning setting.

**Lemma 1** (Theorem 1.a of Shah et al. (2015)). *For a sample size $n \geqslant c_1 tr(\Sigma_{\mathcal{D}}^{-1})$, any estimator $\hat{\theta}$ based on $n$ samples has a lower bound as:*

$$\sup_{\theta^\star \in \Theta_B} \mathbb{E}\left[ \|\hat{\theta} - \theta^\star\|_{\Sigma_{\mathcal{D}}}^2 \right] = \Omega\left(\frac{d}{n}\right) .$$

**Remark 6.** Here $c_1$ is a constant independent of data. This lemma is to establish an information-theoretical lower bound for single-token reward learning.

**Lemma 2** (Lemma 3.1 of Zhu et al. (2023); see also Shah et al. (2015)). *W.p. at least $1 - \delta$, the estimation error of the MLE estimator $\hat{\theta}_{\mathrm{MLE}}$ has an upper bound:*

$$\|\hat{\theta}_{\mathrm{MLE}} - \theta^\star\|_{\Sigma_{\mathcal{D}}}^2 = \mathcal{O}\left(\frac{d + \log(1/\delta)}{n}\right) .$$

**Definition 3** ($\ell_1$-regularized estimator).

$$\hat{\theta}_{\ell_1} \in \underset{\theta \in \Theta_B}{\arg\min} \mathcal{L}_{\mathrm{MLE}}(\theta) + \gamma\|\theta\|_1 .$$

**Lemma 3** (Theorem 3.3 of Yao et al. (2025)). *Consider $\|\theta^\star\|_0 = k$, then w.p. at least $1 - \delta$, the $\ell_1$-regularized estimator $\hat{\theta}_{\ell_1}$ with an appropriate $\gamma = \Theta\left(\sqrt{\frac{\log(d) + \log(1/\delta)}{n}}\right)$ has an upper bound:*

$$\|\hat{\theta}_{\ell_1} - \theta^\star\|_{\Sigma_{\mathcal{D}}}^2 = \mathcal{O}\left(\sqrt{\frac{k\log(d) + k\log(1/\delta)}{n}}\right) .$$

**Definition 4** (Relative $\ell_1$-regularized estimator). *Given $\tau \in \Theta_B$, the relative $\ell_1$-regularized estimator is defined as:*

$$\hat{\theta}_{\mathrm{rel}\ell_1} \in \underset{\theta \in \Theta_B}{\arg\min} \mathcal{L}_{\mathrm{MLE}}(\theta) + \gamma\|\theta - \tau\|_1 .$$

**Lemma 4** (Generalized version of Lemma 3). *Consider $\tau \in \Theta_B$, $\|\theta^\star - \tau\|_0 = k$, then w.p. at least $1 - \delta$, the relative $\ell_1$-regularized estimator $\hat{\theta}_{\mathrm{rel}\ell_1}$ with an appropriate $\gamma = \Theta\left(\frac{\log(d) + \log(1/\delta)}{n}\right)$ has an upper bound:*

$$\|\hat{\theta}_{\mathrm{rel}\ell_1} - \theta^\star\|_{\Sigma_{\mathcal{D}}}^2 = \mathcal{O}\left(\sqrt{\frac{k\log(d) + k\log(1/\delta)}{n}}\right) .$$

Proof of this lemma is given in Appendix C.10.4.

### C.10.2 FORMAL STATEMENT OF THEOREM 10

**Assumption 12** (Task configuration). *Recall that in DTSP task, we have $r^\star(a,b) = \beta \mathbf{r}_{\text{sparse}}^\top \psi(a) + \beta e_1^\top \psi(a,b)$, where $a, b \in \mathcal{V}$, $\psi(a), \psi(a,b), \mathbf{r}_{\text{sparse}} \in \mathbb{R}_d$, and $\|\mathbf{r}_{\text{sparse}}\|_0 = k$, $k \ll d$. We further assume $B, L \in \mathbb{R}_+$, $\Theta_B := \{\theta \in \mathbb{R}_d : \|\theta\|_2 \leqslant B\}$, $\mathbf{r}_{\text{dense}}, \mathbf{r}_{\text{sparse}}, e_1 + \mathbf{r}_{\text{dense}} + \mathbf{r}_{\text{sparse}} \in \Theta_B$, $\|\psi(a)\|_2 \leqslant L$, and $\psi(a,b) = \psi(b) + (\mathbf{r}_{\text{dense}}^\top \psi(a))e_1$.*

**Assumption 13** (Preference data collection). *For DTSP task, we first collect a single-token dataset $\mathcal{D}^\dagger = \{a_1^{(i)}, a_2^{(i)}\}_{i=1}^n$, and then duplicate it as $\mathcal{D}^\ddagger = \{a_1^{(i)} a_1^{(i)}, a_2^{(i)} a_2^{(i)}\}_{i=1}^n$, and ask human annotators to label these pairs. Now we have collected a dual-token preference dataset $\mathcal{D} = \{y_w^{(i)}, y_l^{(i)}\}_{i=1}^n$, where $y_w^{(i)} = a_1^{(i)} a_1^{(i)}$ and $y_l^{(i)} = a_2^{(i)} a_2^{(i)}$ w.p. $\sigma(r^\star(a_1^{(i)}, a_1^{(i)}) - r^\star(a_2^{(i)}, a_2^{(i)}))$. And we further assume that the Gram matrix $\Sigma_{\mathcal{D}} := \frac{1}{n} \sum_{i=1}^n (\psi(a_w^{(i)}) - \psi(a_l^{(i)}))(\psi(a_w^{(i)}) - \psi(a_l^{(i)}))^\top$ is non-singular, $tr(\Sigma_{\mathcal{D}}^{-1}) = \mathcal{O}(d)$, and $n \geqslant c_1 tr(\Sigma_{\mathcal{D}}^{-1})$, where $c_1$ is the constant in Lemma 1.*

**Theorem 14** (Formal separation theorem). *Under token-level linear parameterization and Assumptions 12 and 13, there exists an environment for DTSP task, s.t. by estimating from a preference dataset $\mathcal{D}$ with $n$ samples under $\theta_1 = e_1$ constraint, the estimation error of the reward model $\hat{\theta}_r$ can be reduced to $\tilde{\mathcal{O}}(\sqrt{k \log d / n})$ using a (computationally efficient) $\ell_1$-regularized estimator:*

$$\hat{\theta}_{r, \text{rel}\ell_1} \in \operatorname*{argmin}_{\theta_0 + e_1 + \mathbf{r}_{\text{dense}} \in \Theta_B, \theta_1 = e_1} -\frac{1}{n} \sum_{i=1}^n \log \sigma(r_\theta(y_w^{(i)}) - r_\theta(y_l^{(i)})) + \gamma \|\theta_0\|_1 \ ,$$

*i.e., w.p. $1 - \delta$,*

$$\frac{1}{n} \sum_{i=1}^n \left[ (r^\star(y_w^{(i)}) - r^\star(y_l^{(i)})) - (r_{\hat{\theta}_{r,\text{rel}\ell_1}}(y_w^{(i)}) - r_{\hat{\theta}_{r,\text{rel}\ell_1}}(y_l^{(i)})) \right]^2 = \mathcal{O}\left( \sqrt{\frac{k \log(d) + k \log(1/\delta)}{n}} \right) \ ,$$

*while the estimation error of any estimator for the DPO model $\hat{\theta}_p$ is lower bounded by $\Omega(d/n)$:*

$$\frac{1}{n} \sum_{i=1}^n \left[ (r^\star(y_w^{(i)}) - r^\star(y_l^{(i)})) - (r_{\hat{\theta}_p}(y_w^{(i)}) - r_{\hat{\theta}_p}(y_l^{(i)})) \right]^2 = \Omega\left( \frac{d}{n} \right) \ .$$

### C.10.3 PROOF OF THEOREM 14

Let $\pi_{\text{ref}}(\cdot|a)$ be identical for all $a$, then we have

$$\log \mathbb{E}_{\omega \sim \pi_{\text{ref}}(\cdot|a)} \exp(\psi(a,b)_1) = \mathbf{r}_{\text{dense}}^\top \psi(a) + C_5 \ ,$$

for $\forall a \in \mathcal{V}$, where $C_5 \in \mathbb{R}$ is an offset.

Recall that:

$$(\theta_{r,0}^\star)^\top \psi(a) = \mathbf{r}_{\text{sparse}}^\top \psi(a) + C_3 \ ,$$

$$(\theta_{p,0}^\star)^\top \psi(a) = \log \mathbb{E}_{\omega \sim \pi_{\text{ref}}(\cdot|a)} \exp(r^\star(a,b)/\beta) + C_4 = \mathbf{r}_{\text{sparse}}^\top \psi(a) + \log \mathbb{E}_{\omega \sim \pi_{\text{ref}}(\cdot|a)} \exp(\psi(a,b)_1) + C_4 \ ,$$

we thus have $\theta_{r,0}^\star = \mathbf{r}_{\text{sparse}}$ and $\theta_{p,0}^\star = \mathbf{r}_{\text{sparse}} + \mathbf{r}_{\text{dense}}$, due to the non-singularity of the Gram matrix.

We can have a $\ell_1$-regularized estimator for the reward model:

$$\hat{\theta}_{r, \text{rel}\ell_1} \in \operatorname*{argmin}_{\theta_0 + \tau_1 \in \Theta_B, \theta_1 = e_1} -\frac{1}{n} \sum_{i=1}^n \log \sigma(r_\theta(a_w^{(i)} a_w^{(i)}) - r_\theta(a_l^{(i)} a_l^{(i)})) + \gamma \|\theta_0\|_1 \ ,$$

$$\implies \hat{\theta}_{r, \text{rel}\ell_1, 0} \in \operatorname*{argmin}_{\theta_0 + \tau_1 \in \Theta_B} -\frac{1}{n} \sum_{i=1}^n \log \sigma(\beta(\theta_0 + \tau_1)^\top (\psi(a_w^{(i)}) - \psi(a_l^{(i)})) + \gamma \|\theta_0 + \tau_1 - \tau_1\|_1 \ ,$$

where $\tau_1 := e_1 + \mathbf{r}_{\text{dense}}$. Then Lemma 4 implies there exists appropriate $\gamma$, such that w.p. $1 - \delta$,

$$\frac{1}{n} \sum_{i=1}^n \left[ (\hat{\theta}_{r,\text{rel}\ell_1,0} - \mathbf{r}_{\text{sparse}})^\top (\psi(a_w^{(i)}) - \psi(a_l^{(i)})) \right]^2 = \mathcal{O}\left( \sqrt{\frac{k \log(d) + k \log(1/\delta)}{n}} \right) \ ,$$

and thus w.p. $1 - \delta$,

$$\frac{1}{n} \sum_{i=1}^{n} \left[ (r^{\star}(y_w^{(i)}) - r^{\star}(y_l^{(i)})) - (r_{\hat{\theta}_{r,\mathrm{rel}\ell_1}}(y_w^{(i)}) - r_{\hat{\theta}_{r,\mathrm{rel}\ell_1}}(y_l^{(i)})) \right]^2$$

$$= \frac{\beta^2}{n} \sum_{i=1}^{n} \left[ (\mathbf{r}_{\mathrm{sparse}} + \mathbf{r}_{\mathrm{dense}} + e_1)^{\top} (\psi(a_w^{(i)}) - \psi(a_l^{(i)})) - (\hat{\theta}_{r,\mathrm{rel}\ell_1,0} + \mathbf{r}_{\mathrm{dense}} + e_1)^{\top} (\psi(a_w^{(i)}) - \psi(a_l^{(i)})) \right]^2$$

$$= \frac{\beta^2}{n} \sum_{i=1}^{n} \left[ (\mathbf{r}_{\mathrm{sparse}} - \hat{\theta}_{r,\mathrm{rel}\ell_1,0})^{\top} (\psi(a_w^{(i)}) - \psi(a_l^{(i)})) \right]^2$$

$$= \mathcal{O} \left( \sqrt{\frac{k \log(d) + k \log(1/\delta)}{n}} \right) .$$

Note that

$$\log \sigma(\hat{r}_{\theta_p}(a_w^{(i)} a_w^{(i)}) - \hat{r}_{\theta_p}(a_l^{(i)} a_l^{(i)})) = \log \sigma(\beta(\theta_{p,0} + e_1)^{\top} (\psi(a_w^{(i)}) - \psi(a_l^{(i)}))) ,$$

then Lemma 1 implies that for any estimator $\hat{\theta}_p$, we have

$$\sup_{e_1 + \mathbf{r}_{\mathrm{dense}} + \mathbf{r}_{\mathrm{sparse}} \in \Theta_B} \frac{1}{n} \sum_{i=1}^{n} \left[ (\hat{\theta}_{p,0} + e_1 - \mathbf{r}_{\mathrm{sparse}} - \mathbf{r}_{\mathrm{dense}} - e_1)^{\top} (\psi(a_w^{(i)}) - \psi(a_l^{(i)})) \right]^2 = \Omega \left( \frac{d}{n} \right) .$$

Now observe the data-induced semi-norm of surrogate reward learning:

$$\frac{1}{n} \sum_{i=1}^{n} \left[ (r^{\star}(y_w^{(i)}) - r^{\star}(y_l^{(i)})) - (\hat{r}_{\hat{\theta}_p}(y_w^{(i)}) - \hat{r}_{\hat{\theta}_p}(y_l^{(i)})) \right]^2$$

$$= \frac{\beta^2}{n} \sum_{i=1}^{n} \left[ (\mathbf{r}_{\mathrm{sparse}} + \mathbf{r}_{\mathrm{dense}} + e_1)^{\top} (\psi(a_w^{(i)}) - \psi(a_l^{(i)})) - (\hat{\theta}_p + e_1)^{\top} (\psi(a_w^{(i)}) - \psi(a_l^{(i)})) \right]^2$$

$$= \frac{\beta^2}{n} \sum_{i=1}^{n} \left[ (\hat{\theta}_{p,0} + e_1 - \mathbf{r}_{\mathrm{sparse}} - \mathbf{r}_{\mathrm{dense}} - e_1)^{\top} (\psi(a_w^{(i)}) - \psi(a_l^{(i)})) \right]^2 .$$

And thus there exists an environment for DTSP, s.t.

$$\frac{1}{n} \sum_{i=1}^{n} \left[ (r^{\star}(y_w^{(i)}) - r^{\star}(y_l^{(i)})) - (\hat{r}_{\hat{\theta}_p}(y_w^{(i)}) - \hat{r}_{\hat{\theta}_p}(y_l^{(i)})) \right]^2 = \Omega \left( \frac{d}{n} \right) .$$

### C.10.4 Proof of Lemma 4

**Lemma 5** (Lemma D.4 of Yao et al. (2025))**.**

$$\mathcal{L}_{\mathrm{MLE}}(\theta^{\star} + \theta') - \mathcal{L}_{\mathrm{MLE}}(\theta^{\star}) - \nabla \mathcal{L}_{\mathrm{MLE}}(\theta^{\star})^{\top} \theta' \geqslant \Theta(\|\theta'\|_{\Sigma_{\mathcal{D}}}^2) ,$$

*for* $\forall \theta' \in \mathbb{R}^d$ *s.t.* $\theta' + \theta^{\star} \in \Theta_B$.

We take $\gamma = \Theta \left( \sqrt{\frac{\log(d) + \log(1/\delta)}{n}} \right)$, where the specific value of $\gamma$ is determined in Theorem 3.3 of Yao et al. (2025). By the definition of the relative $\ell_1$-regularized estimator, we have:

$$\mathcal{L}_{\mathrm{MLE}}(\hat{\theta}_{\mathrm{rel}\ell_1}) + \gamma \|\hat{\theta}_{\mathrm{rel}\ell_1} - \tau\|_1 \leqslant \mathcal{L}_{\mathrm{MLE}}(\theta^{\star}) + \gamma \|\theta^{\star} - \tau\|_1$$

$$\iff \gamma \|\theta^{\star} - \tau\|_1 - \gamma \|\hat{\theta}_{\mathrm{rel}\ell_1} - \tau\|_1 \geqslant \mathcal{L}_{\mathrm{MLE}}(\hat{\theta}_{\mathrm{rel}\ell_1}) - \mathcal{L}_{\mathrm{MLE}}(\theta^{\star}) .$$

By Lemma 5, we have:

$$\mathcal{L}_{\mathrm{MLE}}(\hat{\theta}_{\mathrm{rel}\ell_1}) - \mathcal{L}_{\mathrm{MLE}}(\theta^{\star}) - \nabla \mathcal{L}_{\mathrm{MLE}}(\theta^{\star})^{\top} (\hat{\theta}_{\mathrm{rel}\ell_1} - \theta^{\star}) \geqslant \Theta(\|\hat{\theta}_{\mathrm{rel}\ell_1} - \theta^{\star}\|_{\Sigma_{\mathcal{D}}}^2) .$$

Thus

$$\Theta(\|\hat{\theta}_{\mathrm{rel}\ell_1} - \theta^{\star}\|_{\Sigma_{\mathcal{D}}}^2) \leqslant \gamma \|\theta^{\star} - \tau\|_1 - \gamma \|\hat{\theta}_{\mathrm{rel}\ell_1} - \tau\|_1 - \nabla \mathcal{L}_{\mathrm{MLE}}(\theta^{\star})^{\top} \left[ (\hat{\theta}_{\mathrm{rel}\ell_1} - \tau) - (\theta^{\star} - \tau) \right]$$

$$\leqslant \gamma \|\theta^\star - \tau\|_1 - \gamma \|\hat{\theta}_{\mathrm{rel}\ell_1} - \tau\|_1 + \|\nabla\mathcal{L}_{\mathrm{MLE}}(\theta^\star)\|_\infty \|\hat{\theta}_{\mathrm{rel}\ell_1} - \tau\|_1 + + \|\nabla\mathcal{L}_{\mathrm{MLE}}(\theta^\star)\|_\infty \|(\theta^\star - \tau)\|_1 \,,$$

where the second inequality is by Hölder's inequality. Next, we upper bound $\|\nabla\mathcal{L}_{\mathrm{MLE}}(\theta^\star)\|_\infty$. As shown in Appendix D.3 of Yao et al. (2025), w.p. $1 - \delta$, we have $\|\nabla\mathcal{L}_{\mathrm{MLE}}(\theta^\star)\|_\infty \leqslant \gamma$. Thus, w.p. $1 - \delta$, we have:

$$\Theta(\|\hat{\theta}_{\mathrm{rel}\ell_1} - \theta^\star\|_{\Sigma_{\mathcal{D}}}^2) \leqslant (\|\nabla\mathcal{L}_{\mathrm{MLE}}(\theta^\star)\|_\infty + \gamma) \|\theta^\star - \tau\|_1 + (\|\nabla\mathcal{L}_{\mathrm{MLE}}(\theta^\star)\|_\infty - \gamma) \|\hat{\theta}_{\mathrm{rel}\ell_1} - \tau\|_1$$

$$\leqslant 2\gamma \|\theta^\star - \tau\|_1 \,,$$

$$\implies \|\hat{\theta}_{\mathrm{rel}\ell_1} - \theta^\star\|_{\Sigma_{\mathcal{D}}}^2 = \mathcal{O}(\gamma \|\theta^\star - \tau\|_1) \,.$$

Note that $\theta^\star, \tau \in \Theta_B$, thus $\|\theta^\star - \tau\|_2 = \mathcal{O}(1)$. Then by Cauchy-Schwartz inequality and the fact that $\|\theta^\star - \tau\|_0 = k$, we have $\|\theta^\star - \tau\|_1 = \mathcal{O}(\sqrt{k})$, and finally:

$$\|\hat{\theta}_{\mathrm{rel}\ell_1} - \theta^\star\|_{\Sigma_{\mathcal{D}}}^2 = \mathcal{O}\left(\sqrt{\frac{k\log(d) + k\log(1/\delta)}{n}}\right) \,.$$

### C.10.5   FORMAL STATEMENT OF THEOREM 11 AND PROOF

**Lemma 6** (Lemma J.5 of Nika et al. (2024))**.** *If the features $\psi(a)$ are sampled from a $0$-mean distribution and span $\mathbb{R}^d$, then $\log\sum_a \exp(\theta^\top \psi(a))$ is $\kappa$-strongly convex w.r.t. $\theta \in \Theta_B$, where $\kappa$ is an $\mathcal{O}(1)$ constant determined by $\beta, B, L$ and $|\mathcal{V}|$.*

**Theorem 15** (Formal sub-optimality separation theorem)**.** *Under the same setting as Theorem 14, there exists an environment for DTSP task, s.t. the sub-optimality of the RLHF policy model $\pi_{\mathrm{RLHF}} = \operatorname*{argmax}_{\pi \in \Pi} V^\pi_{r_{\hat{\theta}_r}}$ can be reduced to $\mathcal{O}\left(\sqrt[4]{\frac{k\log d + k\log(1/\delta)}{n}} \cdot \left\|\Sigma_{\mathcal{D}}^{-1/2}\right\|_2\right)$, i.e. w.p. $1 - \delta$,*

$$V^{\pi^\star}_{r^\star} - V^{\pi_{\mathrm{RLHF}}}_{r^\star} = \mathcal{O}\left(\sqrt[4]{\frac{k\log d + k\log(1/\delta)}{n}} \cdot \left\|\Sigma_{\mathcal{D}}^{-1/2}\right\|_2\right) \,,$$

*while the sub-optimality of the DPO policy model $\pi_{\mathrm{DPO}} = \pi_{\hat{\theta}_p}$ is lower bounded:*

$$V^{\pi^\star}_{r^\star} - V^{\pi_{\mathrm{DPO}}}_{r^\star} = \Omega\left(\frac{d}{n} \cdot \frac{1}{\|\Sigma_{\mathcal{D}}\|_2}\right) \,.$$

*Proof.* The proof follows the ideas of Theorem 3.2 of Zhu et al. (2023) and Theorem 4.2 of Nika et al. (2024), with appropriate adaptations to our setting.

$$V^{\pi^\star}_{r^\star} - V^{\pi_{\mathrm{RLHF}}}_{r^\star} \leqslant \mathop{\mathbb{E}}_{\substack{a_1 \sim \pi^\star, b_1 \sim \pi^\star(\cdot|a_1), \\ a_2 \sim \pi_{\mathrm{RLHF}}, b_2 \sim \pi_{\mathrm{RLHF}}(\cdot|a_2)}} \left[ (r^\star(a_1, b_1) - r^\star(a_2, b_2)) - \left(\beta\log\frac{\pi_{\mathrm{RLHF}}(a_1, b_1)}{\pi_{\mathrm{ref}}(a_1, b_1)} - \beta\log\frac{\pi_{\mathrm{RLHF}}(a_2, b_2)}{\pi_{\mathrm{ref}}(a_2, b_2)}\right) \right]$$

$$= \mathop{\mathbb{E}}_{\substack{a_1 \sim \pi^\star, b_1 \sim \pi^\star(\cdot|a_1), \\ a_2 \sim \pi_{\mathrm{RLHF}}, b_2 \sim \pi_{\mathrm{RLHF}}(\cdot|a_2)}} \left[ (r^\star(a_1, b_1) - r^\star(a_2, b_2)) - \left(r_{\hat{\theta}_r}(a_1, b_1) - r_{\hat{\theta}_r}(a_2, b_2)\right) \right]$$

$$= \mathop{\mathbb{E}}_{\substack{a_1 \sim \pi^\star, \\ a_2 \sim \pi_{\mathrm{RLHF}}}} \left[ \beta(\mathbf{r}_{\mathrm{sparse}} - \hat{\theta}_{r,0})^\top (\psi(a_1) - \psi(a_2)) \right]$$

$$= \beta(\mathbf{r}_{\mathrm{sparse}} - \hat{\theta}_{r,0})^\top \mathop{\mathbb{E}}_{\substack{a_1 \sim \pi^\star, \\ a_2 \sim \pi_{\mathrm{RLHF}}}} (\psi(a_1) - \psi(a_2))$$

$$\leqslant \beta\|\Sigma_{\mathcal{D}}^{1/2}(\mathbf{r}_{\mathrm{sparse}} - \hat{\theta}_{r,0})\|_2 \|\Sigma_{\mathcal{D}}^{-1/2} \mathop{\mathbb{E}}_{\substack{a_1 \sim \pi^\star, \\ a_2 \sim \pi_{\mathrm{RLHF}}}} (\psi(a_1) - \psi(a_2))\|_2$$

$$= \beta\|\mathbf{r}_{\mathrm{sparse}} - \hat{\theta}_{r,0}\|_{\Sigma_{\mathcal{D}}} \cdot \mathcal{O}\left(\left\|\Sigma_{\mathcal{D}}^{-1/2}\right\|_2\right)$$

$$= \mathcal{O}\left(\sqrt[4]{\frac{k\log d + k\log(1/\delta)}{n}} \cdot \left\|\Sigma_{\mathcal{D}}^{-1/2}\right\|_2\right) \,.$$

The first inequality comes from performance difference lemma (see Appendix C.11); the second equality comes from the observation that all $r_{\theta_r}$ with $\theta_{r,1} = e_1$ can be fitted by the log-linear policy

model; the third and fourth equalities come from simple calculations under our setting; the fifth inequality comes from Cauchy-Schwarz inequality; the sixth equality comes from the fact that $\psi(a)$ is bounded; and the last equation comes from Theorem 14.

Since the optimal policy satisfies $\pi^\star(a, b) = \pi_{\text{ref}}(a, b) \exp(r(a, b)/\beta)/Z$, we have:

$$V_{r^\star}^{\pi^\star} = \mathbb{E}_{a \sim \pi^\star, b \sim \pi^\star(\cdot|a)} \left[ r^\star(a, b) - \beta \log \frac{\pi^\star(a, b)}{\pi_{\text{ref}}(a, b)} \right]$$

$$= \beta \log Z$$

$$= r^\star(a', b') - \beta \log \frac{\pi^\star(a', b')}{\pi_{\text{ref}}(a', b')} , \ \forall a', b' \in \mathcal{V} .$$

Then we have:

$$V_{r^\star}^{\pi^\star} - V_{r^\star}^{\pi_{\text{DPO}}} = \mathbb{E}_{a \sim \pi_{\text{DPO}}, b \sim \pi_{\text{DPO}}(\cdot|a)} \left[ \beta \log \frac{\pi_{\text{DPO}}(a, b)}{\pi_{\text{ref}}(a, b)} - r^\star(a, b) + V_{r^\star}^{\pi^\star} \right]$$

$$= \mathbb{E}_{a \sim \pi_{\text{DPO}}, b \sim \pi_{\text{DPO}}(\cdot|a)} [\beta \log \pi_{\text{DPO}}(a, b) - \beta \log \pi^\star(a, b)]$$

$$= \beta \mathbb{E}_{a \sim \pi_{\text{DPO}}} \left[ \left( \hat{\theta}_{p,0} - \mathbf{r}_{\text{sparse}} - \mathbf{r}_{\text{dense}} \right)^\top (\psi(a) - v) \right] + \beta \log \frac{\mathbb{E}_{a \sim \pi_{\text{ref}}} \exp \left( (\mathbf{r}_{\text{sparse}} + \mathbf{r}_{\text{dense}})^\top (\psi(a) - v) \right)}{\mathbb{E}_{a \sim \pi_{\text{ref}}} \exp \left( (\hat{\theta}_{p,0})^\top (\psi(a) - v) \right)} ,$$

where $v$ can be any vector in $\mathbb{R}^d$. Recall that we require $\pi_{\text{ref}}(\cdot|a)$ to be identical for all $a \in \mathcal{V}$ in the proof of Theorem 14. Here we further construct $\pi_{\text{ref}}$ to be uniform on the first token. Now observe

$$\log \frac{\mathbb{E}_{a \sim \pi_{\text{ref}}} \exp \left( (\mathbf{r}_{\text{sparse}} + \mathbf{r}_{\text{dense}})^\top (\psi(a) - v) \right)}{\mathbb{E}_{a \sim \pi_{\text{ref}}} \exp \left( (\hat{\theta}_{p,0})^\top (\psi(a) - v) \right)} = \log \frac{\sum_{a \in \mathcal{V}} \exp \left( (\mathbf{r}_{\text{sparse}} + \mathbf{r}_{\text{dense}})^\top (\psi(a) - v) \right)}{\sum_{a \in \mathcal{V}} \exp \left( (\hat{\theta}_{p,0})^\top (\psi(a) - v) \right)} .$$

Set $v$ to be $\frac{1}{|\mathcal{V}|} \sum_{a \in \mathcal{V}} \psi(a)$, then we have $\sum_{a \in \mathcal{V}} (\psi(a) - v) = 0$. Since $\Sigma_{\mathcal{D}}$ is already non-singular, we have that $\{\psi(a) - v\}_{a \in \mathcal{V}}$ can span $\mathbb{R}^d$. So we can directly apply Lemma 6, and get

$$\log \sum_{a \in \mathcal{V}} \exp \left( (\mathbf{r}_{\text{sparse}} + \mathbf{r}_{\text{dense}})^\top (\psi(a) - v) \right) - \log \sum_{a \in \mathcal{V}} \exp \left( (\hat{\theta}_{p,0})^\top (\psi(a) - v) \right)$$

$$\geqslant \langle (\mathbf{r}_{\text{sparse}} + \mathbf{r}_{\text{dense}}) - \hat{\theta}_{p,0}, \nabla_\theta \log \sum_{a \in \mathcal{V}} \exp \left( \theta^\top (\psi(a) - v) \right) \big|_{\theta = \hat{\theta}_{p,0}} \rangle + \frac{\kappa}{2} \left\| (\mathbf{r}_{\text{sparse}} + \mathbf{r}_{\text{dense}}) - \hat{\theta}_{p,0} \right\|_2^2$$

$$= - \mathbb{E}_{a \sim \pi_{\text{DPO}}} \left[ \left( \hat{\theta}_{p,0} - (\mathbf{r}_{\text{sparse}} + \mathbf{r}_{\text{dense}}) \right)^\top (\psi(a) - v) \right] + \frac{\kappa}{2} \left\| (\mathbf{r}_{\text{sparse}} + \mathbf{r}_{\text{dense}}) - \hat{\theta}_{p,0} \right\|_2^2 .$$

Therefore, we have

$$V_{r^\star}^{\pi^\star} - V_{r^\star}^{\pi_{\text{DPO}}} \geqslant \frac{\kappa}{2} \left\| (\mathbf{r}_{\text{sparse}} + \mathbf{r}_{\text{dense}}) - \hat{\theta}_{p,0} \right\|_2^2$$

$$= \frac{\kappa}{2} \left\| (\mathbf{r}_{\text{sparse}} + \mathbf{r}_{\text{dense}}) - \hat{\theta}_{p,0} \right\|_2^2 \|\Sigma_{\mathcal{D}}\|_2 \cdot \frac{1}{\|\Sigma_{\mathcal{D}}\|_2}$$

$$\geqslant \frac{\kappa}{2} \left\| (\mathbf{r}_{\text{sparse}} + \mathbf{r}_{\text{dense}}) - \hat{\theta}_{p,0} \right\|_2 \left\| \Sigma_{\mathcal{D}} \left( (\mathbf{r}_{\text{sparse}} + \mathbf{r}_{\text{dense}}) - \hat{\theta}_{p,0} \right) \right\|_2 \cdot \frac{1}{\|\Sigma_{\mathcal{D}}\|_2}$$

$$\geqslant \frac{\kappa}{2} \langle (\mathbf{r}_{\text{sparse}} + \mathbf{r}_{\text{dense}}) - \hat{\theta}_{p,0}, \Sigma_{\mathcal{D}} \left( (\mathbf{r}_{\text{sparse}} + \mathbf{r}_{\text{dense}}) - \hat{\theta}_{p,0} \right) \rangle \cdot \frac{1}{\|\Sigma_{\mathcal{D}}\|_2}$$

$$= \frac{\kappa}{2} \left\| (\mathbf{r}_{\text{sparse}} + \mathbf{r}_{\text{dense}}) - \hat{\theta}_{p,0} \right\|_{\Sigma_{\mathcal{D}}}^2 \cdot \frac{1}{\|\Sigma_{\mathcal{D}}\|_2}$$

$$= \Omega \left( \frac{d}{n} \cdot \frac{1}{\|\Sigma_{\mathcal{D}}\|_2} \right) .$$

The first inequality comes from Lemma 6; the second equality comes from the non-singularity of $\Sigma_{\mathcal{D}}$; the third inequality comes from a standard property of the spectral norm; the fourth inequality comes from Cauchy-Schwartz inequality; the fifth equality is a simple algebraic equality; and the last equation comes from Theorem 14.

## C.11 OMITTED CALCULATIONS

**Calculation of the sub-optimality with respect to the mis-specification error.**

First, note that for any $\pi \in \Delta(\mathcal{Y})$, we have:

$$
\begin{aligned}
V_{r^\star}^{\pi^\star} - V_{r^\star}^{\pi} &= \mathop{\mathbb{E}}_{y \sim \pi^\star}\left[ r^\star(y) - \beta \log \frac{\pi^\star(y)}{\pi_{\mathsf{ref}}(y)} \right] - \mathop{\mathbb{E}}_{y \sim \pi}\left[ r^\star(y) - \beta \log \frac{\pi(y)}{\pi_{\mathsf{ref}}(y)} \right] , \\
&= \mathop{\mathbb{E}}_{y \sim \pi^\star}\left[ r^\star(y) - \beta \log \frac{\pi^\star(y)}{\pi(y)} - \beta \log \frac{\pi(y)}{\pi_{\mathsf{ref}}(y)} \right] - \mathop{\mathbb{E}}_{y \sim \pi}\left[ r^\star(y) - \beta \log \frac{\pi(y)}{\pi_{\mathsf{ref}}(y)} \right] \\
&= -\mathsf{KL}\left( \pi^\star \| \pi \right) + \mathop{\mathbb{E}}_{y \sim \pi^\star, y' \sim \pi}\left[ \left( r^\star(y) - r^\star(y') \right) - \left( \beta \log \frac{\pi(y)}{\pi_{\mathsf{ref}}(y)} - \beta \log \frac{\pi(y')}{\pi_{\mathsf{ref}}(y')} \right) \right] \\
&\leqslant \mathop{\mathbb{E}}_{y \sim \pi^\star, y' \sim \pi}\left[ \left( r^\star(y) - r^\star(y') \right) - \left( \beta \log \frac{\pi(y)}{\pi_{\mathsf{ref}}(y)} - \beta \log \frac{\pi(y')}{\pi_{\mathsf{ref}}(y')} \right) \right] .
\end{aligned}
$$

We call it the performance difference lemma (Lemma 1 of Shi et al. (2025)).

For RLHF, we have:

$$
\begin{aligned}
V_{r^\star}^{\pi^\star} - V_{r^\star}^{\pi_{\mathsf{RLHF}}} &\leqslant \mathop{\mathbb{E}}_{y \sim \pi^\star, y' \sim \pi_{\mathsf{RLHF}}}\left[ \left( r^\star(y) - r^\star(y') \right) - \left( \beta \log \frac{\pi_{\mathsf{RLHF}}(y)}{\pi_{\mathsf{ref}}(y)} - \beta \log \frac{\pi_{\mathsf{RLHF}}(y')}{\pi_{\mathsf{ref}}(y')} \right) \right] \\
&\leqslant \max_{y, y' \in \mathcal{Y}}\left[ \left( r^\star(y) - r^\star(y') \right) - \left( \beta \log \frac{\pi_{\mathsf{RLHF}}(y)}{\pi_{\mathsf{ref}}(y)} - \beta \log \frac{\pi_{\mathsf{RLHF}}(y')}{\pi_{\mathsf{ref}}(y')} \right) \right] \\
&\leqslant \underbrace{\max_{y, y' \in \mathcal{Y}}\left[ \left( r^\star(y) - r^\star(y') \right) - \left( r_\phi(y) - r_\phi(y') \right) \right]}_{\text{reward model mis-specification error}} \\
&\quad + \underbrace{\max_{y, y' \in \mathcal{Y}}\left[ \left( r_\phi(x, y) - r_\phi(x, y') \right) - \left( \beta \log \frac{\pi_{\mathsf{RLHF}}(y|x)}{\pi_{\mathsf{ref}}(y|x)} - \beta \log \frac{\pi_{\mathsf{RLHF}}(y'|x)}{\pi_{\mathsf{ref}}(y'|x)} \right) \right]}_{\text{policy model mis-specification error}} ,
\end{aligned}
$$

where the first inequality is by performance difference lemma, and the last two inequalities are by symmetry and the properties of max. And if $\mathcal{F} \subseteq \mathcal{F}_\Pi$, by the definition of $\pi_{\mathsf{RLHF}}$, we have

$$
V_{r^\star}^{\pi^\star} - V_{r^\star}^{\pi_{\mathsf{RLHF}}} \leqslant \underbrace{\max_{y, y' \in \mathcal{Y}}\left[ \left( r^\star(y) - r^\star(y') \right) - \left( r_\phi(y) - r_\phi(y') \right) \right]}_{\text{reward model mis-specification error}} .
$$

For DPO, by performance difference lemma, we have:

$$
\begin{aligned}
V_{r^\star}^{\pi^\star} - V_{r^\star}^{\pi_{\mathsf{DPO}}} &\leqslant \mathop{\mathbb{E}}_{y \sim \pi^\star, y' \sim \pi_{\mathsf{DPO}}}\left[ \left( r^\star(y) - r^\star(y') \right) - \left( \beta \log \frac{\pi_{\mathsf{DPO}}(y)}{\pi_{\mathsf{ref}}(y)} - \beta \log \frac{\pi_{\mathsf{DPO}}(y')}{\pi_{\mathsf{ref}}(y')} \right) \right] \\
&\leqslant \underbrace{\max_{y, y' \in \mathcal{Y}}\left[ \left( r^\star(y) - r^\star(y') \right) - \left( \beta \log \frac{\pi_{\mathsf{DPO}}(y)}{\pi_{\mathsf{ref}}(y)} - \beta \log \frac{\pi_{\mathsf{DPO}}(y')}{\pi_{\mathsf{ref}}(y')} \right) \right]}_{\text{policy model mis-specification error}} \\
&= \underbrace{\max_{y, y' \in \mathcal{Y}}\left[ \left( r^\star(y) - r^\star(y') \right) - \left( \hat{r}_{\mathsf{DPO}}(y) - \hat{r}_{\mathsf{DPO}}(y') \right) \right]}_{\text{surrogate reward model mis-specification error}} .
\end{aligned}
$$

The first inequality is by performance difference lemma, the second inequality is by symmetry and the property of max, and the last equality is just another interpretation.

Therefore, we can see that the sub-optimality of each algorithm can be upper bounded by the linear model mis-specification error.

**Calculation of token-level structure of the optimal solution for DPO.** As motivated by Rafailov et al. (2024), we show the token-level structure of the optimal solution for DPO as:

$$
\pi^\star(y_t | y_{0 \ldots t-1}) = \pi_{\mathsf{ref}}(y_t | y_{0 \ldots t-1}) \exp\left( \frac{q^\star(y_t | y_{0 \ldots t-1}) - q^\star(y_{t-1} | y_{0 \ldots t-2})}{\beta} \right) ,
$$

$$\pi^\star(y_0) = \pi_{\mathsf{ref}}(y_0) \exp\left(\frac{q^\star(y_0) - \beta \log Z}{\beta}\right) \ ,$$

where $Z := \sum_y \pi_{\mathsf{ref}}(y) \exp(r^\star(y)/\beta)$, and the $q^\star$ function is determined in a recursive way:

$$q^\star(y_t|y_{0...t-1}) = \begin{cases} \beta \log \sum_{s \in \mathcal{V}} \pi_{\mathsf{ref}}(s|y_{0...t}) \exp(q^\star(s|y_{0...t})/\beta) & y_t \text{ is not the terminal token;} \\ r^\star(y_{0...t}) & y_t \text{ is the terminal token.} \end{cases}$$

To prove this, we define a $q'$ function as:

$$q'(y_0) = \beta \log Z + \beta \log \frac{\pi^\star(y_0)}{\pi_{\mathsf{ref}}(y_0)} \ , \ q'(y_t|y_{0...t-1}) = q'(y_{t-1}|y_{0...t-2}) + \beta \log \frac{\pi^\star(y_t|y_{0...t-1})}{\pi_{\mathsf{ref}}(y_t|y_{0...t-1})} \ .$$

For the initial token, by definition we have:

$$\pi^\star(y_0) = \pi_{\mathsf{ref}}(y_0) \exp\left(\frac{q'(y_0) - \beta \log Z}{\beta}\right) \ . \tag{9}$$

And then for a $y$ with $y_N$ as the terminal token, we have:

$$\begin{aligned} \beta \log \frac{\pi^\star(y)}{\pi_{\mathsf{ref}}(y)} &= \sum_{t=0}^N \beta \log \frac{\pi^\star(y_t|y_{0...t-1})}{\pi_{\mathsf{ref}}(y_t|y_{0...t-1})} \\ &= q'(y_0) - \beta \log Z + \sum_{t=1}^N q'(y_t|y_{0...t-1}) - q'(y_{t-1}|y_{0...t-2}) \\ &= -\beta \log Z + q'(y_N|y_{0...N-1}) \ . \end{aligned}$$

Note that $\pi^\star(y) = \pi_{\mathsf{ref}}(y) \exp(r^\star(y)/\beta)/Z$, we have:

$$\beta \log \frac{\pi^\star(y)}{\pi_{\mathsf{ref}}(y)} = -\beta \log Z + r^\star(y) \ ,$$

thus

$$q'(y_N|y_{0...N-1}) = r^\star(y) \ . \tag{10}$$

Then by definition:

$$q'(y_t|y_{0...t-1}) = q'(y_{t-1}|y_{0...t-2}) + \beta \log \frac{\pi^\star(y_t|y_{0...t-1})}{\pi_{\mathsf{ref}}(y_t|y_{0...t-1})} \ ,$$

we have:

$$\pi_{\mathsf{ref}}(y_t|y_{0...t-1}) \exp\left(\frac{q'(y_t|y_{0...t-1}) - q'(y_{t-1}|y_{0...t-2})}{\beta}\right) = \pi^\star(y_t|y_{0...t-1}) \ , \tag{11}$$

and thus

$$\sum_s \pi_{\mathsf{ref}}(s|y_{0...t-1}) \exp\left(\frac{q'(s|y_{0...t-1}) - q'(y_{t-1}|y_{0...t-2})}{\beta}\right) = 1 \ ,$$

which yields:

$$q'(y_{t-1}|y_{0...t-2}) = \beta \log \sum_{s \in \mathcal{V}} \pi_{\mathsf{ref}}(s|y_{0...t-1}) \exp(q'(s|y_{0...t-1})/\beta) \ . \tag{12}$$

Combining Equations (9) to (12), we show that $q^\star$ exists and is equivalent to $q'$.

**Calculation of the underlying "real" objective.** When ground-truth reward is non-realizable for the reward model, while the reward model is realizable for the policy model, for a given reward model $r_\phi$, the policy model outputs the policy $\pi_{\theta^\star(r_\phi)}$ which satisfies:

$$\pi_{\theta^\star(r_\phi)} := \operatorname*{argmax}_{\pi_\theta \in \Pi} V_{r_\phi}^{\pi_\theta} = \operatorname*{argmax}_{\pi_\theta \in \Pi} \mathbb{E}_{y \sim \pi_\theta} r_\phi(y) - \beta \mathsf{KL}\left(\pi_\theta \| \pi_{\mathsf{ref}}\right) \ .$$

The solution is given by:

$$\pi_{\theta^\star(r_\phi)}(y) = \frac{1}{Z(\phi)}\pi_{\mathsf{ref}}(y)\exp\left(\frac{1}{\beta}r_\phi(y)\right) \ ,$$

where $Z(\phi) := \sum_{y \in \mathcal{Y}} \pi_{\mathsf{ref}}(y)\exp(r_\phi(y)/\beta)$ is the partition function.

The goal of preference-based policy learning is to find a policy $\pi_\theta$ that maximizes $V_{r^\star}^{\pi_\theta}$. Thus, the reward learning should aim to find $r_\phi$ that maximizes:

$$\begin{aligned}
V_{r^\star}^{\pi_{\theta^\star(r_\phi)}} &= \mathop{\mathbb{E}}_{y \sim \pi_{\theta^\star(r_\phi)}}\left[r^\star(y) - \beta\log\frac{\pi_{\theta^\star(r_\phi)}(y)}{\pi_{\mathsf{ref}}(y)}\right] \\
&= \beta\log Z(\phi) + \mathop{\mathbb{E}}_{y \sim \pi_{\theta^\star(r_\phi)}}\left[r^\star(y) - r_\phi(y)\right] \ ,
\end{aligned}$$

which does not align with maximizing MLE.

Note that

$$\nabla_\phi\left\{\mathop{\mathbb{E}}_{y \sim \pi_{\theta^\star(r_\phi)}}\left[r^\star(y) - r_\phi(y)\right]\right\} = \underbrace{\mathop{\mathbb{E}}_{y \sim \pi_{\theta^\star(r_\phi)}}\nabla_\phi\log\pi_{\theta^\star(r_\phi)}[r^\star(y) - r_\phi(y)]}_{\text{term 1}} - \underbrace{\mathop{\mathbb{E}}_{y \sim \pi_{\theta^\star(r_\phi)}}\nabla r_\phi(y)}_{\text{term 2}} \ .$$

And we have:

$$\begin{aligned}
&\text{term 1} \\
&= \mathop{\mathbb{E}}_{y \sim \pi_{\theta^\star(r_\phi)}}\nabla_\phi\log\pi_{\theta^\star(r_\phi)}(y)[r^\star(y) - r_\phi(y)] \\
&= \mathop{\mathbb{E}}_{y,y' \sim \pi_{\theta^\star(r_\phi)}}\nabla_\phi\log\pi_{\theta^\star(r_\phi)}(y)[r^\star(y) - r^\star(y') - r_\phi(y) + r_\phi(y')] \qquad \text{(policy gradient theorem)} \\
&= \frac{1}{2}\mathop{\mathbb{E}}_{y,y' \sim \pi_{\theta^\star(r_\phi)}}\left[\nabla_\phi\log\pi_{\theta^\star(r_\phi)}(y) - \nabla_\phi\log\pi_{\theta^\star(r_\phi)}(y')\right]\left[r^\star(y) - r^\star(y') - r_\phi(y) + r_\phi(y')\right] \ ,
\end{aligned}$$

and

$$\begin{aligned}
&\text{term 2} \\
&= \mathop{\mathbb{E}}_{y \sim \pi_{\theta^\star(r_\phi)}}\nabla r_\phi(y) \\
&= \mathop{\mathbb{E}}_{y \sim \pi_{\theta^\star(r_\phi)}}\beta\nabla_\phi\left[\log\pi_{\mathsf{ref}}(y) + \log\exp(r_\phi(y)/\beta)\right] \\
&= \mathop{\mathbb{E}}_{y \sim \pi_{\theta^\star(r_\phi)}}\beta\nabla_\phi\left[\log\pi_{\mathsf{ref}}(y) + \log\exp(r_\phi(y)/\beta) - \log Z(\phi)\right] + \beta\nabla_\phi\log Z(\phi) \\
&= \mathop{\mathbb{E}}_{y \sim \pi_{\theta^\star(r_\phi)}}\beta\nabla_\phi\log\pi_{\theta^\star(r_\phi)}(y) + \beta\nabla_\phi\log Z(\phi) \\
&= \beta\nabla_\phi\log Z(\phi) \ . \qquad\qquad\qquad\qquad\qquad \text{(policy gradient theorem)}
\end{aligned}$$

By combining them, we obtain Equation (4) and Equation (5).

Note that

$$\mathcal{L}_{\mathrm{MLE}}(\phi) = -\mathop{\mathbb{E}}_{y,y' \sim \mu}\left[\sigma(r^\star(y) - r^\star(y'))\log\sigma(r_\phi(y) - r_\phi(y')) + \sigma(r^\star(y') - r^\star(y))\log\sigma(r_\phi(y') - r_\phi(y))\right] \ ,$$

and

$$\begin{aligned}
\nabla_q\left[\sigma(p)\log\sigma(q) + \sigma(-p)\log\sigma(-q)\right] &= \sigma(p)\sigma(-q) - \sigma(-p)\sigma(q) \\
&= \sigma(p)(1 - \sigma(q)) - (1 - \sigma(p))\sigma(q) \\
&= \sigma(p) - \sigma(q) \ ,
\end{aligned}$$

we have:

$$\nabla_\phi\mathcal{L}_{\mathrm{MLE}}(\phi) = -\mathop{\mathbb{E}}_{y,y' \sim \mu}\left[\nabla_\phi r_\phi(y) - \nabla_\phi r_\phi(y')\right]\left[\sigma(r^\star(y) - r^\star(y')) - \sigma(r_\phi(y) - r_\phi(y'))\right] \ ,$$

which is Equation ([6](#)).

To further align the MLE objective with the underlying "real" objective, we can have:

$$\nabla_\phi \mathcal{L}_{\text{MLE}}(\phi) \approx - \mathop{\mathbb{E}}_{y,y' \sim \mu} \left[ \nabla_\phi r_\phi(y) - \nabla_\phi r_\phi(y') \right] \sigma'(r_\phi(y) - r_\phi(y')) \left[ (r^\star(y) - r^\star(y')) - (r_\phi(y) - r_\phi(y')) \right] \, ,$$

and we can assign the value of $\sigma'(r_\phi(y) - r_\phi(y'))$ to the sampling probability $\mu(y, y')$. Thus we expect $\mu(y, y') \propto \pi_{\theta^\star(r_\phi)}/\sigma'(r_\phi(y) - r_\phi(y'))$. And under the context of DPO, we have $\pi_{\theta^\star(r_\phi)} = \pi_\theta$ and $r_\phi = \hat{r}_\theta$, and thus $\mu \propto \pi_{\theta^\star(r_\phi)}/\sigma'(\hat{r}_\theta(y) - \hat{r}_\theta(y'))$, which is exactly PILAF sampler.

**Calculation of online IPO.** For online IPO, let's observe its objective function:

$$\mathcal{L}_{\text{IPO}}^{\text{online}}(\pi_\theta) = \mathop{\mathbb{E}}_{(y,y') \sim \text{sg}(\rho_\theta)} p^\star(y > y') \left[ (r_\theta(y) - r_\theta(y')) - \frac{1}{2} \right]^2 + p^\star(y' > y) \left[ (r_\theta(y') - r_\theta(y)) - \frac{1}{2} \right]^2 \, ,$$

and its gradient is:

$$\nabla_\theta \mathcal{L}_{\text{IPO}}^{\text{online}}(\pi_\theta)$$

$$= 2 \mathop{\mathbb{E}}_{(y,y') \sim \text{sg}(\rho_\theta)} \left\{ p^\star(y > y') \left[ (r_\theta(y) - r_\theta(y')) - \frac{1}{2} \right] + p^\star(y' > y) \left[ (r_\theta(y) - r_\theta(y')) + \frac{1}{2} \right] \right\} \nabla_\theta (r_\theta(y) - r_\theta(y'))$$

$$= 2 \mathop{\mathbb{E}}_{(y,y') \sim \text{sg}(\rho_\theta)} \left[ (r_\theta(y) - r_\theta(y')) - \frac{p^\star(y > y') - p^\star(y' > y)}{2} \right] \nabla_\theta (r_\theta(y) - r_\theta(y')) \, ,$$

thus we have:

$$\mathcal{L}_{\text{IPO}}^{\text{online}}(\pi_\theta) \overset{\nabla}{=} \mathop{\mathbb{E}}_{(y,y') \sim \text{sg}(\rho_\theta)} \left[ (r_\theta(y) - r_\theta(y')) - \frac{p^\star(y > y') - p^\star(y' > y)}{2} \right]^2 \, .$$

# D    IMPLEMENTATION DETAILS

**Codebases.**    Our codebase is mainly based on MODPO (Zhou et al., 2024) (https://
github.com/ZHZisZZ/modpo), Online-RLHF (Dong et al., 2024; Xiong et al., 2024)
(https://github.com/RLHFlow/Online-RLHF), Samplers-in-Online-DPO (Shi et al.,
2025) (https://github.com/srzer/Samplers-in-Online-DPO). We are committed
to releasing the codes.

**Datasets.**   We adopt one common training dataset, PKU-SafeRLHF (Ji et al., 2023) (https:
//huggingface.co/datasets/PKU-Alignment/PKU-SafeRLHF).    *SFT:* We train
our initial model on 5k samples of PKU-SafeRLHF-QA (https://huggingface.co/
datasets/PKU-Alignment/PKU-SafeRLHF-QA). *Online training:* We use 10k samples of
PKU-SafeRLHF-Prompt (https://huggingface.co/datasets/PKU-Alignment/
PKU-SafeRLHF-prompt) for training, and 2k samples for evaluation. *Offline training:* We
adopt two preference datasets, PKU-SafeRLHF-safer and PKU-SafeRLHF-better, each
composed of 9k training samples and 2k evaluation samples, following the practice of Zhou et al.
(2024).

**Models.**   Limited by computation resources, our base model is **GPT-2-LARGE-774M** (Rad-
ford et al., 2019) (https://huggingface.co/openai-community/gpt2-large).
Our reward model is **GPT2-LARGE-HARMLESS** model (Yang et al., 2024) (https://
huggingface.co/Ray2333/gpt2-large-harmless-reward_model).

**Hyper-parameters.** The maximum length is set as 256. The prompt template is "BEGINNING OF
CONVERSATION: USER: [prompt] ASSISTANT: [response]". *SFT:* The hyper-parameter setting
is based on Dong et al. (2024). We use a batch size 32. *Online training:* The hyper-parameter
setting is based on Dong et al. (2024). We use a batch size 32, a learning rate $5e-7$, and a gradient
accumulation step 2. We train for 3 iterations, each for 2 epochs. We set $r_{\mathrm{margin}} = 0.4, 1, 4$ for
verifications of Condition 1, and set $r_{\mathrm{margin}} = 1$ for verifications of Conditions 2 to 4. *Offline
training:* The hyper-parameter setting is based on Zhou et al. (2024). We use a batch size 4, a
learning rate $1e-4$, and a gradient accumulation step 2. We train for 3 epochs (when training reward
model on 9k data of PKU-SafeRLHF-safer, we train 6 epochs for higher training accuracy). We
haven't extensively tuned these hyper-parameters.

**Computation resources.** Our experiments are conducted on NVIDIA RTX A6000. *SFT and Online
training:* We adopt 4 workers, each taking up $35,000$M of memory, running for 2-3 hours. *Offline
training:* We adopt 1 worker, which takes up $25,000$M of memory and runs for up to 40 minutes.

