# OpenReview forum: "Understanding the Performance Gap in Preference Learning: A Dichotomy of RLHF and DPO"
_ICLR.cc/2026/Conference — Submitted to ICLR 2026_

### Official Review · Reviewer_NeaD · 2025-10-18

**Soundness:** 1
**Presentation:** 1
**Contribution:** 2
**Rating:** 2
**Confidence:** 4

**Summary:**

This paper provides a theoretical framework for understanding the performance differences between DPO, Online DPO, and two-stage RLHF through the lens of (reward and policy) model misspecification. The authors systematically analyze four scenarios based on whether misspecification occurs in the policy model, the reward model, both, or neither. The key contribution lies in establishing theoretical conditions under which DPO and Online DPO can outperform RLHF. By examining how different types of misspecification affect algorithm performance, the paper offers insights on algorithm selection tailored to specific optimization contexts. Through empirical experiments comparing DPO and pairwise PPO across each misspecification scenario, the results make connections with the theoretical contributions.

**Strengths:**

1. **Addresses a significant problem in the RLHF community.** Understanding the gap between DAAs and RLHF is a crucial first step toward closing them and improving alignment algorithms in practice.

**Weaknesses:**

1. **Writing requires substantial improvement.** i) Several equations overflow in the preliminaries and theoretical analysis sections. ii) Key notations (e.g., $\rho$ for the regularized value function) are used without proper introduction. iii) : Section 3 should be restructured to highlight only the most significant theoretical contributions with in-depth analysis. Less critical conditions and propositions should be moved to the appendix. Focusing the main text on cases where DPO or Online DPO demonstrably outperform RLHF would strengthen the narrative.
2. **Empirical analysis is insufficient** i)only GPT-2 as the base model, ii) single harmless dataset, iii) only pairwise PPO for baseline comparison, iv) Both DPO and PPO are sensitive to hyperparameter choices. The paper does not demonstrate sufficient effort in hyperparameter tuning, which is critical for fair comparison.
3. **The claim that DPO outperforms RLHF is not rigorously established.** There is a fundamental difference in learning settings: DPO/(RLHF) is an offline/(online) algorithm. The underlying sampling distribution suggests that RLHF should set an upper bound for DPO performance in the absence of model misspecification. However, the authors claim an equivalence relationship between RLHF and DPO without rigorously defining or explaining this claim. The conditions under which this equivalence holds need clearer theoretical justification.

**Questions:**

1. **How do you justify your experimental design for inducing model misspecification?** In your experiments, you freeze certain parts of the LLM or reward model to create misspecification scenarios. However, how can you determine that full parameter tuning does not itself introduce model misspecification? This concern is particularly salient given that you are using models with fewer than 1B parameters, where capacity limitations may inherently lead to misspecification even with full fine-tuning.

---

> ### Author Response · Authors · 2025-11-17
>
> > **Reviewer's Summary**: This paper provides a theoretical framework for understanding the performance differences between DPO, Online DPO, and two-stage RLHF through the lens of (reward and policy) model misspecification. The authors systematically analyze four scenarios based on whether misspecification occurs in the policy model, the reward model, both, or neither. **The key contribution lies in establishing theoretical conditions under which DPO and Online DPO can outperform RLHF.** By examining how different types of misspecification affect algorithm performance, the paper offers insights on algorithm selection tailored to specific optimization contexts. **Through empirical experiments comparing DPO and pairwise PPO across each misspecification scenario, the results make connections with the theoretical contributions.**
>
>
> We appreciate the reviewer’s attempt to summarize our work, but several key aspects of our contribution were misrepresented or omitted. We clarify the main points below.
>
> (i) The summary overlooks an important component of our theoretical contribution. Beyond the misspecification taxonomy in Section 3, **Section 4 establishes a statistical separation between RLHF and DPO in a provable finite-sample setting**, showing that RLHF can achieve substantially better sample efficiency due to reward sparsity.
>
> (ii) The summary states that our “key contribution lies in establishing conditions where DPO or Online DPO can outperform RLHF.” This does not reflect our intended message. **The primary contribution of Section 3 is a comprehensive taxonomy** that characterizes combinations of reward and policy misspecification. Please see further clarifications in our response to W1 and W3.
>
> (iii) **We didn't compare DPO with pairwise PPO.** We are comparing DPO with a pairwise variant of policy gradient (PG) for the first set of experiments, and comparing DPO with reward modeling for the second set of experiments. Please see further clarifications in our response to W2.
>
> > W1: Writing requires substantial improvement.
>
> A:
>
> (i) Thank you for pointing it out. We didn’t break these formulas into separate lines due to page limit. Now we've fixed the equation overflows.
>
> (ii) In line 072, we introduced $\rho$ as arbitrary prefixed distribution over prompts, which in practice is the population distribution of the prompts.  Additionally, we are sorry that we unintentionly deleted the definition of $\pi_{\text{RLHF}}$ when editing the paper. We've added explanations of $r_{\text{RLHF}}$, $\rho$ and human preference dataset in Section 2. We've also refined the notations of the DTSP task and the linear parameterization. If there are still key notations used without proper introduction, please let us know.
>
> (iii) We respectfully disagree with the suggestion to restructure Section 3 around only the cases where DPO or Online DPO outperform RLHF. The purpose of Section 3 is precisely to provide a complete and rigorous characterization of all possible model mis-specification regimes. The central contribution of this section is its systematic taxonomy: by considering every combination of reward/policy model mis-specification, we obtain a unified view that explains when each method (RLHF, DPO, or Online DPO) can be optimal or suboptimal. Please see further clarifications in our response to W3.

---

> ### Author Response · Authors · 2025-11-17
>
> > W2: Empirical analysis is insufficient
>
> A:
>
> (i) and (ii): Please see our global response.
>
> (iii) and (iv): We didn't use pairwise PPO as baseline. Please see the "implementation details" paragraph of Section 5. To make the experiments principled and well-controlled, we compare DPO with pairwise PG, rather than PPO/GRPO variants that rely on numerous heuristics. We use the pairwise formulation because it's equivalent to vanilla PG in expectation (see derivations in our Appendix C.2 line 961-968), and its gradient form is very close to online DPO (see derivations in our Appendix C.2):
>
> $\nabla_\theta\mathcal L_{\text{RL}}(\theta)\propto-\mathbb E_{y,y'\sim\pi}[\nabla\log\pi_\theta(y)-\nabla\log\pi_\theta(y')][(r^\star(y)-r^\star(y'))-(\beta\log\frac{\pi_\theta(y)}{\pi_{\text{ref}}(y)}-\beta\log\frac{\pi_\theta(y')}{\pi_{\text{ref}}(y')})]$,
>
> $\nabla_\theta\mathcal L_{\text{DPO}}(\theta)\propto-\mathbb E_{y,y'\sim\pi}[\nabla\log\pi_\theta(y)-\nabla\log\pi_\theta(y')][\sigma(r^\star(y)-r^\star(y'))-\sigma(\beta\log\frac{\pi_\theta(y)}{\pi_{\text{ref}}(y)}-\beta\log\frac{\pi_\theta(y')}{\pi_{\text{ref}}(y')})]$.
>
> This allows us to keep the data and hyperparameters consistent across methods. As for the hyperparameters: for verifications of Section 3, we directly adopt the well-tuned hyperparameters from https://github.com/RLHFlow/Online-RLHF for DPO, and we use the same hyperparameters for the pairwise PG since its form is close to DPO; for verifications of Section 4, we directly adopt the well-tuned hyperparameters from https://github.com/ZHZisZZ/modpo for DPO and reward modeling. So there is no much need to heavily tune them.
>
> > W3: The claim that DPO outperforms RLHF is not rigorously established. DPO/(RLHF) is an offline/(online) algorithm. The online sampling suggests that RLHF should set an upper bound for DPO performance without model misspecification. However, the authors claim an equivalence relationship between RLHF and DPO without rigorously defining or explaining this claim. The conditions under which this equivalence holds need clearer theoretical justification.
>
> A: We respectfully clarify that our paper does not claim that DPO universally outperforms RLHF. Our theoretical results characterize the regimes in which RLHF, DPO, or online DPO can theoretically outperform others, and we explicitly show that in the fully realizable setting they are exactly equivalent. We summarize and clarify the conditions below to address the reviewer’s concern.
>
> **we do not claim DPO $>$ RLHF**
>
> Sections 3 shows that:
> (i) RLHF can outperform DPO under policy misspecification (Prop. 3--4),
> (ii) DPO can outperform RLHF under reward misspecification (Prop. 5), and
> (iii) neither method dominates under double misspecification (Prop. 6--9); and the comparison depends on the qualities of (surrogate) reward models.
>
> Thus our work does not assert a unilateral advantage for DPO.
>
> **Online vs. offline does not always imply an upper bound**
>
> The reviewer argues that RLHF should set an upper bound for DPO performance in the absence of model mis-specification because RLHF is online while DPO is offline. We respectfully clarify that offline vs. online does not always imply an upper bound. In Section 3.1 we have proven that both algorithms can reach the same class-optimal value when the reward and policy classes are realizable; In Section 3.3 we have proven that RLHF (online) could underperform DPO (offline) under reward model mis-specification.
>
> Here we briefly explain the equivalence of RLHF and DPO in the absence of model mis-specification. The basic setting is that offline DPO and the reward modeling stage of RLHF are conducted on the same data distribution (as shown in Section 2, the default distribution is $\pi_\text{ref}$). Therefore, if the reward model is perfectly learned, then the DPO policy model is also perfectly learned. The RLHF policy model and DPO policy model are thus both optimal and thus equivalent. The online sampling of RLHF here just accelerates the convergence rate. Additionally, DPO can also have online variant. In Section 3.1, we have proven that online DPO is approximating the objective of RLHF.
>
> To conclude, all rigorous definitions and conditions are provided in Section 2 and 3. And Figure 1 illustrates the conditions under which equivalence holds. Please let us know if you still have concerns.
>
> > Q1: How do you justify your experimental design for inducing model misspecification?
>
> A: Please see the "Setup" and "Verifications of Section 3" paragraphs of Section 5. To produce controllable results, our ground-truth reward signals are directly created by GPT2-large-harmless model, which induces a small linear function class.  We can thus expect a fully-tunable GPT2-large model to not be mis-specified as a policy model, because it has a strictly larger scale than GPT2-large-harmless (the LM head is larger than the RM head, and the features are also tunable for the policy model).

---

> > ### Comment · Reviewer_NeaD · 2025-11-26
> >
> > I thank the authorfor your detailed response. However, my major concerns remain aftering reviewing your responses.
> >
> > ### 1. Weak motivation and lack of practical relevance
> >
> > The paper investigates the performance gap between alignment algorithms under a representation gap (model misspecification). After revisiting the revised paper, I now share the concerns raised by Reviewer JrNc: the paper relies on strong and unrealistic assumptions that fundamentally weaken its motivation.
> >
> > While the representation gap is theoretically valid, the work fails to establish practical relevance. Then, what is the contribution of this paper?
> >
> >
> > ### 2. Limited and Disconnected Empirical Analysis
> >
> > The author states: *"To make the experiments principled and well-controlled, we compare DPO with pairwise PG, rather than PPO/GRPO variants that rely on numerous heuristics"*
> >
> > I respectfully disagree with this justification. PPO and GRPO are the standard online RLHF methods used in practice. Excluding them does not make your experiments more principled, it rather only disconnects them from both your theoretical claims and real-world practice. If your theory applies to online RLHF, your experiments must include these methods to validate your insights.
> >
> >
> > ### 3. Regarding the derivation of pairwise PG
> > Thank you for pointing to the derivation for pairwise PG. However, I have several concerns:
> > 1. Lines 964-965: Why does switching to the pairwise sampling distribution result in maximizing r*(y) - r̂(y) while minimizing r*(y') - r̂(y')? This transition needs clearer justification.
> > 2. Error in Line 964: The log-trick cannot be applied here because the reparameterized reward model r̂_θ explicitly depends on θ. The correct derivation requires the product rule:
> > ∇θ E{y~π_θ}[r*(y) - r̂_θ(y)] = Σ (∇π_θ) · [r*(y) - r̂_θ(y)] + π_θ · ∇[r*(y) - r̂_θ(y)]
> > 3. Line 968 from Line 966: Which symmetry property allows this derivation step? Please clarify.
> >
> >
> >
> > ### 4. The problem with your claims
> >
> > **Claim 1:** DPO is equivalently good as RLHF given no/both model mispecifications under the idealized setting (having access to infinite preference data and the optimization is without statistical or computational error)
> >
> > This is fundamentally incorrect. As DPO loss is estimated using the sampling distribution of some fixed policy (in your preliminary that is the reference policy), it is an offline learning algorithm. So DPO loss is restricted to the dataset distribution, having access to infinite preference data under the reference policy distribution does not make up for the gap between DPO and online RLHF.
> >
> > To conclude, online RLHF always set the performance upper bound for offline RLHF given no/both model mispecifications.
> >
> > **Claim 2:** Under the condition of reward model mispecification, DPO is better than RLHF
> >
> > This claim offers limited practical insight because the scenario is unrealistic: you assume 1) a sophisticated LLM architecture with sufficient capacity (ensuring policy realizability) and 2) a severely limited reward model with insufficient capacity to do reward modelling on infinite preferecen data.
> >
> > In practice, reward modeling is considered simpler than generative modeling. Reward models are typically finetuned from the same architecture as the policy, with comparable parameter sizes. So when would reward misspecification occur without policy misspecification?
> >
> > ### 5. Empirical model mis-specification setup
> > You state: *"ground-truth reward signals are directly created by GPT2-large-harmless model, which induces a small linear function class"*
> >
> > This suggests you finetune only the reward head of GPT-2-large with frozen base weights to create a restricted linear class. This raises a critical issue: such a restrictively finetuned model cannot serve as a meaningful ground truth.
> >
> > The empirical analysis requires substantial improvement to support your theoretical claims.

---

> ### Author Response · Authors · 2025-11-26
>
> Dear reviewer NeaD, we would first like to clarify a few points that may have been misunderstood.
> > Lines 964-965: Why does switching to the pairwise sampling distribution result in maximizing r*(y) - r̂(y) while minimizing r*(y') - r̂(y')? This transition needs clearer justification.
>
> This is a basic result of policy gradient theorem. Please note that given $y$, we have $\mathbb E_{y'\sim \pi_\theta}\nabla\log\pi_\theta(y)[r^\star(y')-\hat r_\theta(y')]=\nabla\log\pi_\theta(y)\cdot \mathbb E_{y'\sim \pi_\theta}[r^\star(y')-\hat r_\theta(y')]$. And thus $\mathbb E_{y\sim\pi_\theta,y'\sim \pi_\theta}\nabla\log\pi_\theta(y)[r^\star(y')-\hat r_\theta(y')]=[\mathbb E_{y\sim\pi_\theta}\nabla\log\pi_\theta(y)]\cdot [\mathbb E_{y'\sim \pi_\theta}[r^\star(y')-\hat r_\theta(y')]]=0$. Therefore, $\mathbb E_{y\sim\pi_\theta}\nabla\log\pi_\theta(y)[r^\star(y)-\hat r_\theta(y)]=\mathbb E_{y\sim\pi_\theta,y'\sim \pi_\theta}\nabla\log\pi_\theta(y)[(r^\star(y)-r^\star(y'))-(\hat r_\theta(y)-\hat r_\theta(y'))]$.
>
> > Error in Line 964: The log-trick cannot be applied here.
>
> This is a basic result of policy gradient theorem. Please note that $\hat r_\theta(y)=\beta\log\pi_\theta(y)-\beta\log\pi_{\text{ref}}(y)$, thus we have $\sum_y\pi_\theta(y)\nabla \hat r_\theta(y)=\beta\sum_y\pi_\theta(y)\nabla \log\pi_\theta(y)=\beta\sum_y\nabla \pi_\theta(y)=0$. Therefore, $\sum_y\pi_\theta (y)\nabla[r^\star(y)-\hat r_\theta(y)]=0$.
>
> > Line 968 from Line 966: Which symmetry property allows this derivation step? Please clarify.
>
> Symmetry means: $\mathbb E_{y\sim\pi_\theta,y'\sim \pi_\theta}\nabla\log\pi_\theta(y)[(r^\star(y)-r^\star(y'))-(\hat r_\theta(y)-\hat r_\theta(y'))]=\frac{1}{2}\mathbb E_{y\sim\pi_\theta,y'\sim \pi_\theta}\nabla\log\pi_\theta(y)[(r^\star(y)-r^\star(y'))-(\hat r_\theta(y)-\hat r_\theta(y'))]+\frac{1}{2}\mathbb E_{y'\sim\pi_\theta,y\sim \pi_\theta}\nabla\log\pi_\theta(y')[(r^\star(y')-r^\star(y))-(\hat r_\theta(y')-\hat r_\theta(y))].$ Then it equals to $\frac{1}{2}\mathbb E_{y\sim\pi_\theta,y'\sim \pi_\theta}[\nabla\log\pi_\theta(y)-\nabla\log\pi_\theta(y')][(r^\star(y)-r^\star(y'))-(\hat r_\theta(y)-\hat r_\theta(y'))].$
>
> > You state: "ground-truth reward signals are directly created by GPT2-large-harmless model, which induces a small linear function class." This suggests you finetune only the reward head of GPT-2-large with frozen base weights to create a restricted linear class.
>
> This doesn't suggest we finetune only the reward head of GPT-2-large with frozen base weights. We use an off-shelf reward model (https://huggingface.co/Ray2333/gpt2-large-harmless-reward_model), and they don't freeze the base weights during training.
>
> The reward model we use is nonlinear during training: the entire GPT-2-large backbone is unfrozen and updated, so the mapping $x \mapsto h_\theta(x)$ is a fully expressive nonlinear transformation learned end-to-end. Therefore the reward model is not trained as a restricted linear model.
>
> However, after the training process is completed, the reward model can be viewed as belonging to a linear function class. Once the transformer parameters $\theta$ are fixed, the reward takes the form $r(x) = w^\top h_\theta(x)$, which is a linear mapping from a fixed nonlinear feature $h_\theta(x)$. This analysis is standard for almost all  reward models in practice.
>
> Thus, our statement that the ground-truth reward model induces a “small linear function class’’ refers only to this post-training representation of the reward function, not to any restriction during training. The model is fully expressive during training, and only after the features are fixed can the reward function be abstracted as linear for theoretical analysis.
>
> If you still have any concerns about these points, please let us know so that we can provide further details.

---

> ### Author Response · Authors · 2025-11-26
>
> Here we respond to other concerns.
>
> > The paper investigates the performance gap between alignment algorithms under a representation gap (model misspecification). After revisiting the revised paper, I now share the concerns raised by Reviewer JrNc: the paper relies on strong and unrealistic assumptions that fundamentally weaken its motivation. While the representation gap is theoretically valid, the work fails to establish practical relevance. Then, what is the contribution of this paper?
>
> First we want to clarify that **the assumptions we used are all common assumptions in the preference learning theory community**:
>
> - The realizability assumption: It is standard in the theory of preference learning and RLHF, as seen in [1,2,3,4,5,7,8,9]. In Section 3, we mainly analyze cases where this assumption does not hold (e.g., policy model mis-specification, reward model mis-specification) to understand the consequences. This is one of our main contribution.
> - The bounded reward assumption: It is also standard in the community to prove bounds, as seen in [1,2,3,4,5,6,7,8,9]. In our paper, it is used only in Theorems 2, and 10.
> - The linear reward/policy parameterization and the static function assumption: It is a common assumption in preference optimization papers: many inspirations to NNs are derived from (log-)linear class, as seen in [1,2,3,4,5,6,8]. In our paper, it is used only in Section 4. Please see our response to W2 for further clarifications.
>
> And our analysis does not require the data coverage assumptions (Assumptions 4.1, 4.2 in [4] and Assumption 3.2 in [9]).
>
> [1] Zhao, Heyang, et al. "Sharp analysis for kl-regularized contextual bandits and rlhf." NeurIPS 2025.
>
> [2] Zhao, Heyang, et al. "Logarithmic Regret for Online KL-Regularized Reinforcement Learning." ICML 2025.
>
> [3] Xiong, Wei, et al. "Iterative preference learning from human feedback: Bridging theory and practice for RLHF under KL-constraint." ICML 2024.
>
> [4] Song, Yuda, et al. "The importance of online data: Understanding preference fine-tuning via coverage." NeurIPS 2024.
>
> [5] Yao, Yunzhen, et al. "Leveraging Sparsity for Sample-Efficient Preference Learning: A Theoretical Perspective." ICML 2025.
>
> [6] Nika, Andi, et al. "Reward Model Learning vs. Direct Policy Optimization: A Comparative Analysis of Learning from Human Preferences". ICML 2024.
>
> [7] Yang, Kunhe, et al. "Distortion of AI Alignment: Does Preference Optimization Optimize for Preferences?" NeurIPS 2025.
>
> [8] Zhu, Banghua, et al. "Principled Reinforcement Learning with Human Feedback from Pairwise or K-wise Comparisons." ICML 2023.
>
> [9] Xie, Tengyang, et al. "Exploratory Preference Optimization: Harnessing Implicit Q*-Approximation for Sample-Efficient RLHF." ICLR 2025.
>
> Then we would like to clarify that our Section 3 assumes exact optimization with infinite samples, which is idealized, but still leads to **insights that different methods are stronger in different settings.** This systematic result can help as a high-level guideline for practitioners, e.g. if you can obtain strong reward models, then you should use RLHF rather than DPO; if you can only train weak reward models due to limited resources, then you should use DPO rather than RLHF, which aligns with empirical observations of the community [10, 11].
>
> [10] Xu, Shusheng, et al. "Is DPO Superior to PPO for LLM Alignment? A Comprehensive Study." ICML 2025.
>
> [11] Ivison, Hamish, et al. "Unpacking DPO and PPO: Disentangling Best Practices for Learning from Preference Feedback." NeurIPS 2024.
>
> And we emphasize that **our Section 4 indeed gives results for the realistic finite sample case, and clearly proves a sample complexity separation. Again, this was not known before, and is our main contribution.** We hope the reviewer can acknowledge our contribution in establishing a separation in the approximate optimization setting.

---

> ### Author Response · Authors · 2025-11-26
>
> > In practice, reward modeling is considered simpler than generative modeling. Reward models are typically finetuned from the same architecture as the policy, with comparable parameter sizes. So when would reward misspecification occur without policy misspecification?
>
> We agree with the reviewer that in current post-training of frontier models, the reward model is often picked as sufficiently strong. We also emphasize this point in the paragraph "Observation under token-level parameterization" of Section 3.3, where we showed "the weak reward, strong
> policy model regime may be less common in practice", from the theoretic perspective of token-level parameterization.
>
> But even if the weak reward situation doesn’t happen in RLHF for current frontier models, this result is still helpful to know **if one wants to obtain smaller and cheaper reward models.** For example, if practitioners wish to post-train a 13B model using an existing preference dataset, they must decide whether to (i) train a reward model on that dataset and then perform RLHF, or (ii) directly apply DPO on the same dataset. Our analysis provides a principled understanding of how the relative capacity of the reward model affects performance in such scenarios, and therefore informs this design choice.

---

> ### Author Response · Authors · 2025-11-26
>
> > online RLHF always set the performance upper bound for offline RLHF given no/both model mispecifications.
>
> Please read our rebuttal to your W3 and Section 3.1/3.4, where we proved RLHF is equivalent to DPO given no/isomorphic model mis-speicfications. This is not an intuitive claim, we provide both intuitive explanations and theoretical proofs (proofs are in Appendix C.1 and C.5) under rigorous definitions.
>
> Here we want to emphasize again that **the reward modeling stage of RLHF are conducted on the same offline data distribution, so its performance is also restricted to the dataset distribution, though its policy optimization stage is online.** For example, under no model mis-specification, DPO policy is already optimal (please note that $\pi_{\text{ref}}(y)> 0,\forall y$, in frontier LMs, and thus the fixed distribution has a good coverage, that's why DPO will converge to an optimal policy) and thus RLHF cannot outperform DPO.
>
> Please note that **online sampling could enhance convergence rate, but here we only focus on the performance after convergence.** Additionally, DPO can also have online variant. In Section 3.1, we have proven that online DPO is approximating the objective of RLHF.
>
> In realistic finite-data or other misspecified regimes, RLHF can indeed outperform DPO, and our theory explicitly predicts this gap. But under the idealized assumptions (no/isomorphic model mis-specification and exact optimization), there is no performance gap.
>
> To conclude, this equivalence result contrasts with the common intuition in the community that “online RLHF always set the performance upper bound for offline RLHF given no/both model mispecifications.” The fact that our analysis rigorously characterizes when this intuition fails further underscores the contribution of Section 3.

---

> ### Author Response · Authors · 2025-11-26
>
> > PPO and GRPO are the standard online RLHF methods used in practice. Excluding them does not make your experiments more principled, it rather only disconnects them from both your theoretical claims and real-world practice. If your theory applies to online RLHF, your experiments must include these methods to validate your insights.
>
> Our theoretical results and experimental validations of Section 4 focus on the (surrogate) reward qualities, and thus don't involve RL. So our understanding is that the reviewer's concern is about the theoretical results of Section 3 and corresponding experiments. We therefore focus our reply on Section 3 and its experimental validations.
>
> First, we would like to clarify that our theory in Section 3 naturally includes RL variants like GRPO and PPO because we study exact optimization and the RL policy is directly $\pi_\text{RLHF}$=$\underset{\pi}{\text{argmax}}{V^\pi_{r_\text{RLHF}}}$, independent of the particular RL algorithm. Therefore, we didn't "disconnect them from our theoretical claims".
>
> Our main contribution of Section 3 is to provide a systematic taxonomy in theory, which has been proved rigorously. The experiments here are not intended to provide broad empirical claims; they are careful demonstrations that each theoretical scenario behaves exactly as predicted.
>
> It is well-known that PPO is equivalent to PG with clipped importance sampling ratio, adaptive KL penalties and a critic network, and GRPO is equivalent to PG with clipped importance sampling ratio to stabilize training. In our controlled setting, there is no much need to use these heuristics since the scale is small and training is already very stable. Additionaly, it is shown that clipping is rarely necessary for LMs post-training [3,4]. And PG (also known as REINFORCE) is already a very strong RL algorithm for LM post-training in practice [3]. Using PG rather than PPO/GRPO removes extra heuristics, making the experiments closer to the theoretical assumptions of exact optimization, and doesn't "disconnect them from real-world practice".
>
> Moreover, **empirical evidence from prior work already supports the predictions of Section 3 when PPO is used.** For example:
> - Table 6 of [1] demonstrates that PPO is stronger than DPO with a perfect reward signal, supporting our claim in Section 3.2. Their models are LLama 1/2 and Code Llama, and their benchmarks are SafeRLHF and APPS.
> - Table 5 of [2] demonstrates that PPO can beat DPO under a good reward model, supporting our claim in Section 3.2 and Section 3.4. Their models are Tulu2 series 8/13B.
> - Figure 2 of [5] demonstrates that DPO can beat PPO under a weak reward model, supporting our claim in Section 3.3 and Section 4. Their reward model is initialized from the gpt2-large model and only trained for 3 epochs on the preference
> datasets, and is thus weak.
>
> In summary, our theoretical results apply directly to PPO/GRPO; our PG-based controlled experiments are intentionally minimal to cleanly illustrate theoretical phenomena; and the broader empirical literature using PPO already corroborates the same predictions.
>
> [1] Xu, Shusheng, et al. "Is DPO Superior to PPO for LLM Alignment? A Comprehensive Study." ICML 2025.
>
> [2] Ivison, Hamish, et al. "Unpacking DPO and PPO: Disentangling Best Practices for Learning from Preference Feedback." NeurIPS 2024.
>
> [3] Ahmadian, Arash, et al. "Back to Basics: Revisiting REINFORCE-Style Optimization for Learning from Human Feedback in LLMs." ACL 2024.
>
> [4] Oertell, Owen, et al. "Heuristics Considered Harmful: RL With Random Rewards Should Not Make LLMs Reason."
>
> [5] Rafailov, Rafael, et al. "Direct Preference Optimization: Your Language Model is Secretly a Reward Model." NeurIPS 2023.
>
> Please let us know if you still have any concerns. And we will include these discussions in the revision if the reviewer is satisfied with them.

---

### Official Review · Reviewer_sYS3 · 2025-10-29

**Soundness:** 4
**Presentation:** 4
**Contribution:** 4
**Rating:** 8
**Confidence:** 3

**Summary:**

This paper analyzes the relative strengths of DPO, Online-DPO, and RLHF under reward model misspecification , policy misspecification, or both. The theoretical analysis in this work both differs significantly from prior work and is potentially of more practical interest to practitioners. The authors show that with a misspecified reward model, DPO is superior to RLHF; with a misspecified policy model, RLHF is superior to DPO, and when both models are misspecified there exist environments where either method is better. The authors then connect these findings to statistical efficiency when learning from limited data, showing that RLHF can reduce the estimation error relative to DPO. Finally, the authors present some empirical evidence corroborating their theoretical analysis.

**Strengths:**

This is really great work. The paper is excellently written and easy to follow. The claims are novel, interesting, and rigorous. The paper clearly fills a gap in existing literature focused on characterizing the differences between DPO and RLHF.

**Weaknesses:**

The empirical analysis could be more thorough, for example by evaluating with other preference datasets. I would also appreciate further analysis or explanation on the gap between DPO and RLHF in the right most plot of Figure 3 as this contradicts my intuition.

**Questions:**

(Does not impact my score)
The authors pose the question “What key property enables a (surrogate) reward model to subsequently help learn good policies?” and then answer this question with Eq. 3. I do not quite follow the connection, can you clarify?

---

> ### Author Response · Authors · 2025-11-17
>
> > W1: The empirical analysis could be more thorough, for example by evaluating with other preference datasets.I would also appreciate further analysis or explanation on the gap between DPO and RLHF in the right most plot of Figure 3 as this contradicts my intuition.
>
> A: Please see our global response.
>
> > Q1: The authors pose the question “What key property enables a (surrogate) reward model to subsequently help learn good policies?” and then answer this question with Eq. 3. I do not quite follow the connection, can you clarify?
>
> A:
>
> Eq.(3) reflects that the quality of a (surrogate) reward model can be characterized by the pairwise reward differences evaluated on an on-policy distribution.
>
> In Eq.(2), we see that RLHF and DPO can be unified as: first learning a reward model $r_\phi$, then getting a policy model $\pi_{\theta^\star(r_\phi)}$ which is optimal under $r_\phi$. So the key question is how can we learn the reward model $r_\phi$ to optimize $V_{r_\phi}^{\pi_{\theta^\star(r_\phi)}}$.
>
> Then Eq.(3) indicates that $V_{r_\phi}^{\pi_{\theta^\star(r_\phi)}}$ can be measured using a $\ell_2$ metric $\mathcal L(\phi):=\mathbb E_{y,y'\sim\text{sg}(\pi_{\theta^\star(r_\phi)})}[(r^\star(y)-r^\star(y'))-(r_\phi(y)-r_\phi(y'))]^2$. Therefore, to get a reward model to help learn good policies, we suffice to optimize $\mathcal L(\phi)$.
>
> In Appendix B, we then show that both DPO and reward modeling implicitly optimize approximations of this same objective. Their gradients approximate the form of $\nabla_\phi\mathcal L(\phi)$. We've included detailed discussions in Appendix B.
>
> Therefore, we can see that much of the design space in RLHF/DPO (e.g., online data collection or increasing model capacity) can be viewed as attempts to reduce the mismatch of the training objective with $\mathcal L(\phi)$.
>
> We hope this explanation makes the connection clearer.

---

### Official Review · Reviewer_JrNc · 2025-11-01

**Soundness:** 2
**Presentation:** 2
**Contribution:** 2
**Rating:** 2
**Confidence:** 5

**Summary:**

This paper provides a theoretical comparison between RLHF and DPO under model mis-specification.
It formalizes their performance difference using a KL-regularized bandit value function and decomposes it into representation and statistical gaps.
Under exact optimization, RLHF or DPO can outperform each other depending on whether the reward or policy model is more expressive.
Under finite-sample approximation, RLHF enjoys better statistical efficiency through reward sparsity.
However, the analysis relies on strong assumptions — realizability of the ground-truth reward, linear/tabular parameterization, and static function classes — which are unrealistic for large-scale LLM alignment.
The study also models preference learning as a single-step bandit problem, neglecting long-horizon, sequence-level dependencies in real LLMs.
Experiments on the PKU-SafeRLHF dataset qualitatively support the theory but are limited in scope and benchmark diversity.
Overall, the paper offers conceptual clarity on when RLHF or DPO might be preferable, but lacks practical validation under realistic LLM settings.
Its strength lies in theoretical framing; its weakness lies in empirical and modeling real-world.

**Strengths:**

The main strengths lie in its conceptual clarity and theoretical framing of alignment methods.

1. It presents a unified mathematical framework comparing RLHF and DPO via a KL-regularized value formulation, clarifying their equivalence and divergence under model mis-specification.
2. The analysis introduces a fine-grained decomposition of performance gaps—representation vs. statistical efficiency—which helps articulate the trade-offs between reward-first and direct optimization approaches.
3. Controlled experiments on the PKU-SafeRLHF dataset qualitatively support the theory, showing alignment between the proposed theoretical taxonomy and empirical behavior.

**Weaknesses:**

Despite its conceptual contributions, the work suffers from several limitations in practical realism and empirical grounding.

1. The analysis relies heavily on strong and unrealistic assumptions such as realizability, linear parameterization, and static function classes, which do not reflect modern LLM alignment settings.
2. Its sample-efficiency comparison is based on an artificial sparse-linear toy model (DTSP), limiting its external validity to large-scale, non-linear sequence models.
3. Empirical validation is narrow—restricted to a single dataset and small-scale models—without evaluation on standard alignment benchmarks or real-world LLM deployments.
4.  The writing is dense and not well organized, with frequent notational jumps and unclear figure references that obscure the main intuition behind the theoretical claims.

**Questions:**

The paper presents intuitively interesting findings on how RLHF and DPO differ under model mis-specification, but the theoretical and empirical support currently feels limited.

1. Could the authors explain why the experiments were conducted only on the PKU-SafeRLHF dataset? Was this choice due to computational constraints, or do the authors believe it sufficiently represents broader alignment scenarios?
2. Are there any additional experimental results (e.g., on different preference datasets or model scales) that could further validate the proposed theoretical claims?
3. The paper would benefit from clearer figures or visualizations that directly illustrate the theoretical insights (e.g., representation gap or sample-efficiency separation). Do the authors plan to provide additional figures or analyses that better connect the theory with experimental outcomes?

---

> ### Author Response · Authors · 2025-11-17
>
> > **Reviewer's Summary**: However, the analysis relies on strong assumptions — realizability of the ground-truth reward, linear/tabular parameterization, and static function classes — which are unrealistic for large-scale LLM alignment. The study also models preference learning as a single-step bandit problem, neglecting long-horizon, sequence-level dependencies in real LLMs.
>
> We appreciate the reviewer’s attempt to summarize our work. Below, we respectfully clarify several misunderstandings regarding our assumptions and scope.
>
> (i) **None of our results relies on tabular parameterization. Only the result in Section 4 requires linear parameterization and the realizability of the ground truth reward.** The main contribution of the exact-optimization analysis in Section 3 is precisely to go beyond tabular and relizability assumptions, which we highlight in the introduction: "the assumptions of tabular parameterization and realizability often do not hold in practice". Please see further clarification in our response to W1.
>
> (ii) **Our theoretical results are not restricted to single-step bandits.** The bandit model in Section 2 is just a general formulation $\pi\in\Delta(\mathcal Y)$ (neither assuming tabular bandit nor linear bandit) to write down the training objectives, and is widely used in almost every paper in this domain.
>
> Our analysis in Section 3 is horizon-agnostic: all value definitions, regularized optimization objectives, and mis-specification conditions hold for any policy class and any underlying decision process, including long-horizon, sequence-level tasks. None of the equivalence or separation results rely on the system being single-step. We also explicitly analyze token-level parameterization, which is long-horizon, in Section 3.3. Please see the "Observation under token-level parameterization" paragraph of Section 3.3 and the "Difference in token-level linear parameterization" paragraph of Section 4 for more details.
>
> Moreover, Section 4 explicitly addresses the reviewer’s concern. There, we highlight the unique challenges of multi-step MDPs, and prove that statistical separations between RLHF and DPO can arise only in settings with a sequential structure (we show that two-step setting is very different from one-step setting!). The minimalist construction in Section 4 is designed precisely to illustrate how MDP dynamics introduce qualitatively different behavior compared with the single-step model.
>
> > W1: The analysis relies heavily on strong and unrealistic assumptions.
>
> We provide explanations for each assumption we used:
>
> - The realizability assumption: It is standard in the theory of preference learning and RLHF, as seen in [1,2,3,4,5,7,8,9]. In Section 3, we mainly analyze cases **where this assumption does not hold** (e.g., policy model mis-specification, reward model mis-specification) to understand the consequences. This is one of our main contribution.
> - The bounded reward assumption: It is also standard in the community to prove bounds, as seen in [1,2,3,4,5,6,7,8,9]. In our paper, it is used only in Theorems 2, and 10.
> - The linear reward/policy parameterization and the static function assumption: It is a common assumption in preference optimization papers: many inspirations to NNs are derived from (log-)linear class, as seen in [1,2,3,4,5,6,8]. In our paper, it is used only in Section 4. Please see our response to W2 for further clarifications.
>
> And our analysis does not require the data coverage assumptions (Assumptions 4.1, 4.2 in [4] and Assumption 3.2 in [9]).
>
> [1] Zhao, Heyang, et al. "Sharp analysis for kl-regularized contextual bandits and rlhf." NeurIPS 2025.
>
> [2] Zhao, Heyang, et al. "Logarithmic Regret for Online KL-Regularized Reinforcement Learning." ICML 2025.
>
> [3] Xiong, Wei, et al. "Iterative preference learning from human feedback: Bridging theory and practice for RLHF under KL-constraint." ICML 2024.
>
> [4] Song, Yuda, et al. "The importance of online data: Understanding preference fine-tuning via coverage." NeurIPS 2024.
>
> [5] Yao, Yunzhen, et al. "Leveraging Sparsity for Sample-Efficient Preference Learning: A Theoretical Perspective." ICML 2025.
>
> [6] Nika, Andi, et al. "Reward Model Learning vs. Direct Policy Optimization: A Comparative Analysis of Learning from Human Preferences". ICML 2024.
>
> [7] Yang, Kunhe, et al. "Distortion of AI Alignment: Does Preference Optimization Optimize for Preferences?" NeurIPS 2025.
>
> [8] Zhu, Banghua, et al. "Principled Reinforcement Learning with Human Feedback from Pairwise or K-wise Comparisons." ICML 2023.
>
> [9] Xie, Tengyang, et al. "Exploratory Preference Optimization: Harnessing Implicit Q*-Approximation for Sample-Efficient RLHF." ICLR 2025.

---

> ### Author Response · Authors · 2025-11-17
>
> > W2: Its sample-efficiency comparison is based on an artificial sparse-linear toy model (DTSP), limiting its external validity to large-scale, non-linear sequence models.
>
> A: We would like to clarify that our results can extend beyond DTSP task. In the paragraph "Observation under token-level parameterization" of Section 3.3, we show that for non-terminal tokens, the optimal solution for pure reward learning is $(\theta_{r,t}^\star)^\top\psi(y_{0\ldots t})=(\theta_{t}^\star)^\top\psi(y_{0\ldots t})$ while the optimal solution for DPO training is $(\theta_{p,t}^\star)^\top\psi(y_{0\ldots t})=\log\mathbb E_{s\sim\pi_\textsf{ref}}\exp(q^\star(s\vert y_{0\ldots t})/\beta)$. Then the solution structure for DPO model doesn't align with its inductive bias, leading to potential model mis-specification (Recall that in section 3.2 we show that policy model mis-specification makes RLHF better than DPO).
>
> For general policy model class beyond log-linear model class, the Eq. (1) that $\pi^\star(y_t\vert y_{0\ldots t-1})\propto \pi_\textsf{ref}(y_t\vert y_{0\ldots t-1})\exp(q^\star(y_{0\ldots t})/\beta)$ still holds. This observation shows that the policy model must learn the $q^\star$ function, while the reward model only needs to learn the reward. Because $q^\star$ mixes both the reward and $\pi_\text{ref}$, the policy model faces a more complex target, making it more vulnerable to model mis-specification and sample inefficiency. And to prevent policy model mis-specification, $d_P$ is often required to be larger than $d_R$, which further leads to increased sample complexity. We have included these discussions in the "Concluding remarks" paragraph of Section 4.
>
> However, we admit that it is hard to provide rigorous statistical separation results without log-linear assumption.
>
>
> > W3: Empirical validation is narrow without evaluation on standard alignment benchmarks or real-world LLM deployments.
>
> A: Please see our global response.
>
> > W4: The writing is dense and not well organized, with frequent notational jumps and unclear figure references that obscure the main intuition behind the theoretical claims.
>
> A: Thanks for pointing it out. We've added explanations of $r_{\text{RLHF}}$, $\rho$ and human preference dataset in Section 2. We've refined the notations of the DTSP task and the linear parameterization. We've refined the notations of the DTSP task. We've fixed some equation overflows. We also added texts to Section 5 that explicitly explain how each figure corroborates our theory.
>
> > Q1: Could the authors explain why the experiments were conducted only on the PKU-SafeRLHF dataset?
>
> A: Please see our global response.
>
> > Q2: Are there any additional experimental results (e.g., on different preference datasets or model scales) that could further validate the proposed theoretical claims?
>
> A: Please see our global response.
>
> > Q3: Do the authors plan to provide additional figures or analyses that better connect the theory with experimental outcomes
>
> A: Please see our response to W4.
>
> If you have any additional concerns, please do not hesitate to let us know. We are more than willing to address them and sincerely appreciate your valuable feedback and support.

---

### Official Review · Reviewer_uT9m · 2025-11-03

**Soundness:** 2
**Presentation:** 2
**Contribution:** 3
**Rating:** 4
**Confidence:** 4

**Summary:**

This paper provides a comprehensive theoretical analysis of Reinforcement Learning from Human Feedback (RLHF) and Direct Preference Optimization (DPO) under various settings, including both exact and approximate optimization. The authors compare RLHF and DPO under different types of misspecification, showing that RLHF is optimal under policy misspecification, while DPO is optimal under reward model misspecification. They further present empirical results to support and validate their theoretical findings.

**Strengths:**

a. This paper provides a theoretical analysis comparing RLHF and DPO under different model misspecification settings, a topic that has been rarely explored in the existing literature. The analysis offers valuable insights into the performance gap between RLHF and DPO.

b. The paper also discusses two optimization scenarios, including both the ideal (exact) and practical (approximate) optimization settings. The authors provide a statistical analysis for both RLHF and DPO, contributing to a more comprehensive understanding.

**Weaknesses:**

a. The paper appears to have been written somewhat hastily, and the presentation could be improved. For instance, in Section 2, the authors mention that RLHF proceeds in two stages but only describe the first (reward learning) stage, omitting the second (policy learning) stage. As a result, $\pi_{\textrm{RLHF}}$ is never formally defined, which makes it more difficult to follow the subsequent theorems.

b. The experimental evaluation is limited to a relatively small model (GPT-2-774M). Given that current practice relies on much larger models, the validity and generality of the experimental verification are questionable.

c. Although the paper discusses different misspecification settings, it does not analyze suboptimality with respect to the misspecification error. This weakens the understanding of the performance gap between RLHF and DPO.

d. In the more important approximate optimization setting, the authors do not provide a theoretical analysis of the performance gap as they do in the exact optimization case, but instead rely only on an empirical proxy. A more concrete statistical characterization of the performance under approximate optimization would make the results stronger.

**Questions:**

Please refer to the weaknesses section. I am open to increasing my score if the concerns are addressed.

---

> ### Author Response · Authors · 2025-11-17
>
> > W1: The paper appears to have been written somewhat hastily, and the presentation could be improved.
>
> A: We are sorry that we unintentionly deleted the definition of $\pi_{\text{RLHF}}$ when editing the paper. We've added an explanation of this notation along with explanations of $\rho$ and human preference dataset in Section 2. We've refined the notations of the DTSP task and the linear parameterization. We've fixed some equation overflows. We also added texts to section 5 that explicitly explain how each figure corroborates our theory.
>
> > W2: The experimental evaluation is limited to a relatively small model (GPT-2-774M).
>
> A: Please see our global response.
>
> > W3: The paper doesn't discuss suboptimality with respect to the misspecification error.
>
> A: Thanks for pointing it out. We will first explain the hardness of analyzing the suboptimality w.r.t. the mis-specification error, and then present a upper bound result.
>
> Prior work offered a suboptimality result with an $\epsilon_{\text{app}}$ term as shown in table 1 of [1], which, however, is not tight. We have to admit that it is hard to qualitatively analyze the suboptimality with respect to the misspectification error. Recall our unification of RLHF and DPO (see Eq. (2)), we only need to compare the qualities of (surrogate) reward models, and the main difference lies in the fact that $r_{\text{RLHF}}\in \mathcal F$ while $\hat r_{\text{DPO}}\in\mathcal F_\Pi$ (as defined in Section 3.4). As shown in our Prop. 8 and 9, $\mathcal F\subseteq \mathcal F_\Pi$ cannot guarantee $\pi_{\text{DPO}}$ is better, and $\mathcal F_\Pi\subseteq \mathcal F$ cannot guarantee $\pi_{\text{RLHF}}$ is better. Our key insight is that the reward modeling MLE objective is not the real underlying objective, so sometimes reward model with strong representation power can lead to overoptimization. Therefore, the sub-optimality is affected by both the model mis-specification error and the data distribution. These discussions are all covered in Section 3.
>
> But we can provide a qualitative analysis on bounding the suboptimality. We can show that $V_{r^\star}^{\pi^\star}-V_{r^\star}^{\pi_{\text{RLHF}}}\le\max_{x,y,y'}\left[(r^\star(x,y)-r^\star(x,y'))-(r_{\text{RLHF}}(x,y)-r_{\text{RLHF}}(x,y'))\right]+\max_{x,y,y'}\left[(r_{\text{RLHF}}(x,y)-r_{\text{RLHF}}(x,y'))-(\beta\log\frac{\pi_{\text{RLHF}}(y\vert x)}{\pi_{\text{ref}}(y\vert x)}-\beta\log\frac{\pi_{\text{RLHF}}(y'\vert x)}{\pi_{\text{ref}}(y\vert x)})\right]$ (bounded by reward model mis-specification error and policy model mis-specification error), and
> $V_{r^\star}^{\pi^\star}-V_{r^\star}^{\pi_{\text{DPO}}}\le\max_{x,y,y'}\left[(r^\star(x,y)-r^\star(x,y'))-(\hat r_{\text{DPO}}(x,y)-\hat r_{\text{DPO}}(x,y'))\right]$ (bounded by policy model mis-specification error). Therefore, the sub-optimality is upper bounded by the linear mis-specification error. We've included this analysis in Section 3 and Appendix C.11.
>
> Our another effort to address this challenge is Eq.(3), which accurately measures the suboptimality. Yet how to practically leverage this metric is left as a future direction. Currently it only serves a intuitive guidance for the empirical proxy.
>
> > W4: In the approximate optimization setting, the authors do not provide a theoretical analysis of the performance gap as they do in the exact optimization case, but instead rely only on an empirical proxy.
>
> A: Thanks for your suggestion. Now we've provided a separation on the sub-optimality of RLHF and DPO, which is based on the separation on the empirical proxy. The performance gap is:
>
> $V_{r^\star}^{\pi^\star}-V_{r^\star}^{\pi_{\textup{RLHF}}}=\tilde{\mathcal O}\left(\sqrt[4]{\frac{k\log d}{n}}\cdot\sqrt{\Lambda_1}\right)$ and $V_{r^\star}^{\pi^\star}-V_{r^\star}^{\pi_{\textup{DPO}}}=\Omega\left(\frac{d}{n}\cdot \Lambda_2\right)$,
>
> where $\Lambda_1,\Lambda_2$ are data-dependent coefficients, which are $\mathcal O(1)$ when the data features are uniformly distributed. There is a strict separation when $n=\mathcal O(d)$ because $d\gg k$. Please refer to our Theorem 11 in the revised Section 4 for more details.
>
> [1] Nika, Andi, et al. "Reward Model Learning vs. Direct Policy Optimization: A Comparative Analysis of Learning from Human Preferences". ICML 2024.
>
> If you have any additional concerns, please do not hesitate to let us know. We are more than willing to address them and sincerely appreciate your valuable feedback and support.

---

### Author Response · Authors · 2025-11-17
**Global Response**

We thank all the reviewers for their insightful comments. We have addressed all your individual comments. We also appreciate the reviewers' recognition of the novelty and theoretical contributions of our work:
- "a topic that has been rarely explored in the existing literature", "valuable insights", "contributing to a more comprehensive understanding" (uT9m);
-  "excellently written and easy to follow", "the claims are novel, interesting, and rigorous", "clearly fills a gap in existing literature" (sYS3);
- "strength lies in theoretical framing", "intuitively interesting findings" (JrNc);
- "addresses a significant problem in the RLHF community", "a crucial first step" (NeaD).

Most concerns raised by reviewers relate to writing clarity and experimental presentation.

**Writing.**

We have revised our paper to improve clarity:
- added more explanations for notations and definitions;
- refined notations of the DTSP task and the linear parameterization;
- fixed equation overflows;
- added texts to explain how each figure corroborates our theory.

We also add discussions on
- qualitatively bounding the sub-optimality using model mis-specification error (Section 3),
- how the insights generalize to general model parameterizations (Section 4),
- how to establish a theoretical separation in sub-optimality (Section 4).

The updates are marked in orange.

---

> ### Author Response · Authors · 2025-11-17
>
> **Experiments.**
>
> Our primary contributions are theoretical, and the purpose of the experiments is to validate the qualitative predictions of our theoretical analysis, but not to serve as large-scale empirical benchmarks. We intentionally use controlled, small-scale setups for two reasons:
> (1) to keep experimental conditions interpretable and aligned with the assumptions in each theoretical scenario, and
> (2) to avoid obscuring the predicted effects with noise from large-scale training dynamics.
>
> Our experiments consist of two parts, each directly tailored to support the corresponding theoretical claims:
>
> 1. Verifying Section 3: In this part, we conduct controlled experiments to simulate the different model mis-specification scenarios analyzed in our theory. The empirical results precisely align with our theoretical predictions:
> - Figure 2 supports the claims in Theorem 2. It shows that even with no model mis-specification, as the reward scale increases, the second-order deviation in the online DPO objective grows, giving RLHF a clear advantage.
> - Figure 3 (left) validates Propositions 3 and 4. Under policy mis-specification, RLHF consistently outperforms DPO, as predicted by our theory.
> - Figure 3 (middle) validates Proposition 5. Under reward mis-specification, the trend reverses, and DPO outperforms RLHF.
> - Figure 3 (right) is consistent with the principle from Equation (2). Under double mis-specification, performance depends on the quality of the learned (surrogate) reward. The plot shows DPO outperforming RLHF when the mis-specified reward model is severely weak (in this case we deploy a GPT-2 model with all layers frozen except the last block, which is severely mis-specified).
>
> 2. Verifying Section 4: Figure 4 verifies our theory that two-stage reward learning of RLHF can be more sample-efficient than DPO. The plots show that as the number of training samples decreases, the evaluation accuracy of pure reward learning (RLHF's first stage) degrades much more slowly than that of surrogate reward learning (DPO) across two different preference types.
>
> Section 3 establishes performance comparisons under different forms of model mis-specification, and these results do not rely on any assumptions about the explicit reward/policy form. The experiments here are not intended to provide broad empirical claims; they are careful demonstrations that each theoretical scenario behaves exactly as predicted.
>
> Section 4 provides a minimalist example in which we can rigorously prove a statistical separation, and then offers an intuitive extension to more general cases. Since the broader regime is not theoretically characterized, we provided further empirical verifications
>
> Although our experiments are intentionally small-scale, our conclusions are supported by several recent large-scale empirical studies [1,2,3,4,5]:
>
> - Table 1 of [1] show that DPO is worse than RLHF when learning the reward, supporting our claim in Section 4. Their models are  Gemma-2-2B-IT, Qwen-2.5-1.5B-Instruct, Qwen-2.5-3B-Instruct, Llama-3.2-1B-Instruct, Llama3.2-3B-Instruct, and Llama-3.1-8B-Instruct, and the datasets are UltraFeedback and RewardMATH.
> - Figure 4 of [2] supports our claim in Section 3.3 that policy model requires larger scale to prevent mis-specification; Figure 6 of [2] supports our claim in Section 3.2 that online DPO cannot improve DPO much when the small policy model is mis-specified. Their model is pythia-1.4B.
> - Table 6 of [3] demonstrates that RLHF is stronger than DPO with a perfect reward signal, supporting our claim in Section 3.2. Their models are LLama 1/2 and Code Llama, and their benchmarks are SafeRLHF and APPS.
> - Table 5 of [4] demonstrates that RLHF can beat DPO under a good reward model, supporting our claim in Section 3.2 and Section 3.4. Their models are Tulu2 series 8/13B.
> - Figure 2 of [5] demonstrates that DPO can beat PPO under a weak reward model, supporting our claim in Section 3.3 and Section 4. Their reward model is initialized from the gpt2-large model and only trained for 3 epochs on the preference datasets, and is thus weak.
>
>
> These large-scale studies support the qualitative trends predicted in our theoretical analysis.
>
> [1] Razin, Noam, et al. "Why is Your Language Model a Poor Implicit Reward Model?"
>
> [2] Swamy, Gokul, et al. "All Roads Lead to Likelihood: The Value of Reinforcement Learning in Fine-Tuning."
>
> [3] Xu, Shusheng, et al. "Is DPO Superior to PPO for LLM Alignment? A Comprehensive Study." ICML 2025.
>
> [4] Ivison, Hamish, et al. "Unpacking DPO and PPO: Disentangling Best Practices for Learning from Preference Feedback." NeurIPS 2024.
>
> [5] Rafailov, Rafael, et al. "Direct Preference Optimization: Your Language Model is Secretly a Reward Model." NeurIPS 2023.

---

### Meta-Review · Area_Chair_K4YT · 2026-01-06

**Summary:**

This paper studies why RLHF and DPO can behave differently even when they are both trained from pairwise preferences. As reviewers also agree, It is an interesting research question.
The main framing is a representation gap, and the paper splits it into two regimes. In the exact optimization regime, it characterizes how the relative expressiveness of the reward model class and the policy class affects the best achievable policy, and it shows that RLHF, DPO, and online DPO can each be preferred depending on the type of misspecification. In the finite sample regime, it argues that an additional, implicit gap appears because DPO effectively learns a surrogate objective that can be statistically harder to estimate than a direct reward model, and it provides a construction where RLHF needs fewer samples than DPO when the ground truth reward has sparse structure. The paper also includes empirical checks on a preference dataset to support the qualitative takeaways.

While the claims are very general, the analyses appear to rely on specific constructions.
The paper utilizes a specific multi armed bandit example, where reward model does not appear to be necessarily standard. The AC is not familiar with token-level linear parameterization or how relevant it is in the related literature. More discussion on this aspect could be helpful. The construction in Section 4 is also fairly tailored, and it is not obvious from the theorem statements alone how much of the phenomenon survives under more standard parameterizations or other structured rewards.

**Reviewer Concerns:**

The evaluation signal for this paper in this stage is unfortunately weaker than usual because two of the reviews were identified as unreliable and containing hallucinated (or clearly incorrect assessments), which the authors also acknowledged. The AC therefore put little weight on those reviews when forming a decision, and the AC focused on the remaining reviews and their own (limited) reading (limited because of having to overseeing a large of number of papers).

Among the reviews, the feedback appears split, with some reviewers valuing the framing and the theoretical results, and others questioning whether the paper’s headline message is supported strongly enough given how stylized the cleanest examples are. The revised paper and rebuttal seem to have improved the presentation issues, but the remaining concerns are more about generality.
A more thorough discussion and justification may be provided on this.

**Reviewer Scores:**

The main reservation is that the paper’s framing reads quite general, but the clearest strict separations rely on constructions that may be rather specific.
Given the rebuttal and the revised presentation, the AC would expect reviewers whose main concerns were clarity and organization to have moved slightly upward. However, it is hard to expect large score swings from reviewers focused on the scope and generality of the key separations, because those concerns are not purely presentational. Overall, it is suspected that the post discussion score distribution would still look split and borderline rather than converging to a clear accept.

Given that there is not a sufficient amount of support signal to push the paper above the acceptance threshold, it appears that another round of reviews would be necessary to properly validate the strength and generality of the main claims. While the core message is interesting and potentially valuable, the current evaluation record is weakened by the lack of reliable reviewer signal and by the fact that several substantive concerns hinge on how broadly the stylized theoretical constructions should be interpreted. It is a tough call and the AC understands that this would be disappointing to the authors but believes that the paper can certainly benefit from explicit/thorough discussions.

Also, there appear to be recent related work which the authors are encourage to include comparisons in the revision (though the current submission was not penalized for not including the references)
https://arxiv.org/pdf/2507.07981
https://arxiv.org/pdf/2510.20413

---

### Decision · Program_Chairs · 2026-01-26

Reject